# Development of crystal orientation fabric, microstructures and deformational regimes in the deep sections and overall layered structures of the Dome Fuji ice core, Antarctica

Tomotaka Saruya[1], Atsushi Miyamoto[2], Shuji Fujita[1,3], Kumiko Goto-Azuma[1,3], Motohiro Hirabayashi[1], Akira Hori[4], Makoto Igarashi[1], Yoshinori Iizuka[5], Takao Kameda[4], Hiroshi Ohno[4], Wataru Shigeyama[3*], Shun Tsutaki[1,3]

[1] National Institute of Polar Research, Tokyo 190-8518, Japan
[2] Institute for the Advancement of Graduate Education, Hokkaido University, Sapporo 060-0817, Japan
[3] Polar Science Program, Graduate Institute for Advanced Studies, SOKENDAI, Tokyo 190-8518, Japan
[4] Kitami Institute of Technology, Kitami 090-8507, Japan
[5] Institute of Low Temperature Science, Hokkaido University, Sapporo 060-0819, Japan

*Currently at: JEOL Ltd., Tokyo, Japan

**Correspondence**: Tomotaka Saruya (saruya.tomotaka@nipr.ac.jp), Atsushi Miyamoto (miyamoto@high.hokudai.ac.jp), Shuji Fujita (sfujita@nipr.ac.jp)

An in-depth examination of the rheology in the deep sections of polar ice sheets is crucial for understanding glacial flow. This study investigates the crystalline textural properties of a 3035-meter-long Antarctic deep ice core, drilled at an inland plateau dome, with a focus on the lowermost ~20%. Inland plateau domes provide a unique opportunity to study ice deformation processes with minimal influence from simple shear. We analyze the crystal orientation fabric of both $c$- and $a$-axes, comparing them with various other properties measured from the ice core. To examine the distribution and texture of the $c$- and $a$-axes, we employed three methods: the dielectric tensor method (DTM) for bulk properties of thick sections, Laue X-ray diffraction for detailed crystal grain orientations, and an automatic ice fabric analyzer for grain-by-grain analysis using thin sections. Microstructural observations were made using optical microscopy. DTM provided preferred $c$-axis orientations as eigenvalues with high sampling frequency, spatial resolution, and continuity, offering statistical significance. Laue X-ray diffraction clarified detailed preferred orientations of both $c$- and $a$-axes, while the automatic fabric analyzer tracked $c$-axis orientation variations among individual grains. Microstructural images revealed dynamic recrystallization. In the uppermost ~80% thickness zone (UP80%), the clustering strength of single-pole $c$-axis fabric steadily increased, reaching a maximum at the bottom, driven primarily by vertical compression. Below 1800 m in the UP80%, layers with varying dusty impurities showed different rates of cluster strength growth, a pattern that persisted until 2650 m. In the lowermost ~20% zone (LO20%), the $c$-axis clustering strength trend changed around 2650 m, with substantial fluctuations below this depth. Impurity-rich layers maintained stronger clustering, while impurity-poor layers showed relaxation, likely due to the emergence of new grains with $c$-axis orientations offset from the existing cluster and migration recrystallization. These impurity-poor layers exhibited features of bulging and migrating grain boundaries, with a decrease in grain aspect ratio. Additionally, in such ice, the $a$-axis

organization progressed, displaying one or two sets of three preferred orientations within the *a*-axis girdle, orthogonal to the *c*-axis cluster, possibly due to crystal twinning. These findings confirm that dislocation creep is the primary deformation mechanism in polar ice sheets. In the LO20%, dynamic recrystallization plays a critical role, with more pronounced effects in impurity-poor layers than in impurity-rich layers, enabling the continuation of dislocation-creep-based deformation and forming *a*-axis organization within the *a*-axis girdle, particularly under higher temperatures closer to the bed. Additionally, *c*-axis layers and cluster axes rotate meridionally due to rigid-body rotation from simple shear strain above subglacial slopes. Insights into these nonlinear, irreversible processes offer vital clues for understanding the 3D structure of polar ice sheets, leading to inhomogeneous deformation, the formation of folds, faults, and mixing between layers at various thickness scales.

# 1. Introduction

## 1.1 Modeling ice sheets: challenges and insights from crystals to continents

The polar ice sheets are massive bodies of ice on Earth. With ongoing global warming, there is a deep concern about the trend of these ice sheets contributing to sea level rise (e.g., Church et al., 2013). Continuous improvement of reliable predictive models for this phenomenon is crucial. However, there are many key processes that govern the flow of ice sheets, making the modelling of these sheets complex. Although many essential processes exist, many are not included in modelling efforts due to insufficient understanding (e.g., Pattyn et al., 2008). Deciphering the dynamic layer structure within the ice sheet is one of the most critical challenges (e.g., Young et al., 2017).

## 1.2 Crystal anisotropy and ice sheet dynamics

The response of individual ice crystals to stress exhibits a pronounced anisotropy in deformation. Crystals readily undergo shearing along the slip systems within their basal planes, whereas shearing on alternate slip systems is significantly more difficult, nearly a hundredfold (e.g., Duval et al., 1983). The overall rate of deformation in a polycrystalline aggregate under stress is influenced by the orientation of its constituent crystals. If the crystals within an aggregate are randomly oriented, the material will behave isotropically; however, if the orientations of the crystals are not random, the material will show anisotropic behavior. The study of anisotropic ice deformation has been conducted historically. They were done through theoretical research (e.g., Alley, 1992; Azuma, 1995; Azuma and Goto-Azuma, 1996; Castelnau et al., 1996; Gödert and Hutter, 1998; Johnson, 1977), and through laboratory experiments (e.g., Duval, 1981; Duval and Le Gac, 1982; Shoji and Langway Jr., 1985; Pimienta et al., 1988; Budd and Jacka, 1989; Castelnau et al., 1998; Goldsby and Kohlstedt, 2001; Qi et al., 2019; Fan et al., 2020). These laboratory-based studies are characterized by experimental setting of strain rate by more than several orders of magnitude, and under temperature close to melting point, typically between -20 and 0 ℃. Therefore, these laboratory-based knowledge can be a valuable reference mainly for such conditions. To better understand rheology of polar ice sheets, analyses of ice texture sampled from polar ice sheets play another essential role. Examples of Antarctic deep ice core include, Dome Fuji (hereinafter, DF) ice core (e.g., Azuma et al., 2000 and Saruya et al., 2022b), EPICA Dome C (hereinafter, EDC) ice core (e.g., Wang et al., 2003, Durand et al., 2007, 2009), Byrd ice core (Gow and Williamson, 1976) , EDML ice core (e.g., Weikusat et al., 2017), Mizuho Station ice core (Fujita et al., 1987; Higashi et al., 1988),  Talos Dome ice core (Montagnat et al., 2012), WAIS Divide ice core (Fitzpatrick et al., 2014) and South Pole (SPICE) ice core (Abbasi et al., 2024). Examples of Greenland ice sheet include GRIP ice core (Thorsteinsson et al., 1997), NEEM ice core (Montagnat et al., 2014), EGRIP ice core (Stoll et al., 2021a, 2022; Richards et al., 2023) and among others. In these studies, the influence of anisotropy on the movement of ice aggregates is so substantial that comprehending its impact on the expansive flow patterns of ice sheets is essential (e.g., Castelnau et al., 1998; Mangeney et al., 1997; Paterson, 1991; Russell-Head and Budd, 1979; Thorsteinsson et al., 1999). For instance, during the internal deformation of the ice sheet, the flow forms preferred orientations of crystal axes, at the same

time, the ice flow is modulated based on them. The ice flow is also modulated by the concentration of ionic species and
microparticles, as well as disturbances or folds in the layer structure (e.g., Cuffey and Paterson, 2010, Durand et al., 2007,
Saruya et al., 2022b). These factors can introduce either positive or negative feedback, modulating the flow characteristics of
the ice sheet. Considering this complex nature of ice, we need a systematic understanding of the layered internal structure of
polar ice sheets over space and time. An important method for understanding these is to conduct analysis of ice cores in terms
of ice dynamics, which provides a "ground truth" perspective of the ice.

## 1.3 Advantages of ice cores from dome regions in central plateau area of the ice sheets

The ice in the dome summit regions of central plateau areas on ice sheets offers an opportunity for studying ice deformation
processes. In these regions, conditions are relatively simplified because it can be presumed that they are least affected by
simple shear deformation. This deformation is typically caused by surface slope and gravity. Because of this simplicity, dome
regions in central area of the ice sheets are often chosen as ice core drilling sites for the purpose of exploring the history of
past climate changes. At the same time, these sites are ideal for investigating how processes related to deformational progress
(such as dislocation creep and recrystallization) develop in a relatively simplified stress-strain environment and under moderate
temperature gradients. Among the drilling sites in Antarctica shown in Figure 1, we can presume that so far only DF and EDC
sites meet this condition. In fact, in the case of inland domes, ice deforms mainly by compression along the vertical. This is
confirmed by the preferred orientation of $c$-axes measured along cores drilled at these sites (as in references listed above),
which exhibit a single pole distribution of the $c$-axes. Similarly, this has been confirmed in the Greenland ice sheet with the
GRIP core (Thorsteinsson et al., 1997). We note that there are also local domes at the edge of the ice sheet plateau, for example,
Talos Dome (Urbini et al., 2008). Such a dome is characterized by lower elevation (of 2318 m), location facing to the coast,
with annual accumulation rate much larger than those of central plateau (by 3~4 times), and migration at least in the last few
centuries. Such a dome has higher instability, and it is suitable to investigate impact of more shear in contrast to the domes in
the central plateau area. Evolution of preferred orientation of $c$-axes for the Talos Dome core was presented by Montagnat et
al. (2012).

## 1.4 Exploring deep ice rheology and crystal properties

In ice sheets, the rheology in deep ice (deepmost several hundred meters) is complicated by geothermal effects and
increased stresses from bedrock topography. So far, there have been limited reports on the crystal properties of ice near the
bed at dome summits of central plateau of the ice sheets (e.g., Thorsteinsson et al., 1997; Durand et al., 2009, Faria et al.,
2014a, Ohno et al., 2016). The DF ice core was drilled on the Antarctic plateau (Figure 1). This location is at 77°19' S latitude
and 39°42' E longitude, and 3,810 m above sea level. Annual surface mass balance (SMB) has ranged from approximately 24
to 28 kg m$^{-2}$ y$^{-1}$ over the last 5000 years (Oyabu et al., 2023), and the annual mean surface air temperature is –54.4°C
(Yamanouchi et al., 2003). For the DF ice core, Ohno et al. (2016) revealed the air hydrates and water isotope composition of

the deepest 1% thickness (3000–3035 m). These properties were found to retain the basic layered structure of ice core signals except in the deepest few meters. However, the investigation did not include the crystalline textural properties. For the EDC core (Figure 1), Durand et al. (2009) investigated the $c$-axis orientations of ice down to the very deep parts, using thin sections of ice (40 × 110 mm in width and 0.4 mm in thickness) sampled at every 11 m depth.

Deep ice is characterized by higher temperature, under which enhancement of molecular transport phenomena is an issue (e.g., Petrenko and Whitworth, 1999). The molecular transport phenomena include plastic deformation, the molecular diffusion process and recrystallization. Dynamic recrystallization within ice sheet has been widely investigated historically, as reviewed in papers or textbooks (e.g., Poirier 1985; Humphreys and Haterly 2004; Faria et al., 2014a, b) or individual papers (e.g., De La Chapelle et al. 1998; Weikusat et al. 2009, Kipfstuhl et al. 2009, Montagnat et al. 2012, 2014; Stoll et al., 2021a). Exploring deep ice rheology and crystal properties, dynamic recrystallization is one of main focuses in this paper.

## 1.5 Advancing ice sheet dynamics: The development and role of crystal orientation fabric

The crystal orientation fabric contains information of deformational history, grain growth and recrystallization (e.g., Cuffey and Paterson, 2010; Faria et al., 2014a, b). Saruya et al. (2022b) investigated the layer structures of the $c$-axis fabric in the uppermost (shallower) ~80% thickness zone (hereinafter, UP80%) of the DF ice core. They used thick sections of ice (ranging in thickness from 33 to 79 mm) employing the principle of the radio-wave birefringence to obtain information from a volume much larger than that of thin sections of ice in typical thickness of about 0.3–0.5 mm. The sampling volumes between the two methods (thick-section method and thin-section method) differ by around two orders of magnitude. Using the dielectric tensor method (hereinafter, DTM), they continuously measured "dielectric anisotropy" (denoted as $\Delta\varepsilon$) along the ice cores with high resolution, involving ice volumes of cylinders measuring 16–38 mm in effective radio beam diameter and 33–79 mm in radio propagation thickness. This $\Delta\varepsilon$ is defined as the difference in relative permittivity along the vertical (denoted as $\varepsilon_v$) and in the horizontal plane (denoted as $\varepsilon_h$). Thus, $\Delta\varepsilon = \varepsilon_v - \varepsilon_h$. $\Delta\varepsilon$ is an indicator of the clustering strength of $c$-axes. $\Delta\varepsilon$ is linearly compatible with the normalized eigenvalues of the second order tensor (Saruya et al., 2022a, b). The normalized eigenvalues are often used to express the degree of the $c$-axes clustering. Thus, continuous profiling with DTM is an innovative and powerful tool to directly measure the normalized eigenvalues for preferred orientation of $c$-axes with symmetry (such as single pole, girdle type or superposition of them). These authors found that $\Delta\varepsilon$ steadily increases with depth, showing fluctuations in the UP80%. In addition, significant decreases in $\Delta\varepsilon$ were found at depths of major glacial to interglacial transitions. These changes in $\Delta\varepsilon$ are explained as variations in the deformational history of the vertical compression. Moreover, fluctuations in $\Delta\varepsilon$ along neighboring depths were enhanced during the glacial/interglacial transitions. Furthermore, the $\Delta\varepsilon$ data exhibited a positive correlation with the concentration of chloride ions and an inverse correlation with the amount of dust particles. Since these factors originate from atmospheric deposition, Saruya et al. (2022b) proposed that the fluctuations in the clustering strength might be common across wide area of ice sheets. This hypothesis for the Dome Fuji core, in part, differs from an existing hypothesis for the EDC ice core given by Durand et al. (2007). That is, the changes in the clustering strength (which increase

sharply with depth) were associated with positive feedback between variations in ice viscosity and the impact of a shear stress component, also increasing with depth (Durand et al., 2007). A basic difference is that Saruya's hypothesis did not need to assume the presence of shear deformation as a prerequisite to explain the observed fluctuations of the clustering strength whereas Durand et al. (2007) suggested the impact of shear and the positive feedback was important.

Following this previous study, research focusing on the lowermost (deepest) approximately 20% thickness zone (ranging from about 2400 m to the deepest ice at 3035 m) remains to be done, to examine textural data for the entire thickness of the ice sheet. Hereinafter, we refer to this zone as LO20%. A major difficulty was the inclination of the ice sheet layers, which became steeper at deeper depths (Dome Fuji Ice Core Project Members hereinafter referred to as DFICPM, 2017). In the present study, we extend the investigation to the LO20%. To investigate this, we utilized the thick-section-based method of (i) DTM, and thin-section-based methods such as (ii) the Laue X-ray diffraction method and (iii) the microstructural observations. We compare textural data by these methods with various data analyzed from the ice core. This paper provides an advanced and most updated dataset of crystal orientation fabric for both *c*-axis and *a*-axis, and other textural data for the entire thickness of the ice sheet at DF. We then discuss development of crystal orientation fabric, variations in the microstructure, layered structures, and rheology of polar ice sheets, addressing differences in the existing hypotheses. Our new data will provide important clues for understanding the development of the layer structure of ice sheets, which leads to inhomogeneous deformation of layers across various layer thickness scales.

Furthermore, retrieving continuous ice core records that correspond to ages of more than 1 million years presents a significant challenge in palaeoclimatology (e.g., Wolff et al., 2022). International Partnerships in Ice Core Sciences (IPICS) identified the retrieval of multiple ice cores that extends to 1.5 million years (symbolically named as the "oldest-ice cores") as one of the most important challenges for ice core studies. Identifying suitable sites for drilling of such ice will require improved knowledge of englacial layers under various ice conditions.

## 2. DF ice cores

At DF, deep ice cores have been drilled twice (e.g., Motoyama et al., 2020). The first core, "DF1" measuring 2503 m in length, was drilled in the 1990s. The second, "DF2," a 3035-m-long core, was drilled between 2004 and 2007 in a borehole 44 m away from the previous borehole (Figure 1c). We used the DF2 core in this study. The drilling sites are situated above a subglacial slope, positioned between bedrock high in the east and a subglacial trough with an ice thickness of about 3100 m in the west (Figures 1b and 1c). An ice thickness of about 2850 m marks the boundary between thawed bed (in thicker ice) and a frozen bed (in thinner ice) (Fujita et al., 2012). It is speculated that spatially inhomogeneous basal melting has caused the layer inclination in localized areas within the ice sheet (DFICPM, 2017). The layered structures incline by less than about 5° in the UP80% while in the lower 20% (LO20%), the inclination reaches values up to 45° at a depth of 3000 m.

In terms of the *c*-axis fabric, Azuma et al. (1999, 2000) reported that at DF1 the *c*-axis fabric exhibited the elongated single pole fabric as the dome undergoes deviatoric strain depending on orientations. This elongated single pole fabric is already

observable at shallow depths. In addition, Fujita et al. (2006) discussed, based on polarimetric radar sounding at Dome Fuji,
the amount of radio wave birefringence caused by this elongated single pole, up to depths of about 2200 m. They demonstrated
that the orientations of the elongated single pole are consistent at least to this depth. Figure 1c shows inferred two principal
axes of the elongated *c*-axis fabric, aligning with the orientation of the subglacial slope (WNW) and its orthogonal direction.

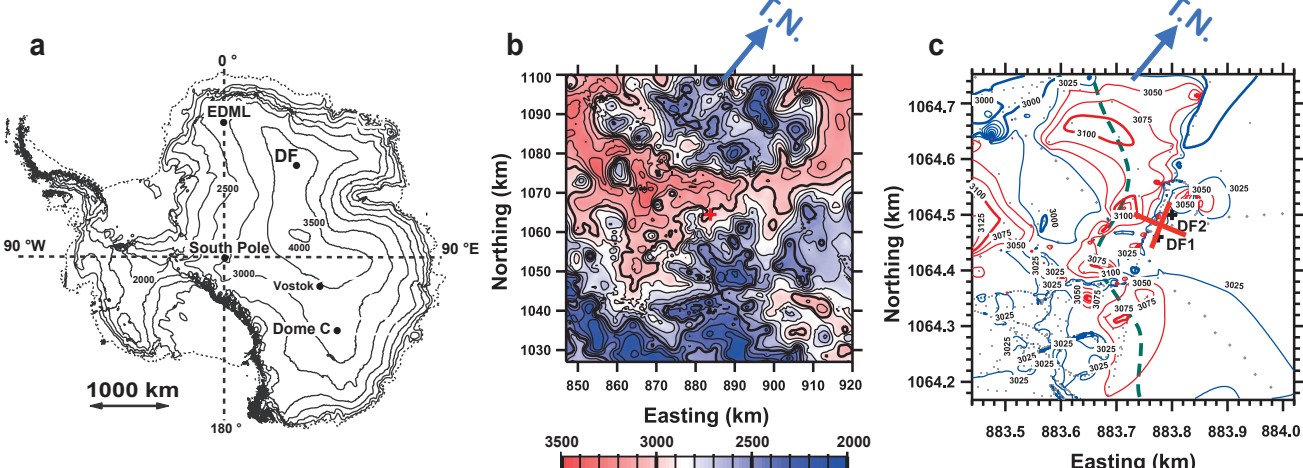

**Figure 1.** Maps of the Coring Site. (a) Surface elevation (DEM by Bamber et al., 2009). (b) and (c) show the thickness of the ice in areas of
72 km square, and 580 m square, respectively. Marker symbols near the center indicate the coring site. T.N. stands for true north. In (c), data
points for ice thickness are represented by dots for the data from Tsutaki et al. (2022) and by markers for the data from Eisen et al. (2020).
A green bold dashed line indicates the presence of the local trough. The DF coring sites are located at the bank of this local trough, aligning
with the estimated drainage routes of subglacial water (as shown in Figure 8d in Tsutaki et al., 2022). On the Figure 1c, two principal axes
of elongated single pole fabric inferred from the polarimetric radar sounding (Fujita et al., 2006) is indicated.

## 3. Methods and samples

We utilized the thick-section-based method of DTM, as well as the thin-section-based methods of the Laue X-ray
diffraction method and the microstructural observation. On one hand, using DTM, *c*-axis fabric data are provided as
eigenvalues with high sampling frequency, high spatial resolution, and continuity, thereby offering statistical significance.
However, this method does not produce data for the crystal axes of individual grains. Additionally, if a cluster of *c*-axes is
significantly inclined from the vertical, it is impossible to derive correct eigenvalues without knowing both the inclination
angle of the *c*-axis cluster and the horizontal orientation of the *c*-axis cluster. On the other hand, the Laue X-ray diffraction
method allows us to clarify detailed information about both the *c*-axis and the *a*-axes of each crystal grain. Additionally, the
automatic fabric analyzer, model G50, originally manufactured by Russel–Head Instruments, is a very useful method for
demonstrating how crystal orientation differs from one grain to another. Using both the G50 and microscopy, we conducted
microstructural observations to investigate the signals of dynamic recrystallization and grain morphology. The judicious use
of these methods provides unprecedentedly rich information on crystalline texture along ice cores.

### 3.1 Dielectric Tensor Method

### 3.1.1 Method

The principle of the open resonator method for determining the relative permittivity ($\varepsilon$) of thin samples have been described
in the literature (Jones, 1976a, b; Cullen, 1983; Komiyama et al., 1991). We have developed this into a method for measuring
the tensorial permittivity of thick samples by using radio birefringence continuously scanning along ice cores (Matsuoka et al.,
1998; Fujita et al., 2009, 2014, 2016; Saruya et al., 2022a, b; Inoue et al., 2024). We apply a microwave beam to thick samples.
The $\varepsilon$ values are volume-weighted averages within the volume encompassed by the Gaussian distribution of the beam. By
setting the angle between the axis of the $c$-axes cluster and the electric field to approximately 45°, radio birefringence occurs
due to macroscopic permittivity of the crystals (e.g., Hargreaves 1978). When we sweep radio frequencies to detect resonances
satisfying TEM $_{0,0,q}$ modes (where TEM stands for transverse electromagnetic and q is an integer), we detect twin resonant
peaks caused by the two permittivity components. These two components correspond to $\varepsilon$ along the axis of the $c$-axes cluster
and its orthogonal axis, representing components on the plane of the electric field vector (orthogonal to the axis of beam
incidence).

### 3.1.2 The open resonators and samples

We utilized an open resonator, employing frequencies between 26.5 and 40 GHz (No.1 in Table A1) for the ice in the
LO20%. This resonator is different from the resonator used for the ice in the UP80%, employing frequencies between 15 and
20 GHz (No.2 in Table A1). Specifications of the two resonators are summarized in Table A1 in Appendix A1. The two
resonators, each having a semi-confocal shape with a flat mirror and a concave mirror, are designed to produce beam diameter
of 16 mm and 38 mm, respectively. The consistency of the data obtained with the two resonators was confirmed by comparing
the $\Delta\varepsilon$ values at depths between 2400 and 2500 m (Figure A1 in Appendix A1). The smaller beam size means that the No.1
resonator is available for smaller sized samples (for example, narrow quadrangular prism samples) and for higher spatial
resolution measurements. The No.1 and the No.2 resonators can measure ice thickness at least 40 mm and 90 mm, respectively.
Sizes of the samples are given in Figure 2 and Table A2. Both upper and lower surfaces were microtomed to make very smooth
and precisely parallel surfaces. The sampling rate for the DTM involves continuous 20 mm step measurements along a 0.5-m-
long core segment from every 2.5-m depth interval. For samples deeper than 2736 m, we rotated the core axis horizontally in
the open resonator setup to achieve detectable twin resonances. The experimental temperatures were in a range of $-30 \pm 1.5$ °C.
Along with the increased layer inclination, the axis of the $c$-axes cluster also exhibited an increased inclination, deviating
from the vertical (hereinafter, inclination angle) in the same direction of the maximum layer slope. We confirmed that these

two are in the same vertical plane throughout the LO20%. Additionally, the horizontal orientation of the $c$-axes cluster varied with depths due to the rotation of the ice cores. Under conditions where both inclination angle and horizontal orientation of the $c$-axes cluster change with depths, we measure only the non-principal components of the permittivity tensor within the ice using DTM. To apply geometrical corrections from the measured non-principal components to the principal components, we utilized information of both inclination angle and horizontal orientation of the $c$-axes cluster, derived from $c$-axis fabric data measured using thin-section method. The procedures for these corrections are detailed in Appendix B. The non-principal components of the dielectric anisotropy ($\Delta\varepsilon'$) were adjusted to align with the principal components ($\Delta\varepsilon$). We also note that the DTM is also useful to detect $\Delta\varepsilon$ values in the girdle type ice fabrics. We can refer to basic principle in Appendix A2, as girdle distribution of $a$-axes in the figure.

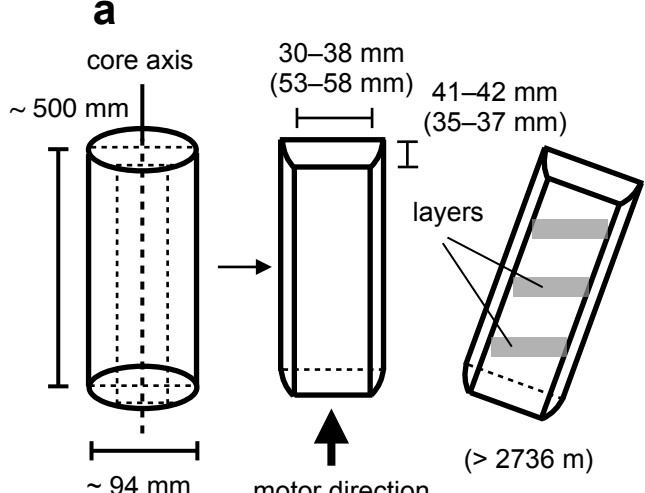

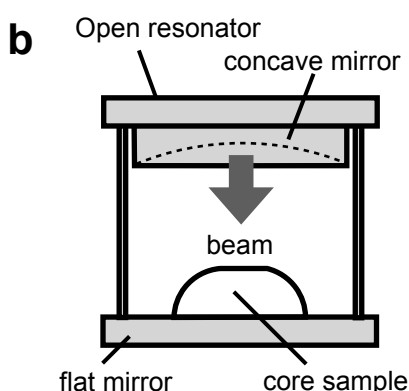

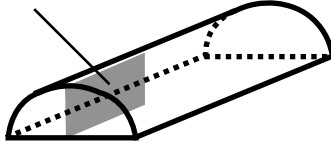


**Figure 2.** Diagrams of the (a) core cutting geometry, (b) setup viewed from the front, and (c) location of a thin section for the Laue X-ray
diffraction method.
**3.2 Laue X-ray diffraction method, microstructure observations and automatic fabric analyzer method**
The Laue X-ray diffraction method was applied to thin sections measuring 100 mm × 45 mm (in the vertical and horizontal
direction, respectively, with a thickness of less than 0.5 mm. This method can determine the orientations of all axes of each

crystal grain with accuracy of ~0.5 degree (Miyamoto et al., 2011). 42 depths within the thickness of the LO20% were selected. After obtaining all the Laue figures, the Laue patterns were analyzed. In each thin section, we express the data as the distribution of *c*-axes and *a*-axes on the Schmidt net diagram. In addition, we calculated the median inclination of *c*-axes with reference to the axis of the *c*-axes cluster. The median inclination is defined as a half apex angle of the cone in which a half of the measured *c*-axes are included from the center axis of the *c*-axes cluster.

Additionally, we conducted microstructure observations using both G50 and optical microscopy to investigate how microstructural evolutions affect the *c*-axis fabric development and fluctuation on the same thin section. These measurements were performed at several selected depths. We prepared thin sections from the vertical plane of the ice cores, measuring 90 mm in the depth direction, 50 mm in the horizontal direction, and 0.5 mm in thickness. In addition, images of *c*-axis fabric were obtained using G50 automatic fabric analyzer to investigate the relationship between grain morphology and *c*-axis orientation of each grain.

Grain numbers included in a Gaussian beam or in a thin section are listed in Table A3. Depending on number of crystal grains in a beam or thin section, statistical significance is different. In some cases, moving averaging along the core is useful to gain more significance.

**3.3 Grain size and layer inclination measurements**

Grain sizes were measured at the DF site. Operators from the ice processing team observed the crystal grains by noting a faint difference in light reflectivity between grains and visible grain boundaries. While freshly cut surfaces did not exhibit such features, the ice cores in storage developed them over time due to the progress of sublimation. To measure the average surface area of each grain, we used three circular gauges with diameters of 10, 20, and 40 mm. The number of grains on the core surface within the circle of the gauge was counted. The number of crystal grains within each circle typically ranged from several to 20. The estimated error was up to 20%. This measurement was performed every 1.5 m to a depth of 2967.5 m. Below this depth, because of very large crystals, it was difficult to define grain size. Additionally, we visually investigated the growth of the inclination angle of the visible layers, such as cloudy bands or tephra layers, from the horizon using a protractor. A detail of the method is given in Appendix A3. In this process, core rotation resulting from core breaks was not considered. In addition to our preliminary report on this point (DFICPM, 2017), this study provides detailed data.

**4. Results**

**4.1 Depth-dependent variation in measured $\Delta\varepsilon'$, corrected $\Delta\varepsilon$ and SD values**

Results on $\Delta\varepsilon'$ are presented in Figure 3. Examples of the continuous variation of $\Delta\varepsilon'$ along 0.5-m core segment are provided in Appendix C. Markers and bars indicate the average values and the standard deviations (SD) determined at 0.5-m intervals along the core sample. Dots represent raw data. Black and red symbols represent data obtained without and with inclined

measurement, respectively. Data from both cases (without and with horizontal sample inclinations within the resonator) in agreement at overlapping depths (2630–2730 m), confirming the experimental principle that the horizontal rotation of the samples in the open resonator give no influence on the measurement of the relative permittivity. We note that in the LO20% $\Delta\varepsilon'$ values have a large-scale tendency to decrease with increasing depth. The scatter of the raw data tends to be larger at greater depths.

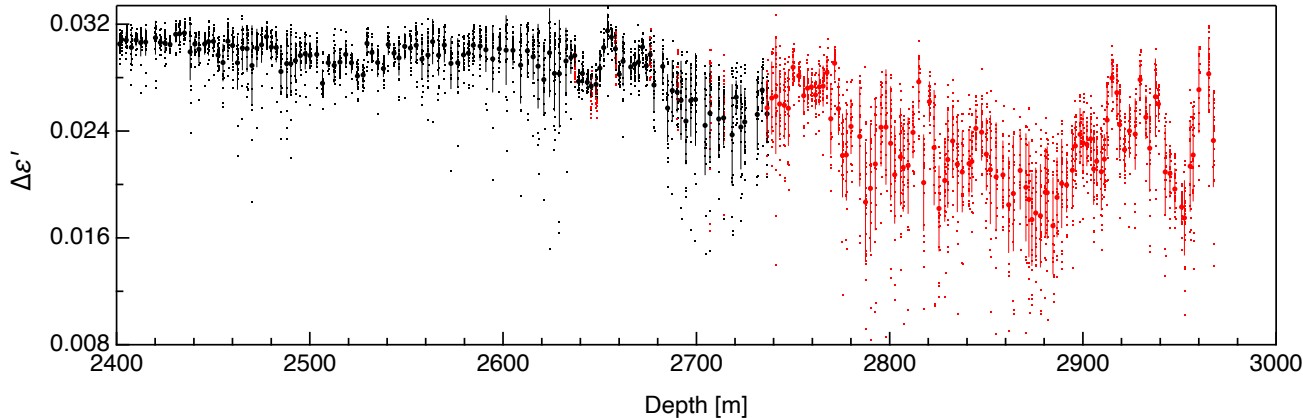

**Figure 3.** Results on $\Delta\varepsilon'$ in the LO20%. Dots represent raw data, recorded at every 0.02-m step. Markers and bars indicate the average values and SDs for each 0.5-m step. Black and red markers/dots indicate data obtained without and with rotation of the samples in the open resonator, respectively (see Figure 2).

### 4.2 The *c*- and *a*-axis orientations measured by the Laue X-ray diffraction method

Figure 4 presents examples of the *c*- and *a*-axis orientations obtained through the Laue measurements. Examples from 17 of the 42 measured depths are shown. The three panels, from left to right (views from 1 to 3), represent *c*- and *a*-axis projected on the Schmidt equal net diagram as follows. View 1: Horizontal view from the axis of the microwave beam incidence (the center of the diagram), which is the same view as in Figure A2b in Appendix A2, without information of the inclination angle or the horizontal orientation of the *c*-axes cluster. View 2: Horizontal view from the orthogonal direction (axis) of the inclination angle. This view is attained by horizontal rotation of the system using the vertical axis, by moving the center of the *c*-axes cluster to the periphery of the diagram. This operation relates to the horizontal orientation. The center of the diagram is parallel to the girdle of the *a*-axes. This view corresponds to that in Figure A2b'. View 3: View seen from the center axis of the *c*-axes cluster. In all diagrams, red dots and dark blue dots represent the orientations of *c*-axis and *a*-axis within the space, respectively. Additionally, green triangle markers indicate the orientation of the vertical in the ice sheet. The fourth column (View 4) indicates normalized density ρ of the *a*-axis along the girdle plane of the *a*-axis. θ (degrees) is angle from the figure top of the View 3.

The *c*–axis fabric generally exhibits a strong single-pole cluster. The strength of this cluster fluctuates with increasing depth. Since each crystal grain of the hexagonal crystal lattice has three equivalent *a*-axes orthogonal to the *c*-axis, the

distribution of the *a*-axes forms a girdle plane orthogonal to the *c*-axes cluster at each depth. As for the inclination angle, it is monotonically larger at greater depths down to 2967 m. Figures 4(m–q) display data from the deepest five depths, which lie within the bottom 2% of the ice sheet. At each depth, red and blue number markers indicate the distributions of the *c*-axis and *a*-axis, respectively. Because of grain growth near the bed of the ice sheet, the largest number of crystal grains in a thin section was six. Therefore, the center of the clusters could not be determined with this very limited number of grains. Except these five depths, both the inclination angle and the horizontal orientation of the *c*-axes cluster were extracted through the observation of *c*-axis fabric, considering views from 1 to 3. Both were used for the corrections from $\Delta\varepsilon'$ to $\Delta\varepsilon$. We present correction procedures in Appendix B. In addition, the median inclination of the *c*-axes cluster values was calculated using the thin-section-based *c*-axis fabric data. Both the inclination angle and median inclination values of the *c*-axes cluster are presented in Figure 5f.

The *a*-axis fabric generally exhibits a strong girdle-type cluster on a plane. As with the fluctuations of the *c*-axis cluster, the strength of this girdle cluster fluctuates with increasing depth as well. Because of the three-equivalent *a*-axes orthogonal to *c*-axis for each crystal grain, the distribution of the *a*-axes in a girdle plane generally have periodicity of 60 degrees. We find in depth dependent profiles in View 4 that normalized density $\rho$ of the *a*-axis along the girdle plane of the *a*-axis, exhibit strong inhomogeneity being dependent on $\theta$. In some cases, it indicates strong 60 degrees periodicity, which is evident in (b), (e) and (g) in Figure 4 and several more examples in the Supplementary Information A. In other cases, it indicates superposition of variations with 60 degrees periodicity: examples are (a), (c), (h), and (k) in Figure 4. We discuss these clusters of the *a*-axes in this paper.

Distribution of *c*-axes of each thin section and *a*-axes anisotropy are further shown in Figures 5c and 5d. In terms of the distribution of *c*-axes, we can find crystal grains with *c*-axis oriented around 30–60 degrees at depths below 2600 m, in particular, except for the impurity-rich layer shown by brown shading. The *a*-axes anisotropy is defined as standard deviation of the *a*-axes density $\rho$ (hereinafter, $SD_\rho$) in panels 4 in Figure 4. Here larger value corresponds to the more anisotropic distribution of *a*-axes. We find that there are relatively small and large anisotropy in the impurity-rich layers and impurity-poor layers, respectively, and that the $SD_\rho$ of the *a*-axis anisotropy exhibit large scale fluctuations with changes in glacial and interglacial periods. It is also very important to note that the $SD_\rho$ is well synchronized with the grain size (Figure 5e), which implies underlining Physics.

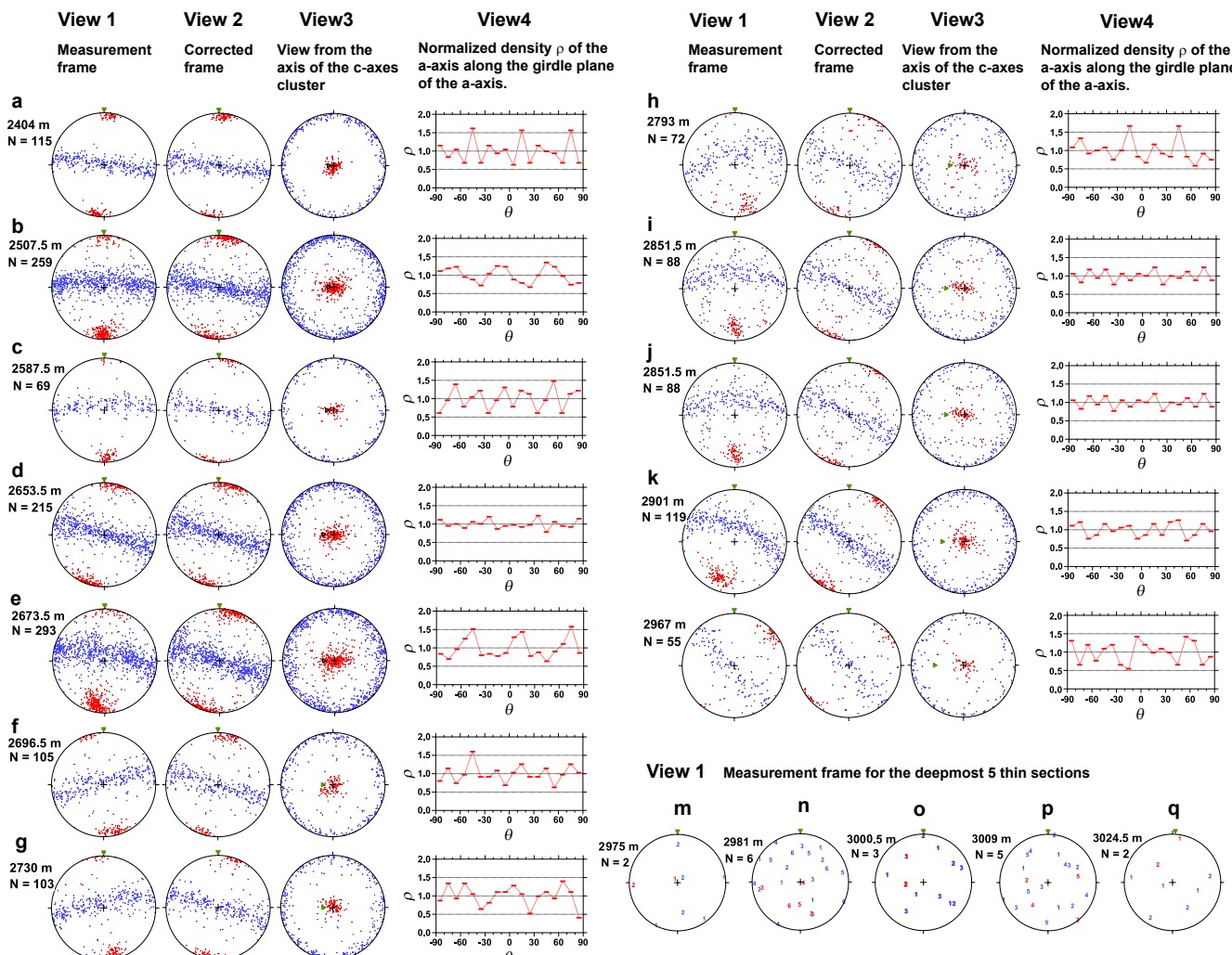

317

**Figure 4.** The *c*-axis distribution data and *a*-axis distribution data obtained from the Laue X-ray diffraction measurements. 17 depths were selected from a total of 42. At each depth of the shallower 12 depths from (a) to (l) in depth range of 2400 and 2967 m, we show three Schmidt equal area net diagrams (e.g., Langway, 1958) to express the distributions of *c*-axis and *a*-axis projected on it with red and blue colors for dots/markers, respectively. For these 12 depths, the left diagram (view 1) is the projection from the measurement frame (refer to Figure A2b). The second left diagram (view 2) is the projection from the corrected frame (see Figure A2b'). The third left diagram (view 3) is the projection from the center of the *c*-axes cluster. Green triangle in each diagram indicates the vertical orientation in the ice sheet. The vertical orientation aligns with the core axis. The fourth column indicates normalized density ρ of the *a*-axis along the girdle plane of the *a*-axis. θ (degrees) is angle from the figure top of the View 3. Diagrams from (m) to (q) represent the projection from the measurement frame for the deepest five depths, ranging from 2975 to 3024.5 m. Because of grain growth near the bed of the ice sheet, number of crystal grains in each thin section was less than 6. Instead of dots, number markers were used to indicate the orientation of each grain.

### 4.3 Depth-dependent variations of corrected Δε and SD values, and eigenvalues

We corrected from $\Delta\varepsilon'$ to $\Delta\varepsilon$ using both the inclination angle and the horizontal orientation of the $c$-axes cluster estimated
from the Laue X-ray diffraction method. Correction procedures are presented in Appendix B. Figure 5a indicates corrected $\Delta\varepsilon$
values and eigenvalues obtained from Laue X-ray diffraction method. As with the Figure 3, markers and bars indicate the
average values and the SD determined at 0.5-m intervals along the core sample. The relationship between $\Delta\varepsilon$ and eigenvalue
is expressed as $a_3^{(2)} = (2\Delta\varepsilon / \Delta\varepsilon_s + 1) / 3$, assuming a single-pole fabric without horizontal anisotropy. Here, $\Delta\varepsilon_s$ represents the
dielectric anisotropy of a single crystal. The $\Delta\varepsilon$ value (see Figures 5a) reaches approximately 0.031 at depth of 2430 and 2654
m. After reaching this maximum level, the $\Delta\varepsilon$ values fluctuate largely. The decrease in $\Delta\varepsilon$ values and larger SD are directly
linked to the scatter in each individual $\Delta\varepsilon$ measurements (represented by dots in Figure 5a). When scatter is pronounced, $\Delta\varepsilon$
value averaged over each 0.5-m segment becomes smaller and the SD larger. At depths greater than about 2900 m, the $\Delta\varepsilon$
values exhibit greater fluctuations over distances on the order of 10 m. It is noted that when the SD values are large, each ice
core displays numerous sharp negative spikes in $\Delta\varepsilon$, as shown by the dots in the panel and an example profile in Figure C2.

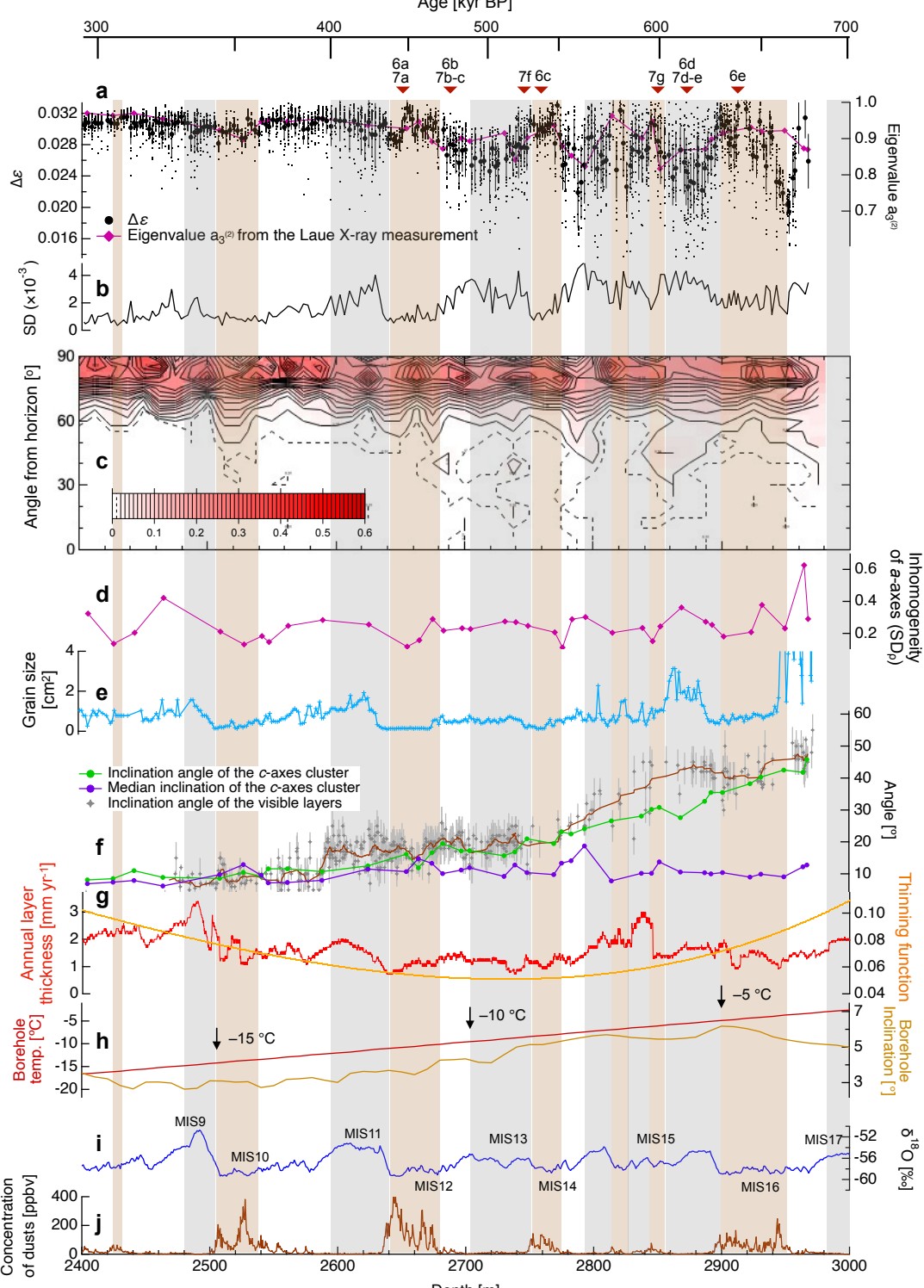


**Figure 5.** Detailed view of the comparison of Δε with various ice core data in the LO20%. (a) Δε (mean and raw data from DTM) and eigenvalues (from the Laue X-ray diffraction method). Possible uncertainty of the eigenvalues in terms of the total number of sampled grains are given in Appendix A4. (b) SD. (c) Distribution of *c*-axes as angles from a plane orthogonal to the cluster of *c*-axes. This was analysed using the data from the Laue X-ray diffraction method. The probability of the presence of *c*-axes is expressed with contour lines. (d) Inhomogeneity of *a*-axes shown as standard deviation of *a*-axes density ρ ($SD_\rho$). (e) Grain size. (f) Inclination angles of the *c*-axes cluster and the visual layers (smoothing) and median inclination. (g) Annual layer thickness and thinning function (DFICPM, 2017). (h) Borehole temperature and inclination (Motoyama et al., 2020). (i) $\delta^{18}O$ (DFICPM, 2017). (j) Concentration of dust particles (DFICPM, 2017). Gray and brown shading indicate interglacial periods and depths with a higher concentration of impurities (representing impurity-rich layers), respectively. The depths at which we observed the microstructure are indicated in the upper part.

## 4.4 Grain size and layer inclination

The results for the grain sizes and inclination angle of the visible layers are presented in Figures 5e and 5f, respectively. Grain size tends to be larger during interglacial periods and significantly smaller during glacial periods, with this difference becoming more pronounced at greater depths. We observed that Δε, SD of Δε, and grain sizes are approximately synchronized, implying the presence of common underlying mechanisms. In grain sizes we could observe only weak increasing trend versus depth. The grain sizes at depths between 2900 and 2950 m remain small despite temperatures higher than –5 °C. At depths greater than 2950 m (in the deepest approximately 2%), the grain sizes become extremely large. Grain size profile within the entire thickness is displayed in Figure 10e along with the grain size reported from the UP80% of the DF1 ice core (Azuma et al.1999, 2000). We observe that in the UP80%, the grain size increased steadily, while in the LO20%, the grain size tends to fluctuate, showing a clear distinction between very small grains of impurity-rich layers and larger grains of impurity-poor layers. There is a transition of the data aspect at depth of about 2500 m.

The inclination angle of the visual layers will not be exactly consistent with the inclination angle of the *c*-axes cluster if simple shear strain occurs in the ice sheet. The simple shear in principle contains components of compression, extension, and rigid-body rotation of the system. In this case, an axis orthogonal to the shear plane and the inclination angle of the *c*-axes cluster will deviate. Indeed, we observed that the inclination angles of the *c*-axes cluster and visual layers deviate over a wide range of depths, deeper than about 2580 m (Figure. 5f). The deviation is pronounced at depths between 2800 and 2900 m. We also note that the inclination angle of the visual layers changes in stepwise manner at 2580 and 2770 m.

## 4.5 Microstructure in the deep sections of DF ice core

### 4.5.1 Microstructural features in glacial and interglacial periods

In the present study, we provide limited but significant examples of noteworthy microstructures in Figures 6 and 7. In Figure 6, five examples include impurity-rich ice (panel a: 2648 m, ~270 ppbv dust concentration; panel c: 2759 m, ~137 ppbv), impurity-poor ice (panel b: 2685 m, ~10 ppbv; panel d: 2872 m, ~3 ppbv), and impurity-rich deep ice (panel e: 2909 m, ~80 ppbv). Here, we use the term impurity-rich to mean that ice contains either insoluble particles (like dust) and/or soluble

impurities such as dissolved ionic species. The concentration of impurity-poor ice is distinctly lower than that of impurity-rich ice; the difference is significant. We provide images of ice thin sections viewed through crossed polaroids, $c$-axis fabric data for each grain obtained using G50, and microscopy images. Black thin lines in the microscopy image indicate grain boundaries. Grain boundaries on the reverse side of a thin section are visible as thinner lines. Illustration of grain boundaries, grain boundaries on the reverse side, and subgrain boundaries are shown in Figure A3 in Appendix A5. The legend for $c$-axis fabric data is given as a circle at the bottom of Figure 6. The color of each grain indicates $c$-axis orientation; the red color means that the $c$-axis has an orientation in the vertical. As for the $c$-axis orientation, at panels (a) and (c), we observe many small grains with $c$-axis orientation distinctly offset from the vertical direction (see $c$-axis fabric image). These are sparsely distributed. The size of such small grains range in the order of a millimeter or much less. Additionally, we observe that flattened (or, elongated in 2D) shape of grans that is slanting. These features were unique in the impurity-rich depths, and absent in impurity-poor depths. At panels (b) and (d), we observe much coarser grains with $c$-axis orientation distinctly offset from the vertical direction. Compared to the cases of the impurity-rich layers, these coarser grains occupy much larger areas in the image, with diameters of a few millimeters. In the two deepest samples (panels (d) and (e)), it is evident that the crystal grain boundaries tend to be distributed as more straight lines, and there are few subgrain boundaries. Panel (e) is a sample of impurity-rich layers; crystal grain is coarser compared to (a) and (c). However, some features of impurity-rich layers observed in panels (a) and (c) are persistently present in (e). That is, flattened grains have slanting features. Additionally, grains with $c$-axis orientation distinctly offset from the vertical direction occupy only minor area within the image.

As for the link between ice fabric eigenvalues and the microstructure, $\Delta\varepsilon$ values are significantly affected by volume fraction of grains with $c$-axis orientation distinctly offset from surrounding grains. Coarser grain is more influential compared to sparsely distributed smaller grains. An example of comparison them is provided in Appendix C. Because the $\Delta\varepsilon$ values are volume-weighted averages within the microwave beam, $\Delta\varepsilon$ values decreases more when there are more grains (in size and number) with $c$-axis orientation distinctly offset from surrounding grains.

1. Polarized images   2. *c*-axis fabric images          3. Optical microscopy images

(a) 2648 m: impurity-rich ice

(b) 2685 m: impurity-poor ice (interglacial)

(c) 2759 m: impurity-rich ice

(d) 2872 m: impurity-poor ice (interglacial)

(e) 2909 m: impurity-rich ice

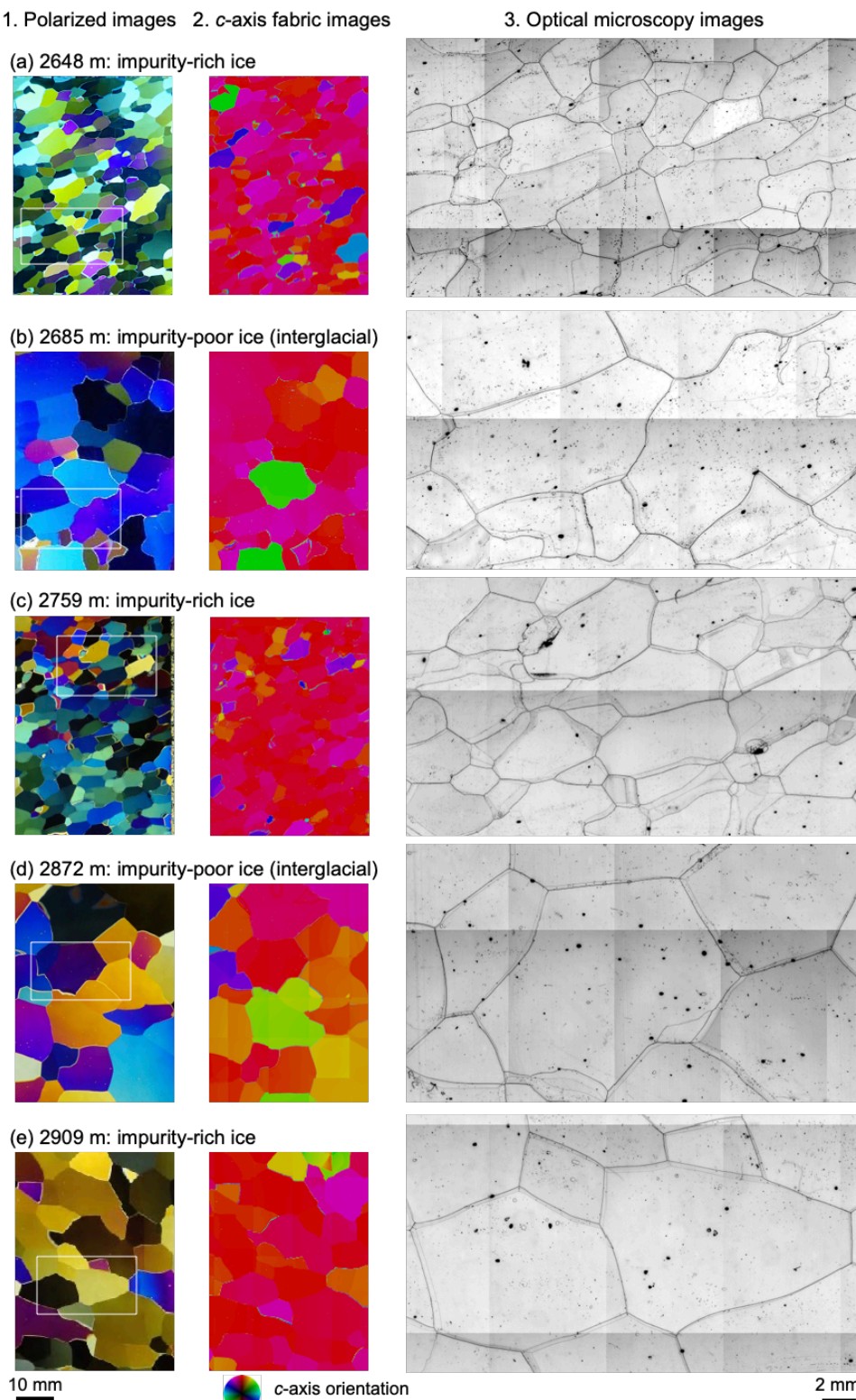

10 mm

*c*-axis orientation

2 mm

395

**Figure 6.** Microstructure images representing examples from five depths. Depths and types of ice are specified in the figure. For each depth, an image of ice thin section viewed through crossed polaroids (left), a *c*-axis fabric image (center) and optical microscopy images (right) are presented. In the optical microscopy images, black thin lines indicate grain boundaries. Grain boundaries on the reverse side of a thin section are visible as thinner lines. White frames in the left column indicate the location of microscopy images shown in the right column. For the *c*-axis fabric images in the mid column, the legend for *c*-axis orientation is given at the bottom: The color of each grain indicates the orientation of *c*-axis. Red color means that *c*-axis has the vertical orientation.

## 4.5.2 Microstructures in the impurity-rich layer: grain elongation and smaller grain size

It is remarkable that the crystal grains in impurity-rich layers tend to be flattened, and their major axes incline away from horizontal directions. These features seem absent in surrounding impurity-poor (interglacial) layers. An example of the slanting and flattened grains is shown in Figure 7a. As for the grain flattening, we analyzed the aspect ratio of the short and long axis of a fitted ellipse in 2D, using ImageJ software. The aspect ratios were distinctly different between in the impurity-rich layer and in the impurity-poor layers. In the impurity-rich layers it ranged 1.9–2.0, while in impurity-poor layers it was smaller, 1.5–1.7. Detailed data in the LO20% are given in Table A4 in Appendix A5. They are displayed in Figure 9 along with the aspect ratio reported from the UP80% of the DF1 ice core (Azuma et al.1999, 2000). We observe that in the UP80%, the aspect ratio increased steadily, while in the LO20%, the aspect ratio tended to decrease, showing a clear distinction between higher values of impurity-rich layers and lower values of impurity-poor layers. Transition from the increasing trend to the decreasing trend is at depths approximately 2500 m (Figure 9). These features of the transition are very similar to the case of the grain size, noted in section 4.4. Azuma et al. (1999) suggested that nucleation-recrystallization is not active above 2500 m depth and that deformation and rotation recrystallization affect the increasing trend of grain size and thus aspect ratio. However, below the transition at ~2500 m, situations are different in terms of nucleation-recrystallization, which we will discuss later in this paper.

Regarding the flattened features within the impurity-rich layers, Azuma et al. (1999, 2000) demonstrated growth of flattening of the grains in the UP80%, which is the basis of the aspect ratio shown in Figure 9. A similar 2D feature, known as the "brick-wall pattern" has been reported at the high-impurity ice layers of the Antarctic EDML ice core (Faria, 2009) but it has not been reported in the Antarctic EDC ice core. For the Greenland NEEM ice core, Kuiper et al. (2020) reported that fine-grained bands with flattened (or elongated in 2D) grains with aspect ratio of 2, which is as large as the maximum level in the DF core. For the EDML core, Weikusat et al. (2017) suggested that shear deformation is responsible for the flattened (or, elongated) grains. Faria et al. (2009) proposed that microscopic grain boundary sliding via microshear was a deformation mechanism for making the brick-wall pattern. According to these authors, a condition favorable for the occurrence of grain boundary sliding is likely a combination of smaller grain size, the presence of significant stress, and higher temperature. Smaller grain size is often achieved by the presence of high impurity content. These conditions are typical in the impurity-rich layer of the deeper sections in ice sheets (Faria et al. 2009). On the one hand, in the LO20%, grain elongation becomes less pronounced in deeper sections as shown in Figure 6e (sample from a depth 2909 m) and Figure 9. We suggest that the flattened features remain, but they are weakened by recovery and recrystallization due to exposure of ice to high temperature close to

melting point for time periods in the order of $10^5$ years. Strength of the flattened feature is dependent on temperature (thus, depth) and impurity concentration.

Another feature in the impurity-rich layer is persistently small grain size (Figure 5e). Such a feature is observed in MIS10, 12, 14 and 16, where the concentration of dust particles is extremely high. The small size is clearly deviated from the growth trend in the UP80% (Figure 10e). This persistently small grain size in ice core is known as "steady state grain size" (e.g., Steinbach et al., 2017). It is believed that steady state grain size is achieved when normal grain growth counteracts rotation recrystallization regardless of the initial grain size (Jacka and Li, 1994). We suggest that the steady grain sizes in the impurity-rich layers established themselves after these layers had reached deeper depths. When the impurity-rich layers were at shallower depths in the past, grain size would have been inversely correlated with dust concentration.

### 4.5.3 Migration recrystallization and grain nucleation

In dynamic recrystallization processes, rotation recrystallization has minimal effect on $c$-axis fabric changes but reduces grain size and the aspect ratio. In contrast, migration recrystallization can significantly modify $c$-axis fabric (e.g., De La Chapelle et al., 1998; Cuffey and Paterson, 2010). When a difference in stored strain energy exists between a few neighboring grains, the grain boundary migrates towards the higher-energy grain (e.g. Faria et al., 2014b). During strain induced migration recrystallization, grain boundaries sometimes become irregular, forming interlocking patterns (Duval and Castelnau, 1995; Faria et al., 2014b). Figures 7b and 7c show examples from a depth of 2685 m presumably formed by strain induced migration recrystallization. In panel (b), top-left large grain (b1) has convex grain boundary toward adjacent grain (b2). In panel (c), bottom grain (c1) has convex grain boundary toward adjacent grain (c3). The presence of numerous subgrain boundaries implies highly and heterogeneously strained region (Faria et al., 2014b; Stoll et al., 2021a). Interlocking grains are shown in Figure 7d (sample from a depth of 2872 m). Grains with various $c$-axis orientations are intricately interwoven (see the $c$-axis fabric image). These features were evident in impurity-poor layers. Contrastingly, in impurity-rich ice, grains with c-axis orientations distinctly offset from the vertical are present as smaller grains surrounded by larger grains (Figures 6a, 6c and 7a).

It is believed that grain nucleation occurs at triple junctions, at grain boundaries as two-sided grains, or similar regions characterized by high concentration of dislocation walls and subgrain boundaries (Faria et al. 2014b). We note that identification for a "just nucleated grain" is very difficult, as it requires time-series observations. Rather, when nucleation occurs, it may grow immediately after nucleation in natural ice samples; observation may be difficult. However, considering morphological features of the small grains with $c$-axis distinctly offset from surrounded by larger grains (Figures 6a, 6c and 7a), we suggest that these may be grains that nucleated at some timing of the deformational history of ice within the ice sheet.

Although there have been several studies investigating grain nucleation with artificially deformed ice (e.g., Montagnat et al., 2015; Chauve et al., 2017), there are not enough examples with natural ice samples. Faria et al. (2014b) suggested that a nucleated grain does not exhibit internal structures and bulges toward a region rich in dislocation walls and subgrain boundaries. Examples of grains with such features are shown in Figures 7e and 7f. Small grains with no internal structures (e1 and f1) are

located at grain boundaries. In case of these grains (e1 and f1), they have $c$-axis orientations close to those of the adjacent

grains (e2 and f2) (see $c$-axis fabric images in panels e and f). Nucleation of grains are further discussed in section 5 in this

paper.

As a phenomenon associated with grain boundary migration, we observed dust particle segregation (Figure 7g). The thin

lines adjacent to the grain boundary represent the reverse side of the grain boundary. Therefore, microparticles are segregated

along the planes of the crystal grains. Grain boundary migration in the deeper parts likely leads to the redistribution of soluble

impurities and dust particles.

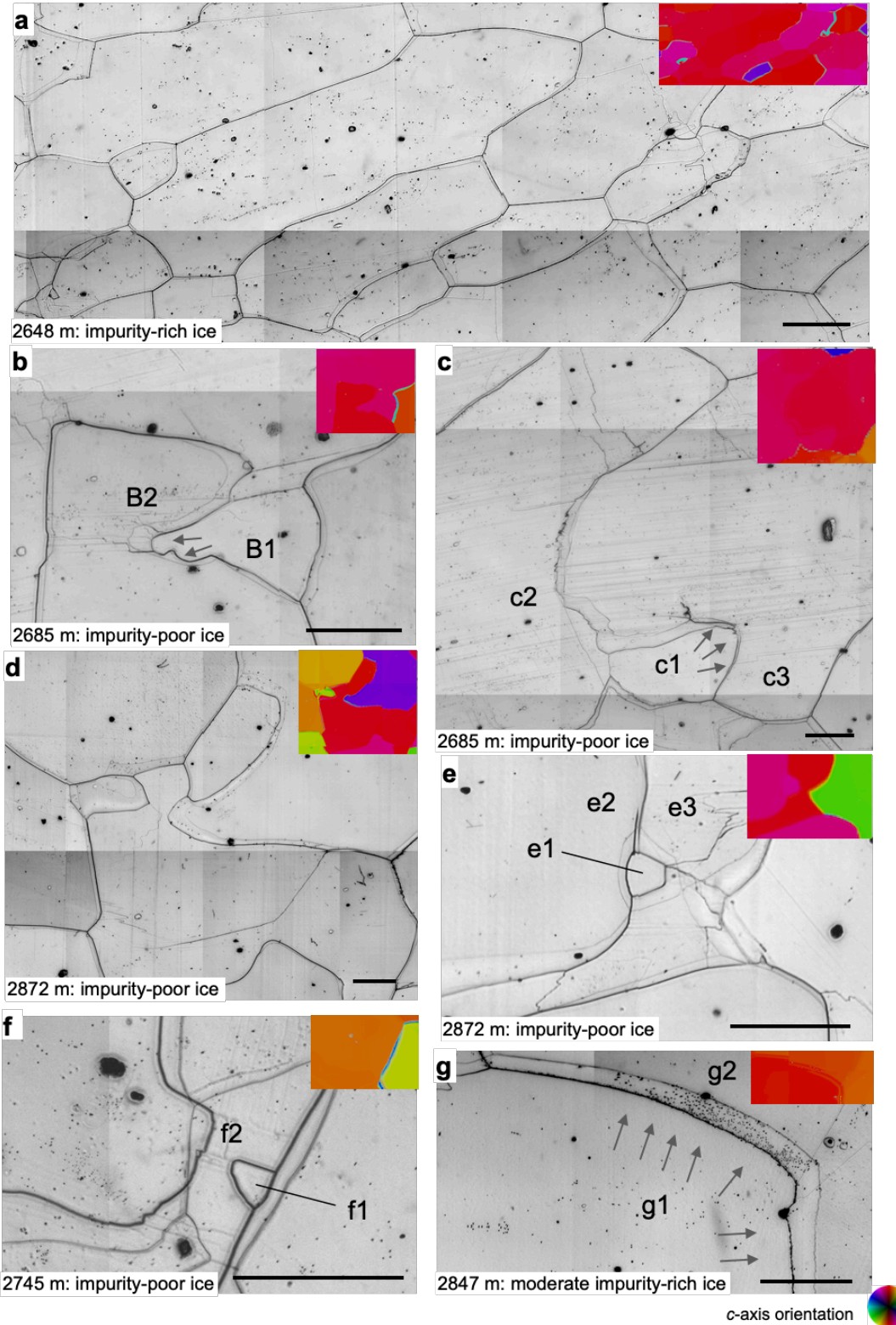

a 2648 m: impurity-rich ice

b B2 B1 2685 m: impurity-poor ice

c c2 c1 c3 2685 m: impurity-poor ice

d 2872 m: impurity-poor ice

e e2 e3 e1 2872 m: impurity-poor ice

f f2 f1 2745 m: impurity-poor ice

g g2 g1 2847 m: moderate impurity-rich ice

*c*-axis orientation

**Figure 7.** Microstructure images illustrating features of grain elongation and migration recrystallization. Images from five selected depths are presented. Depths are indicated in each panel and summarized at the top of Figure 5. All images were observed using vertical thin sections. Black solid lines indicate grain boundaries. Grain boundaries on the reverse side of a thin section are visible as thin lines. Arrows indicate convex grain boundaries. Color coding (*c*-axis fabric image) in each panel indicates the *c*-axis orientation of each grain. The legend for *c*-axis fabric images is displayed at the bottom right. Red-colored grains have a *c*-axis oriented vertically, while green or blue-colored grain are inclined horizontally. (a): Flattened (or elongated in 2D) and slanting grains observed in the impurity-rich layer. (b) and (c): Examples from a depth of 2685 m. (b): Small b1 grain has bulged (cuspidate) grain boundaries. (c): Bottom c1 grain has bulged (cuspidate) grain boundaries. (d): An example of interlocking grains (sample from a depth of 2872 m). Grains with various *c*-axis orientation are intricately interwoven. (e) and (f): Possible examples of grain nucleation (samples from depths of 2872 and 2745 m). Small grains (e1 and f1) exhibit lack of internal structures such as slip bands and subgrain boundaries, while adjacent grains display many slip bands and subgrain boundaries. The small grains are located at grain boundaries as two-sided grains. (g): Segregation of dust particles along the front of a grain boundary at 2847 m depth. Scale bars: 2mm.

## 5. Discussions

### 5.1 Variations in the layer structures in the deeper sections

#### 5.1.1 Temperature and stress conditions

The conditions of ice sheets in Antarctica, in terms of temperature and stress, are located on a boundary zone between dislocation and diffusional creep on the deformation mechanism map (e.g., Shoji and Higashi, 1978; Goodman et al., 1981, Duval et al., 1983). When ice is under temperatures close to the melting point in the LO20%, its viscosity is lower, and diffusion coefficients are higher compared to the colder ice, the rate at which recrystallization occurs also increases with temperature (e.g., Petrenko and Whitworth, 1999). Regarding the stress field, the coring site is situated on a bank very close to a subglacial trench (Figure 1c). Between the subglacial trench and the coring site, the depth difference and distance are each approximately 100 m. This geometry corresponds with the maximum inclination angle of the *c*-axes cluster (approximately 45 degrees) and the inclination angle of the visual layers (approximately 45 degrees) near the base of the ice sheet (Figure 5f). In addition, the deeper trench may act as pathway for flow of subglacial water and the deeper bed is a location for more melt (e.g., Pattyn, 2010, Fujita et al., 2012). Thus, we hypothesize that there is a simple shear strain component directed towards the subglacial trench. The rheology of polycrystalline ice with a single-pole fabric is like that of the single crystal. It easily deforms under simple shear stress. We suggest that the simple shear stress dominates the deformation of ice at the bottom part of DF.

#### 5.1.2 Relationships between the *c*-axis fabric and layer structure parameters

Figure 5 illustrates the relationships between *c*-axis fabric, layered structures, and physicochemical properties. The figure plots (a) $\Delta\varepsilon$ (mean and raw data from DTM) and eigenvalues (from the Laue X-ray diffraction method), (b) SD values, (c) the distribution of *c*-axes relative to a plane orthogonal to the *c*-axes cluster, (d) *a*-axes anisotropy within the girdle, (e) grain sizes, (f) inclination angle of the *c*-axes cluster and the visual layers, and median inclination, (g) the annual layer thickness and

thinning function, (h) borehole temperature and inclination, (i) $\delta^{18}O$, and (j) concentration of dust particles. The relationship between $c$-axis fabric and layered structures evolves with increasing depth (Figures 5a–5f). Schematic diagrams of the relationships between layered structures and orientation of $c$-axes cluster are shown in Figures 8(a–c). The inclination angles of the visual layers and the $c$-axes cluster are approximately consistent at depths shallower than about 2600 m (Figure 5f). This indicates that the system, composed of visible layers and the $c$-axes cluster, rotated together as a rigid body. At depths deeper than about 2600 m, the consistency of the inclination angles in the visual layers and the $c$-axes cluster depends on glacial/interglacial periods. In glacial periods (MIS12, 14, and 16), both angles are approximately consistent, whereas in interglacial periods (MIS11, 13 and 15), they are not; the inclination angle of the $c$-axes cluster is smaller than those of the visual layers. This implies that the crystal grains undergo a simple shear mechanism, rotating less than layer structure rotation in these depth ranges, particularly during interglacial periods. In principle, simple shear is a superposition of pure shear and the rigid-body rotation of the system. The layer inclination is simply caused by the system's rigid-body rotation. Figures 8d and 8e illustrate 2D schematic explanation for configuration of strains and rotations in the ice body above the steep bedrock slope and near the bedrock. The $c$-axes also fall within the system's rigid-body rotation. However, the cluster of the $c$-axes alone will rotate backward due to the compression components within the pure shear by dislocation creep. Thus, the inconsistency of the angles is the evidence for dominance of the simple shear at this depth range. Such features were not reported in the EDC ice core. This implies that these features are due to an environment specific to DF.

Next, we discuss layer thinning. DFICPM (2017) analyzed thinning function (the ratio of an ice layer thickness to its initial thickness at the surface) of the DF2 core from depth sequence of the climate signals and annual layer thickness (see Figure 5g). The thinning function decreased to a local minimum at about 2750 m and then increased again towards greater depths. The authors hypothesized that spatially inhomogeneous basal melting might be linked to this anomalous thinning. In Figure 5g, the broad local minimum of the thinning function is located at 2700–2800 m. This depth range agrees well with starting depths for large inconsistency between the inclination angles of the $c$-axes cluster and the inclination angles of the layers. Importantly, this depth is an approximate boundary where larger fluctuation of the $\Delta\varepsilon$ started. We speculate that the observed phenomena were linked to higher temperatures, thus the activated occurrence of recrystallization, and increased strain of simple shear from the bed. Then, we propose a scenario for development of the layer structure at the deeper part in DF based on the observed data using Figure 8f. An important point is that the DF drilling site is located just above a subglacial slope at the bank of the drainage pathway of the meltwater (Tsutaki et al., 2022). Under the dominance of the vertical normal stress near the dome, horizontal shear appears mainly on subglacial slopes rather than ridges or troughs (Tsutaki et al., 2022). Basal troughs are often influenced by basal melt or connected to deeper troughs of more basal melt. Consequently, troughs tend to serve as rapid pathways for ice flow. Thus, we suggest that the subglacial slope near the trough causes ice to flow towards the center of the trough, shearing the layered conditions. Furthermore, spatially inhomogeneous basal mass loss due to melting lead to an imbalance of force equilibrium in the vertical direction. Below the hardest ice, ice mass is lost locally at the base. As a result, layers can be stretched downwards, creating a convex shape centered around the location of the trough.

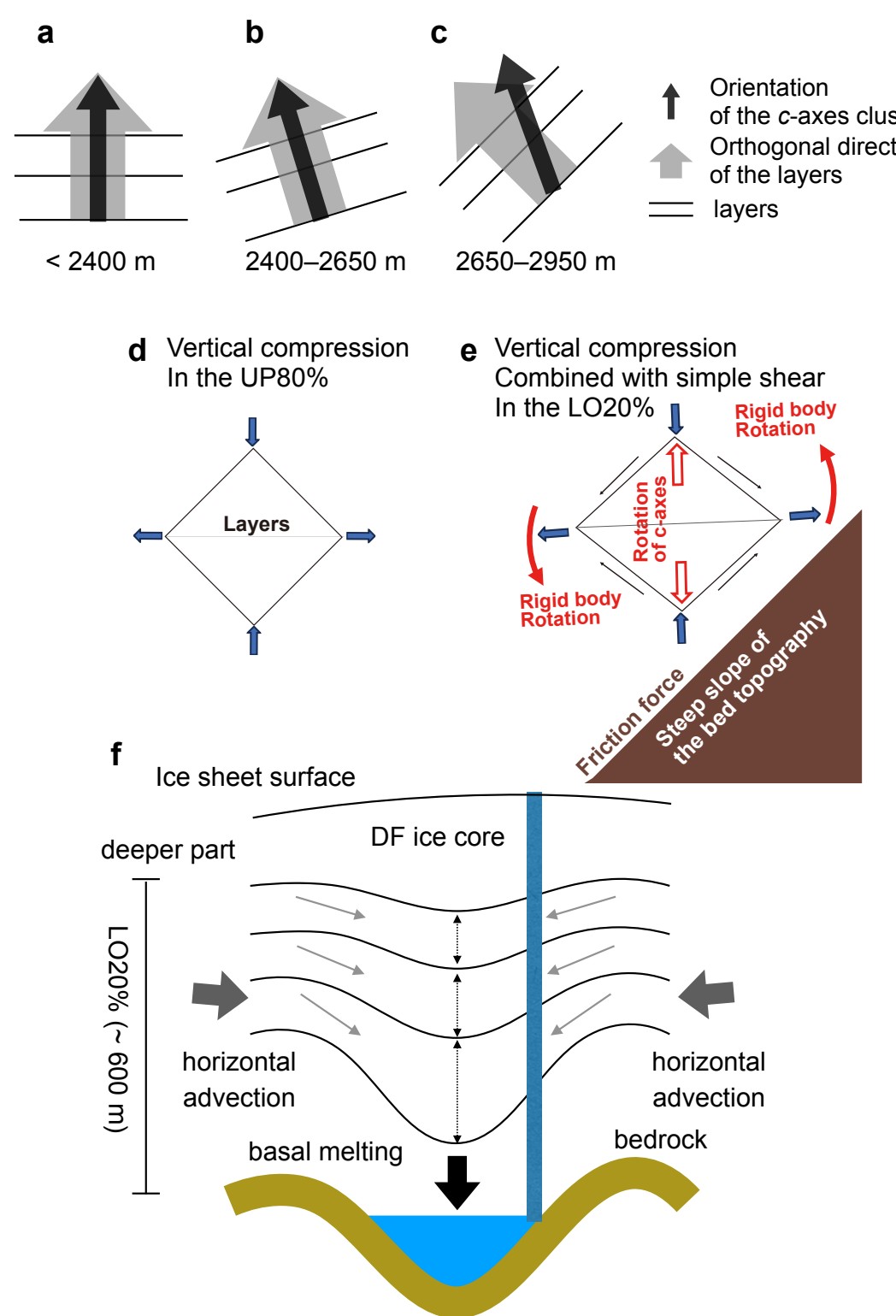

a

< 2400 m

b

2400–2650 m

c

2650–2950 m

Orientation
of the *c*-axes cluster

Orthogonal direction
of the layers

layers

d Vertical compression
In the UP80%

Layers

e Vertical compression
Combined with simple shear
In the LO20%

Rigid body
Rotation

Rotation
of c-axes

Rigid body
Rotation

Friction force

Steep slope of
the bed topography

f

Ice sheet surface

deeper part

DF ice core

LO20% (~ 600 m)

horizontal
advection

horizontal
advection

basal melting

bedrock

**Figure 8.** Schematic diagram of layered structures and 2D schematic explanation for configuration of strains and rotations in the ice body at Dome Fuji. (a–c) Diagram illustrating the relationship between layers and the orientation of the $c$-axes cluster. (a) Far above the bedrock, layers and the orientation of the $c$-axes are consistent, both oriented vertically. (b) Up to a depth of 2650 m, layers and the $c$-axes cluster rotate together as a rigid body. The directions of $c$-axis clustering and layers are consistent, yet they deviate from the vertical. (c) In addition to the rigid body rotation of the system, the cluster of $c$-axes rotates backward due to the compression component of pure shear. The orientations of the $c$-axes clustering and layers diverge. (d) and (e) Rectangular shape means a body of ice considered for deformation. Black thin arrows and blue bold arrows mean components of shear strains and normal strains. Bold red arrows indicate rigid body rotation of the system. Open red arrows indicate the axis toward which the cluster of the $c$-axes rotate. (d) A condition well above the bed, or in the UP80% in this paper. Uniaxial compression in the vertical dominates. (e) A condition under more influence from the bed topography, or in the LO20% in this paper. Because of the friction force between ice and bed, simple shear system rotates the entire system including the internal layers and the $c$-axes cluster together. However, because of the normal components of the strains (both compression in the near-vertical and extension near the horizontal plane), the all the $c$-axes thus the $c$-axes cluster rotates toward the vertical at the same time. This mechanism of the rotation in counter-direction works only for the $c$-axes and not to the layers. In this way, total amount of inclinations becomes larger for the internal layers than the cluster of the $c$-axes. (f) Schematic diagram of deep layered structure. The DF drilling site is located just above the subglacial slope at the bank (see Figure 1c), and horizontal shear along subglacial slopes appears in areas including the borehole near the trough. We suggest that the subglacial slope induces ice flow towards the center of the trough. Spatially inhomogeneous mass loss due to basal melting can lead to an imbalance in force equilibrium in the vertical direction. Layers can stretch downward, forming convex shape centered around the trough.

### 5.1.3 Variations in *c*-axis fabric and texture by the presence of insoluble impurities

Within the UP80%, Saruya et al. (2022b) investigated the controlling factors of $c$-axis fabric by comparing $\Delta\varepsilon$ with various ionic impurities and dust concentrations. They found that $\Delta\varepsilon$ is correlated positively with the concentration of $Cl^-$ ions and inversely with the amount of dust particles. In contrast to UP80%, the relationship between $\Delta\varepsilon$ and $Cl^-$ ions within the LO20% was unclear. For the depths of the LO20%, we compared $\Delta\varepsilon$ with $\delta^{18}O$, grain size, and dust particles (Figure 5). Profiles of concentration of ionic impurities ($Cl^-$, $SO_4^{2-}$ and $Ca^{2+}$) are plotted in entire depths figure (Figure 10). Referring to the $\delta^{18}O$ profile, we observe that depressions in the mean $\Delta\varepsilon$ (caused by a large scatter of data points) occur during interglacial periods at depths below 2650 m; the depressions deepen with larger scatter of the raw data at greater depths. The relative sizes of SDs are smaller in glacial periods and larger in interglacial periods, respectively (Figures 5b). The highly concentrated impurities had an apparent influence on the $\Delta\varepsilon$ values, maintaining a persistently high level of $\Delta\varepsilon$ below 2600 m (Figure 5a). Additionally, grain sizes tend to be small (Figure 5e), and the aspect ratio of the crystal grains remains consistently high in these layers, with values reaching up to 2 (Figure 9). Overall, these textural features—grain sizes, shapes, and $c$-axis fabric—are interdependent and strongly correlated with the presence of impurities.

According to a case study for NEEM ice core by Eichler et al. (2017) and the review by Stoll et al. (2021b), dust particles located not only at grain boundaries but also at grain interiors and triple junctions. To the best of our knowledge, the role of microparticles in ice deformation (dislocation creep) is not well understood. Production of dislocation is one of the possibilities.

In contrast, microparticles may act as sink of dislocation like grain boundaries. Saruya et al. (2022b) suggested two
possibilities: (i) restricted deformation due to the inhibition of dislocation by dust particles and/or (ii) the contribution of
diffusion creep that does not cause $c$-axis rotation because, with this mechanism, only molecules diffuse due to the condition
of smaller grains. Which effect ((i) or (ii)) is more dominant remains unresolved. However, in either case, they can restrict
changes in $c$-axis clustering. What we observed in ice below about 2600 m is that the degree of $c$-axis clustering in the dusty
(i.e., impurity-rich) layers is stronger than in the surrounding layers. Additionally, the consistently smaller grain sizes and
higher aspect ratio of the crystal grains in the impurity-rich layers indicate that grain boundary migration is also restricted in
such ice. This confirms that the movement of dislocations (as line defects) and grain boundaries (as planar defects) within the
crystal lattice is significantly influenced by the presence of insoluble impurities.
Comparing the relationship of $c$-axis fabric with impurities in the UP80% with the LO20%, in the UP80%, impurity-
rich/poor layers have smaller/larger $\Delta\varepsilon$ values. In the LO20%, some thickness between 2400 m and 2650 m is a kind of
transition zone; in it we find no clear correlation between $\Delta\varepsilon$ values and impurity. Below depths of about 2650 m, the trend is
seemingly reversed. The $c$-axis fabric clustering strength exhibited substantial fluctuations. There, more impurity-rich layers
maintain clustering persistently. In impurity-poor layers, relaxation of the $c$-axis fabric clustering occurred, as represented by
numerous negative spikes of $\Delta\varepsilon$. A similar relationship exists between the aspect ratio and impurities in the UP80% and LO20%
sections. In Figure 9, we observe an increasing trend in aspect ratio with depth in the UP80%. In contrast, the LO20% shows
a distinct split in the data: an aspect ratio close to 2 in impurity-rich layers and much smaller values in impurity-poor layers.
In summary, the presence of insoluble impurities in the ice sheet has a significant impact on the $c$-axis fabric, grain size,
and aspect ratio of crystal grains. In impurity-rich layers, particularly below 2600 m, grain boundary migration is restricted,
leading to consistently smaller grain sizes and higher aspect ratios. The relationship between these textural features and
impurities varies between the UP80% and LO20% sections, with a stronger correlation observed in the latter. These findings
highlight the complex interplay between impurities and the deformation mechanisms within the ice, suggesting that the
presence of insoluble particles plays a crucial role in controlling the microstructural evolution of deep ice.

**5.1.4 Cause of $c$-axis fabric fluctuations inferred from microstructures**

The $c$-axis fabric contains information on several factors: (i) deformational history, (ii) grain growth, and (iii)
recrystallization (e.g., Cuffey and Paterson, 2010; Faria et al., 2014a, b). Ultimately, all these factors are sensitive to the
deposition of insoluble impurities. Insoluble impurities restrict grain boundary migration through Zener pinning mechanisms
and also impede dislocation movement (Alley and Woods, 1996; Durand et al., 2006). These characteristics result in slow
growth of $c$-axis clustering in impurity-rich layers within the UP80% and slow relaxation of $c$-axis clustering in the LO20%.
This leads to an apparent reversed correlation between $\Delta\varepsilon$ values and impurity levels in both the UP80% and the LO20%. In
terms of the distribution of $c$-axes (Figure 5c), we can find crystal grains with $c$-axis oriented around 30–60 degrees at depths
below 2600 m. It is believed that new grains tend to form with an orientation favorable for basal glide (e.g., Alley, 1992;
Humphreys and Hatherly, 2004, Cuffey and Paterson, 2010), that is, approximately 45 degrees from compressional axis.
However, it is also noteworthy that, in Figure 5c, distribution of $c$-axis density approximately from 30 degrees from
compressional axis (60 degrees from horizon) is always denser than 45 degrees or 60 degrees. It is possible that it has some
underlying mechanisms in terms of nucleation recrystallization relative to the existing $c$-axis cluster. Additionally, crystal
grains with $c$-axis oriented around 30–60 degrees tend to appear more in the impurity-poor zones where grain size is larger
with low level of $\Delta\varepsilon$. We suggest that these grains with a $c$-axis orientation of around 30–60 degrees represent nucleated grains
in the deeper part, and that these grains might grow and eliminate old grains by migration recrystallization. At the same time,
ice crystals will recover the $c$-axis orientations available for the continuation of dislocation-creep-based deformation. The
hypothesis that these grains are the result of nucleation is consistent with the observation of the corresponding grains, as
described in Section 4.5.
We propose that grain nucleation and migration recrystallization lead to significant changes in crystal orientation fabric; a
decreasing in $\Delta\varepsilon$ values and less clustered $c$-axis fabric in impurity-poor layers can be explained only by growth of grains with
different $c$-axis orientations. Conversely, such grains are much smaller and limited in volume in the impurity-rich layer. Grain
coarsening by migration recrystallization does not appear to occur since insoluble impurities restrict the grain boundary
migration (e.g., Durand et al., 2006; Stoll et al., 2021b) thus growth of the nucleated grains as well. Therefore, the $c$-axis fabric
change (decreasing of $\Delta\varepsilon$ values) caused by nucleation and migration recrystallization would appear strongly only in the
impurity-poor layers. Thus, the dynamic recrystallization process would have greatly contributed to the characteristic behavior
of $c$-axis fabric development Even when grain nucleation occurs in ice both in impurity-rich ice and in impurity-poor ice, its
contribution to $c$-axis fabric changes depend on the surrounding conditions: it may not be immediately apparent in the volume-
weighted average $\Delta\varepsilon$ values in the DTM. However, when such nucleated grains grow, which is the case mostly in impurity-
poor ice, subsequent strain induced migration recrystallization significantly affects the $\Delta\varepsilon$ values because of the growth of
grains having different $c$-axis orientations.
**5.1.5 Variations in $a$-axis fabric**
Grain nucleation will also lead to significant changes in $a$-axis fabric. However, due to the lack of data on $a$-axes
distribution in the UP80% section, we cannot determine the presence or absence of anisotropic $a$-axes organizations in this
upper section. Our observations are as follows: (i) In many depths within the LO20%, the girdle plane of the $a$-axis exhibits
strong inhomogeneity depending on $\theta$ (refer to the panels in View 4 of Figure 4 and Supplementary Information A). (ii) When
the inhomogeneity of the $a$-axis within the girdle is expressed as $SD_\rho$, the $SD_\rho$ tends to be larger in impurity-poor layers and
smaller in impurity-rich layers (Figure 5). (iii) The $SD_\rho$ is well correlated with grain size, aspect ratio, and $\Delta\varepsilon$ values (Figures
5 and 9).
These observations raise several questions: (i) At what depths and how did this $a$-axis organization begin to form and
develop? (ii) What is the physical mechanism behind this phenomenon—is it deformation, recrystallization, or a combination
of both? (iii) What is the spatial scale of the *a*-axis organization? Is it limited to the scale of thin sections, or does it extend
further?

Many of these questions cannot be conclusively answered with the current data alone. However, we can speculate based

on our significant findings. In dislocation creep, if the slip plane is primarily the *c*-plane (the easy-glide plane of hexagonal
ice), it is unlikely that *a*-axis organization occurs geometrically through *c*-plane slip. Since these processes do not seem
interdependent, we exclude this possibility. The organization of the *a*-axis structure among these crystal grains can only occur
due to interactions at the grain boundaries between adjacent crystal grains. Therefore, we speculate that dynamic
recrystallization processes, particularly migration recrystallization associated with nucleation, play a critical role. The
observations listed as (i) to (iii) at the beginning of this section support this speculation.

Unlike migration recrystallization, which can significantly modify *c*-axis fabric (e.g., De La Chapelle et al., 1998; Cuffey

and Paterson, 2010), rotation recrystallization has a minimal effect on *c*-axis fabric changes but does reduce grain size and the
aspect ratio. Similarly, it seems reasonable to think that rotation recrystallization has a minimal effect on *a*-axis fabric changes
as well. Thus, we exclude this as a possible cause of the *a*-axis organization. We speculate that the new grains, with *c*-axis
orientations distinctly offset from the single pole cluster, also have new *a*-axis orientations. When new crystal grains form
through recrystallization, they arrange water molecules to form the most energetically favorable orientation, minimizing energy
between the surfaces of the grains and forming *a*-axis alignment between adjacent grains. Here, we cite two papers by Matsuda
et al. (1976) and Matsuda and Wakahama (1978) on the possibility of crystal twinning in ice sheets. Crystal twinning occurs
when two or more adjacent crystals of the same mineral share some crystal lattice points symmetrically, forming an intergrowth
of tightly bonded crystals. The shared surface is known as the twin plane or composition surface. Matsuda et al. (1976)
investigated ice core samples from the Antarctic ice sheet near Cape Folger, revealing distinct crystallographic changes at
specific depths. The ice layers exhibit a "diamond" pattern in *c*-axis orientations, indicating twinning relationships between
crystal groups. The crystal structure of hexagonal ice has seven directions of oxygen-oxygen bonds, which are connected by
hydrogen bonds, including along the *c*-axis. The angles between these bonds are equal to the tetrahedral angle of 109.5°. These
authors determined the orientations of all seven oxygen-oxygen bond directions from the orientations of the *c*-axis and *a*-axis
measured in polycrystals with a four-maxima "diamond" pattern *c*-axis fabric. As a result, it was found that 2 to 3 of the seven
bonds in each crystal group were aligned with the bond directions of neighboring crystal groups. The alignment of oxygen-
oxygen bond directions between neighboring crystal groups strongly suggests that the neighboring crystals might be in a
twinning relationship. Variations in grain size, texture, and crystallographic orientation are linked to deformation under high
shear stress, with less deformation and twinning observed in certain layers, leading to characteristic *c*-axis patterns. Matsuda
and Wakahama (1978) further investigated several types of polycrystalline ice of different origins. They suggested that a great
majority of the adjoining crystals might be in a twinning relation. They suggested a strong possibility that the structural relation
between crystals plays a major role in the appearance and growth of nuclei which have specific orientations during
recrystallization.

These findings offer insights into the structural evolution of deep ice in response to stress and impurities; this crystal twinning hypothesis by these authors may apply to the current case of DF. Although we did not observe a four-maxima $c$-axis pattern at DF, crystal twinning likely occurs during nucleation and subsequent crystal growth, leading to share $a$-axis orientations between adjacent grains, even before the four-maxima pattern is concretely shaped.

Based on these speculations, we propose some answers to the questions posed earlier: (i) This $a$-axis organization likely began to appear and grow during periods of enhanced nucleation and migration recrystallization, particularly under higher temperatures in the LO20%. In the UP80%, this phenomenon is likely very weak. (ii) The most fundamental cause of $a$-axis rearrangement is nucleation and migration recrystallization, which relieves localized stresses. It is possible that crystal twinning, where some crystal lattice points are shared symmetrically, forms the $a$-axis organization to create the most stable thermodynamic conditions. (iii) The spatial extent of $a$-axis rearrangement is thought to be relatively limited, likely confined to a small region, such as the size of a single thin section or even smaller. This is because it is difficult to consider a far-reaching effect where distant crystal grains align with each other in terms of the $a$-axis. While the $a$-axes tend to align among adjacent and nearby crystal grains, it is natural to consider that the orientations formed in coordination become entirely different as the distance increases.

It is also noteworthy that a few deformation experiments on polycrystalline ice conducted in the laboratory have reported $a$-axis organization (Journaux et al., 2019; Qi et al., 2019; Wang et al., 2024). These experiments were conducted at temperatures near the melting point, with strain rates much higher (by about $10^5$ to $10^7$ times) than those in plateau region of the Antarctic ice sheet. A preferred $a$-axis maximum was detected in all these shear experiments. Additionally, natural examples from sheared ice under temperatures close to the melting point (Monz et al., 2021; Thomas et al., 2021) also exhibit a preferred $a$-axis maximum. These observations can be explained by the same speculation that the $a$-axis organizations were formed due to grain nucleation and migration recrystallization induced by shear strain under temperatures close to the melting point.

**5.2 Development of the overall layered structures**

**5.2.1 Depth-dependent variations of $\Delta\varepsilon$ in the entire core**

Figure 9 shows $\Delta\varepsilon$, its SD, and the aspect ratio across the entire core thickness. In the UP80%, both $\Delta\varepsilon$ and the aspect ratio increase with depth, reaching their maximum levels at the bottom. Below this, between approximately 2400 and 2650 m, the trend changes, with both $\Delta\varepsilon$ and the aspect ratio showing larger fluctuations and sharp drops. The decrease in $\Delta\varepsilon$ values and the increase in SD are directly linked to the scatter in individual $\Delta\varepsilon$ measurements (represented by dots). Pronounced scatter results in smaller averaged $\Delta\varepsilon$ values over each 0.5-m segment and larger SD, with a corresponding decrease in the aspect ratio. At depths greater than about 2800 m, $\Delta\varepsilon$ values exhibit significant fluctuations over distances of about 10 m, and the aspect ratio remains at its smallest level.

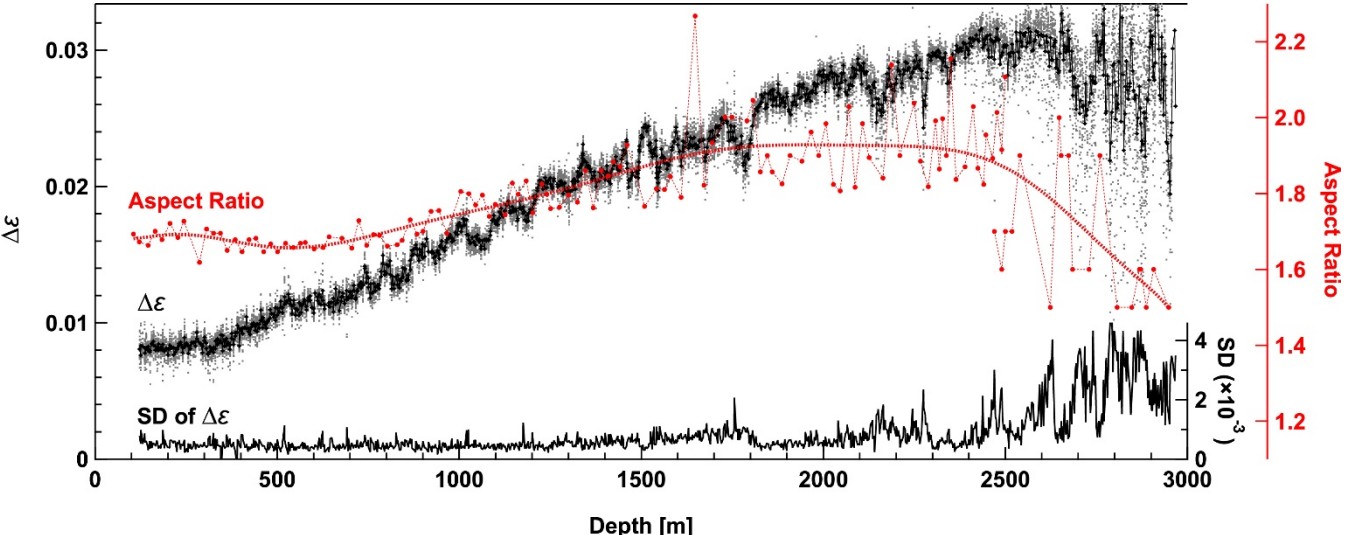

**Figure 9.** Variation of Δε, its SD, and the aspect ratio across the entire depth of the core is presented. Δε is shown as raw data (dots) and as mean values for each 0.5 m interval (markers). The dots represent Δε values at each 0.02 m step, and the SD corresponds to Δε values over each 0.5 m segment. Data from the UP80% are sourced from Saruya et al. (2022b), while aspect ratio data is indicated by red markers and lines. Data for the LO20% are from the current study of the DF2 core, and data for the UP80% are from the DF1 core (Azuma et al., 1999; 2000).

### 5.2.2 Comparison of *c*-axis fabric at deeper part of three summit ice cores

We extend our comparison of *c*-axis fabric data from DF with that of EDC. It is notable that these two sites are similar in terms of glaciological conditions, including surface temperature, annual mean surface mass balance and ice thickness (see Table 2). Figure 10a indicates the relationship between Δε (averages and raw data as with Figures 5a) value and normalized eigenvalue $a_3^{(2)}$. General trends in cluster strength and grain size are very similar in both ice cores across all age scales. Furthermore, Durand et al. (2009) pointed out depth-dependent developments in *c*-axis clustering in deep sections are similar in the GRIP ice core (Thorsteinsson et al., 1997) and EDC core in Antarctica. Indeed, at the three summit sites, the depth zones where the maximum clustering appears are similar (see Table 2). The similarity among these three sites across both hemispheres implies that certain physical mechanisms are driving this similar development of *c*-axis clustering. We hypothesize that the temperature environment within the ice sheet, through mechanisms such as dislocation creep and recrystallization, might have resulted in the commonality of *c*-axis fabric at three points in the ice sheets. In this hypothesis, at the bottom of the UP80%, the thickness of ice is approximately 10% of the original ice equivalent thickness at the time of deposition (see Figure 5g). At this depth, the eigenvalue $a_3^{(2)}$ reaches about 0.93. In the absence (or faint presence) of shear stress, a strongly clustered texture will be difficult, either to compress or shear, thereby necessitating dynamic recrystallization as an accommodation process. This state of saturation of the *c*-axis cluster, along with the common temperature range may be more effective as a condition to trigger nucleation and recrystallization. The deepest 10–20% of the polar ice sheets are

typically characterized by their ability to easily deform under horizontal shear, due to high temperatures and well-clustered $c$-axis fabric. In these depths, dynamic recrystallization plays a critical role, particularly in impurity-poor layers, to recover the potential of $c$-axis orientations available for the continuation of dislocation-creep-based deformation.

In the present study including Saruya et al. (2022b), we did not identify large anomalous step of $c$-axes clustering with increasing depth (Figures 9, 10 and Appendix D). Rather, $c$-axis fabric exhibits fluctuations that appear already at a few hundred-meter depth at Termination I. Thus, there is no indication attributable to presence of simple shear. This argument differs with the existing scientific claim given by Durand et al. (2007). They suggested the impact of shear was important to explain anomalous strengthening the $c$-axes cluster especially at the Termination II. In contrast to this earlier attribution to shear, Saruya et al. (2022b) attribute small depressions of $\Delta\varepsilon$ values at MIS5e to the very low concentration of resolved Cl$^-$ ions, which can substitute for the location of $H_2O$ in the crystal lattice, indicating that ice is harder because of it. This explanation applies to the EDC case because, at EDC, most Cl$^-$ ions are lost to the atmosphere from snow compared to DF at least during the Holocene (i.e., the interglacial period), while at Dome Fuji, they are preserved as NaCl and in solid solution (Oyabu et al., 2020). These conditions suggest that ice becomes harder, preventing the $c$-axis cluster from strengthening at MIS5e as well. In other words, this gives an alternative explanation for the significant depression of the $c$-axis cluster at MIS5e and a notable jump in the $c$-axis cluster at EDC. Additionally, further complexity arises because, at EDC, the surface mass balance (SMB) contrast between glacial and interglacial periods is approximately 20% larger than that of DF (Fujita et al., 2015; Parrenin et al., 2016), which will dilute Cl$^-$ ions more at EDC and also complicate ice flow models.

Table 2: Environment of the DF, EDC, and GRIP sites from dome regions in central plateau area of the ice sheets

| Site | Elevation (m) | Ice thickness (m) | Annual mean SMB (kg m$^{-2}$ y$^{-1}$) | Annual mean surface air temperature (°C) | Depth of temperature at –10 °C (m) | Depth of maximum $c$-axis cluster and corresponding temperature range (°C) | References |
|---|---|---|---|---|---|---|---|
| DF | 3810 | 3028 (±15) | 24–28 | –54.4 | ca. 2700 | 2400–2650 –18 to –12 | This study, DFICPM, 2017, Oyabu et al., 2023, Yamanouchi et al., 2003 |
| EDC | 3233 | 3273 (±5) | 25 | –54.5 | ca. 2800 | 2400–2650 –20 to –13 | EPICA Community Members, 2004, Durand et al., 2009, Buizert et al., 2021 |
| GRIP | 3216 | 3085 | 230 | –30.0 | ca. 2960 | 2600 –20 | Dahl-Jensen, et al., 1993, 1998, Thorsteinsson et al., 1997 |

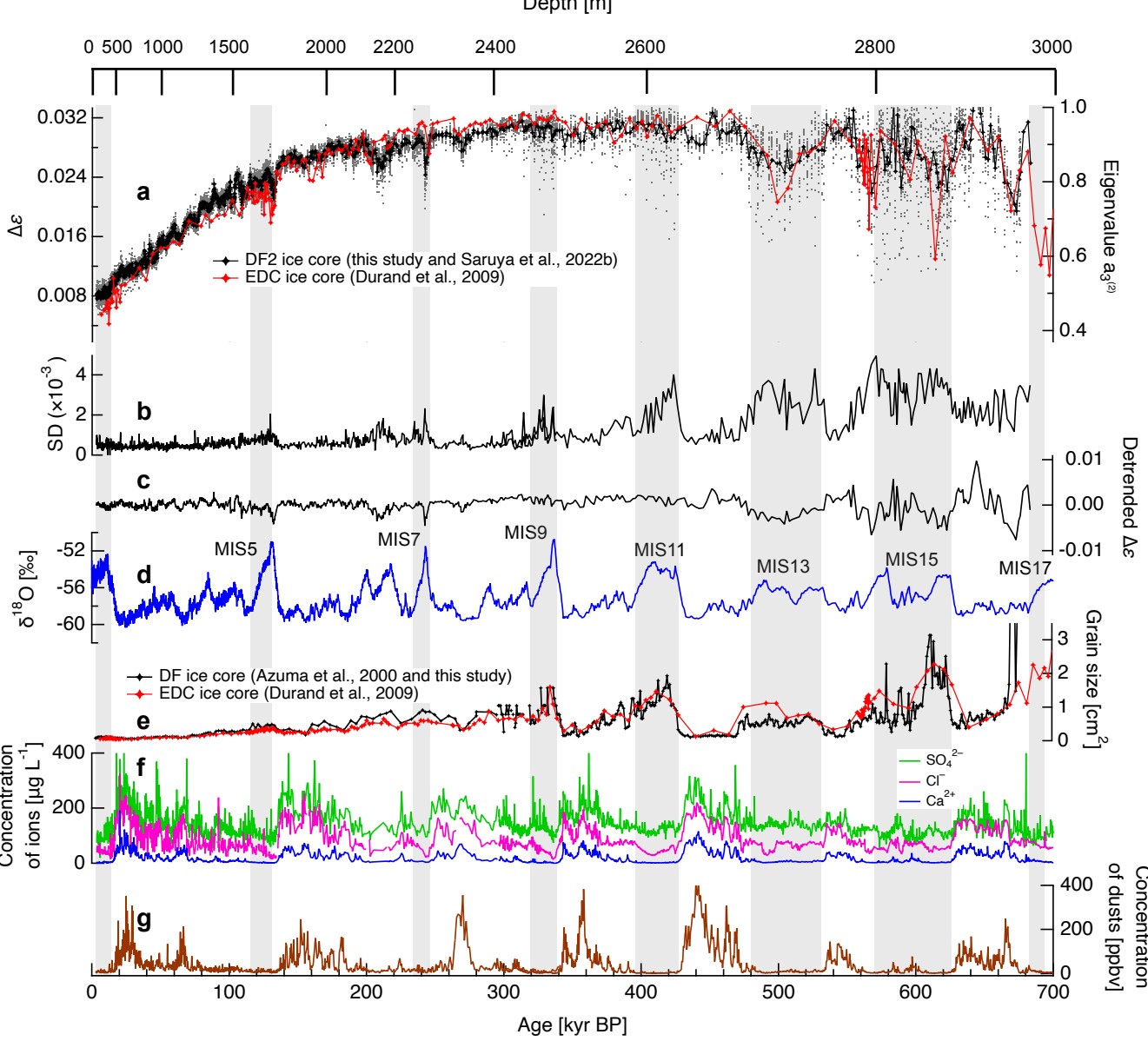

737

**Figure 10.** The *c*-axis fabric data and grain size data from the DF core and the EDC core are compared. Here, we applied the DF2021 age scale (Oyabu et al., 2022) for ages younger than 216 thousand years BP and the AICC2012 age scale (Bazin et al., 2013) for older ages. Light-gray shading indicates the interglacial periods. (a) $\Delta\varepsilon$ presented as raw data (dots) and as mean values for each 0.5-m segment (black markers for DF) or for each thin section (red marker for EDC). The right axis provides a scale for normalized eigenvalues. (b) SD for $\Delta\varepsilon$ in the DF core. (c) Detrended $\Delta\varepsilon$ values in the DF core. (d) $\delta^{18}O$ in the DF ice core. (e) Grain size in DF cores (Azuma et al., 2000 and this study) and EDC core (Durand et al., 2009). (f) Concentrations of $Cl^-$, $SO_4^{2-}$ and $Ca^{2+}$ ions (Goto-Azuma et al., 2019). (g) Concentration of dust particles (DFICPM, 2017).

### 5.3 An overview of the layered structure of the ice sheet

Based on the above discussions, we propose an overview of the layered structure of ice sheets. Polar ice sheets are massive bodies of ice, comprising layers with a wide variety of rheological characteristics, which are dependent on depositional features, that is, historical depositions of aerosols from the atmosphere. The thickness of layers ranges widely, from annual layers (millimeters) to those spanning glacial and interglacial periods. The initial ice fabric forms during firn processes (e.g., Calonne et al., 2017; Fujita et al., 2009, 2016; Montagnat et al., 2020). Another fundamental factor, not covered in the current paper, is the content of ions that either enhance ($Cl^-$ and $F^-$) or impede ($NH_4^+$) deformation (Nakamura and Jones, 1970; Jones and Glen, 1969; Jones, 1967), as well as the presence of insoluble particles like salt inclusions, in addition to the dust particles discussed in this study. In ice physics, $F^-$ and $NH_4^+$ are well-known as major factors that modify the viscosity of ice crystals (as reviewed in textbooks by Petrenko and Whitworth (1999) and Fletcher (1970)). It must be emphasized that these ions can substitute for the location of $H_2O$ in the crystal lattice, thereby modulating the density and behavior of dislocations. These effects from ions are present in the ice crystal lattice from the firn process all the way to the bottom of the ice sheet. For example, Fujita et al. (2014) and Fujita et al. (2016) suggested that the rheology of polar firn is dependent on these ions. Additionally, it must be emphasized that salt inclusions have larger volume fractions than dust particles (Ohno et al., 2005).

Ice with these diverse initial properties, which will persist in the ice, experiences two distinct conditions in the upper 80% and the lower 20% of the ice sheet, as listed in Table 3. Four types of conditions affect the development of crystal orientation fabric, microstructures, and deformational regimes. These are: (i) temperature conditions, (ii) primary strain configurations, (iii) effects of insoluble particles (such as dust, metals, and salts) on texture, and (iv) processes of dynamic recrystallization, including the formation of new grains and migration recrystallization. Consequently, each individual layer has its own trajectory of deformational history over time periods spanning up to $10^6$ years.

This overview has significant implications for practical applications, such as ice sheet modelling, ice core sciences and radioglaciology. Practical application for the ice sheet modelling is beyond the scope of this paper. However, these layered structures correlate directly with the vertical thinning of each layer, making them useful for improving ice core dating models by providing constraints on strain values. Additionally, at locations moving away from the dome, the increase in gravity-driven shear movement of the ice sheet causes inhomogeneous deformation between layers or layer zones, eventually resulting in the formation of folds, faults, and mixing of ice across various layer thickness scales. For ice core sciences, which aim to investigate continuous records of very ancient ice on the million-year scale, choosing a drilling site at the dome is crucial. Drilling at locations away from the dome area carries serious risks of layer disturbances near the bottom. Another implication of this discussion concerns the state of radio echoes from the very deep parts, specifically in the lower 20% of the ice sheet. In glaciology, the presence or absence of echo-free zones is a topic of debate (e.g., Fujita et al., 1999; Drews et al., 2009). On one hand, layers of ice *c*-axis fabric can cause the reflection of detectable radio echoes (e.g., Fujita et al., 2000). On the other hand, phenomena in the lower 20% will change the nature of ice for radio wave scattering. Concretely, reflections are more

likely from large-scale layers corresponding to glacial/interglacial changes rather than shorter time-scale events. Additionally,
layer disturbances accompanied with dynamic recrystallization may be observed as echo-free zones.
**Table 3**. Sequence of physical conditions of ice related to formation of crystal orientation fabric, texture and rheology within
the ice sheet near dome summit in East Antarctica.

| Depth range [a] | Temperature condition (°C) [a] | The primary strain configurations | Insoluble particles (such as dust and salt) slow down… | Influence of dynamic recrystallization on variation of crystal orientation fabric and texture |
|---|---|---|---|---|
| Depths of firn processes (0 – 100 m) | –55 [a] | vertical compression | (unknown) | small |
| UP80% [b] (100 – 2400m) | –55 ~ –16 [a] | vertical compression [c] | the formation of the $c$-axis clustering | small |
| LO20% [b] (2400 – 3035 m) | –16 ~ melting point [a] | vertical compression combined with simple shear | the relaxation of the $c$-axis clustering, and the decrease of the aspect ratio of crystal grains | formation of $c$-axis with orientations distinctly offset from the original $c$-axis cluster formation of preferred $a$-axis fabric, possibly due to crystal twinning the decreasing trend of the aspect ratio of crystal grains |

a) Numbers are all approximate ones.
b) There is no sharp boundary between the UP80% and the LO20%. Rather, the transition occurs gradually and
progressively.
c) Vertical compression will be combined with simple shear progressively from deeper side if we move away from dome.

**6. Conclusions**
*Use of the innovative methods for analysis of crystalline textures, and their outcome*
For enhanced understanding of layer structures and deformation regimes in polar ice sheets, we investigated the DF ice
core using innovative analytical methods. Using the Laue X-ray diffraction method, we clarified detailed information about
both $c$- and $a$-axes for each crystal grain. Microstructural observations provided signals of migration recrystallization and
potentially nucleated grains. With the DTM, we provided $c$-axis fabric data with unprecedented sampling frequency, spatial
resolution, continuity, and statistical significance. Furthermore, by combining data from these two methods, we clarified the
layering of crystal orientation fabric in the LO20%. Coupled with previously reported $c$-axis fabric data in the UP80%, we
obtained comprehensive and high-resolution $c$-axis fabric profiles for the core. The complex sequence includes changes in
depth range from the UP80% to the LO20%, and temperature variations from about –55℃ to the pressure melting point. The
primary strain configurations evolve from vertical compression to a combination of vertical compression and simple shear.
Insoluble particles like dust influence the process by shifting from promoting the slower formation of $c$-axis clustering to

inhibiting its relaxation. Moreover, the activity of dynamic recrystallization increases from less active to more active states. Including these, our major conclusions for each specific point are further listed as follows:

*Development of crystal orientation fabric, microstructure, and layering*

(i) **Presence of transitions:** Both the clustering strength of $c$-axes and the aspect ratio of grains reached a maximum level at the bottom of the UP80% and fluctuated in the LO 20%.

(ii) **The $c$-axis clustering fluctuation over short distances:** In the LO20%, the SD of the clustering strength within many 0.5-m ice segments is much larger than in the UP80%, indicating that the $c$-axis clustering fluctuation over short distances is enhanced in deeper sections.

(iii) **Growth of the $c$-axis clustering, aspect ratio, and grain size:** In the LO20%, more impurity-rich layers maintain stronger $c$-axis clustering, larger aspect ratio, and smaller grain size. In impurity-poor layers, relaxation of the $c$-axis clustering, decreases in the aspect ratio, and growth of crystal size occur due to nucleation and migration recrystallization, altering both the $c$-axis and $a$-axis fabric of impurity-poor layers.

(iv) **The preferred $a$-axis fabric:** In the LO20%, $a$-axis fabric exhibited preferred orientation within the plane of the $a$-axis girdle, in a spatial scale of the thin sections used for the Laue measurements. Organization of the preferred $a$-axis fabric is enhanced in impurity-poor ice. Additionally, the enhancement of the preferred $a$-axis fabric is well correlated with grain growth.

(v) **Roles of the nucleation, the migration recrystallization, and the crystal twinning:** Signals of migration recrystallization, such as bulged grain boundaries and interlocking grains, and potentially nucleated grains were found in impurity-poor layers. Crystal grains in impurity-rich layers showed a flattened (elongated in 2D) shape. These contrasts in microstructures are unique to deeper sections. We argue that the nucleation and the migration recrystallization is the exact physical mechanism that caused the $c$-axis relaxation and the $a$-axis organization. Possibly, crystal twinning, which share some crystal lattice points symmetrically with the neighboring crystals, is forming the $a$-axis organization.

(vi) **Presence of nucleation in the impurity-rich layers:** In impurity-rich layers, features indicating nucleation were observed. However, only weak signals of migration recrystallization were observed. The contrasting microstructures between impurity-rich and impurity-poor layers are closely linked to the differing variations in cluster strength in each layer.

(vii) **Growth of the rigid body rotation:** The layer inclination angle shows stepwise changes at 2580 and 2770 m, being 10° at depths less than 2580 m, 20° at 2770 m, and reaching 45° at 3000 m. Similarly, the orientation of the $c$-axes cluster deviates from the vertical in deeper parts. Until a depth of 2700 m, the inclination angles of the $c$-axes cluster and layers are approximately consistent but deviate from the vertical. The system rotates due to simple shear strain as a rigid body, while the $c$-axes cluster alone rotates backward, resulting from the compression components of the simple shear.

832

*Common and unique features of sites: implications for wider areas in polar ice sheets*

(i) **Atmospheric aerosols primarily determine ice fabric fluctuations:** The fluctuations of the cluster strength versus ice age are common at DF and EDC, suggesting that depositional features from atmospheric aerosols primarily determine these fluctuations. This suggests that *c*-axis fabric layering is essentially common across a wide area of polar ice sheets, if the deposition of chemical ions and dust is similar among sites.

(ii) **The similarity among three dome summit sites:** At the three summit sites in Antarctica (DF and EDC) and Greenland (GRIP), the depth zones where the maximum clustering appears are similar. The similarity among these three sites across both hemispheres may be related to commonality of temperature condition along with the total amount of strain in the LO20%.

(iii) **Unusual thickening of annual layers near the base of the DF ice explained:** Unusual thickening of annual layers near the base of the DF ice core can be explained by the rigid body rotation of the system in the meridional direction near the bed of the ice sheet.

(iv) **An alternative explanation for the depression of the *c*-axis cluster at MIS5e in EDC ice:** The significant reduction of the *c*-axis cluster at MIS5e in EDC is explained by the very low concentration of $Cl^-$ ions in the ice during this period and at this location.

(v) **Implications for the nature of the very deep ice:** We argue that the bottom thickness of the ice sheet deeper than approximately 2600 m plays a special role in shear deformation when the ice sheet moves away from the dome. Uneven strains between the layers will eventually compromise the integrity of the ice body in each part of the ice sheets. Consequently, folding, mixing, and faulting lead to the destruction of resolvable continuity in ice core signals. Ice core drilling, aiming the ancient climatic records, at locations away from the dome area carries serious risks of layer disturbances near the bottom. Echo-free zones for radar sounding can be explainable as zones with layer disturbances accompanied with enhanced dynamic recrystallization.

These numerous findings warrant further examination for a better understanding. The new data and insights should link directly to key processes governing the flow of polar ice sheets. Expanding the knowledge from the dome summit to a three-dimensional dynamic layer structure within the ice sheet is a critical challenge. Deciphering radar sounding data in depth is key to this endeavor. The internal deformation of the ice sheet modulates flow based on crystal orientation fabric, ion concentration, and microparticle presence, leading to differential ice movement or disturbances like folds and mixing at various layer thickness scales. These factors can introduce positive or negative feedback, modulating the flow characteristics of the ice sheet.

*Data availability*

The data used in this paper will be published in the National Institute of Polar Research ADS data repository in conjunction
with the publication of the present manuscript in The Cryosphere.
*Author contributions*
We list author contributions using a standard called CrediT (Ghan et al., 2016) to achieve greater clarity in contributions of all
authors. TS: Conceptualization, Methodology, Validation, Formal analysis, Investigation, Data curation, Writing - Original
draft, Visualization. AM: Conceptualization, Methodology, Validation, Formal analysis, Investigation, Data curation, Writing
- Review & editing, Visualization. SF: Conceptualization, Methodology, Validation, Formal analysis, Investigation, Writing -
Original draft, Supervision, Project administration, Funding acquisition. TK: Investigation, Writing - Review & editing. MI:
Investigation. KG-A, MH, AH, YI, HO, WS, and ST: Writing - Review & editing.
*Competing interests*
The authors declare that they have no conflict of interests.
*Acknowledgements*
The authors are grateful to all the Dome Fuji Deep Ice Core Project Members who contributed to obtaining the ice core samples,
either through logistics, drilling, or core processing. The main logistics support was provided by the Japanese Antarctic
Research Expedition (JARE), managed by the Ministry of Education, Culture, Sports, Science and Technology (MEXT). This
work was supported by JSPS KAKENHI Grant Number 18H05294.

**Appendix A: Supplemental information for Method**

**A1 Specifications the Two Resonators and Consistency of the Δε' Values.**

In this study, we utilized an open resonator for microwaves operating at frequencies between 26.5 and 40 GHz (Resonator No.1), differing from our previous study that used frequencies of 15 and 20 GHz (Resonator No.2) (Saruya et al., 2022a, b), especially in terms of ice core sample size limitations. Specifications of the two resonators, thickness and width of the samples are summarized in Tables A1 and A2, respectively. Using a high-frequency band allows for reduced sample dimensions due to the smaller beam diameter at higher frequencies. To verify the consistency between the current study (Resonator No.1) and the previous one (Resonator No.2), we conducted measurements of the Δε' across both frequency bands at depths of 2400 to 2500 m. Figure A1 presents a comparison of the dielectric anisotropy profiles and the standard deviation for each 0.5-m segment, using approximately 23 data points (0.02-m intervals) measured by both Resonator No.1 and Resonator No.2. No systematic differences were observed between the two resonators, though the standard deviation was slightly higher in Resonator No.1, due to its smaller beam diameter. Grain numbers included in a Gaussian beam or in a thin section are listed in Table A3. Depending on number of crystal grains in a beam or thin section, statistical significance is different. In some cases, moving averaging along the core is useful to gain more significance.

Table A1: Specifications of the two resonators

| Item | Curvature radius of the concave mirror (mm) | Distance between two mirrors (mm) | Scale diameter of the beam on the flat mirror (mm) | Frequencies (GHz) | Depths range (m) |
|---|---|---|---|---|---|
| Symbol | $R$ | $D$ | $\omega$ | $f$ | $z$ |
| Resonator No.1 | 120 | 110 | 16 | 26.5– 40 | 2400–2970 |
| Resonator No.2 | 250 | 225 | 38 | 14–20 | 100–2500 |

Table A2: Thickness and width of the samples

| Depths range (m) | Thickness (mm) | Width (mm) | Resonators used |
|---|---|---|---|
| 2400–2500 | 35–37 | 53–58 | No.1 and No.2 |
| 2500–2970 | 41–42 | 30–38 | No.1 |

Table A3: Grain numbers included in a Gaussian beam or in a thin section.

| Measuring size and grain area | Dimension (mm) | 0.1 cm² | 0.3 cm² | 0.5 cm² | 1 cm² | 2 mm² | 4cm² |
|---|---|---|---|---|---|---|---|

Thick-section-based method

| | | | | | | | |
|---|---|---|---|---|---|---|---|
| DTM No.1 | ~16φ × ~40 | 353 | 68 | 32 | 11 | 4 | 1 |
| DTM No.2 | ~38φ × ~70 | 3490 | 672 | 312 | 110 | 39 | 14 |
| Thin-section-based methods | | | | | | | |
| Laue X-ray diffraction method microscopy and G50 | 100 × 45 90 × 50 | 450 | 150 | 90 | 45 | 23 | 12 |

899

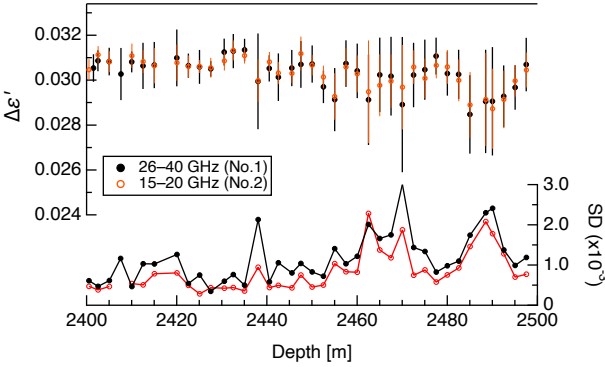

900
**Figure A1.** Comparison of the $\Delta\varepsilon'$ and standard deviation measured by Resonator No.1 (black) and No.2 (red).

## A2 Inclination of the *c*-axes Cluster and the DTM

In the main text, Figure 2 was provided to explain the geometry of the sample and the open resonator. If the *c*-axes cluster aligns with the vertical (i.e., the ice core axis), it is relatively straightforward to set the angle between the *c*-axes cluster and the electric field close to about 45°. This was the case for ice shallower than approximately 2400 m (Saruya et al., 2022b). Figure A2 presents an example of this frame projected onto a Schmidt net. In this frame, the axis of the *c*-axes cluster is always orthogonal to the beam's incidence, thereby lying on the plane of the electric field vector. However, issues arise when the *c*-axes cluster deviates from the vertical. Depending on the inclination angle of the *c*-axes cluster from the zenith and the rotation of the cylindrical ice core in horizontal directions, the axis of the *c*-axes cluster may not align with the plane of the electric field vector (refer to Figure A2b). In such cases, the DTM detects permittivity components projected onto the plane of the electric field vector. These are non-principal components of the tensorial permittivity. Consequently, the raw data of non-principal components of dielectric anisotropy, $\Delta\varepsilon'$ (as shown in Figure A2b), is smaller than the principal components of $\Delta\varepsilon$

when the axis of the $c$-axes cluster aligns with the plane of the electric field vector. Correction of data from $\Delta\varepsilon'$ to $\Delta\varepsilon$ is possible
only if both the inclination angle and the horizontal orientation of the $c$-axes cluster are known (see Figure A2b').
To enhance understanding of the measurement principles, Figure A2 is explained in greater detail here. Figure A2a depicts
the case where the $c$-axes cluster of single pole fabric aligns with the vertical. The diagram's center represents the incident axis
of the microwave beam in the resonator. Thus, the electric field vector is always orthogonal to the beam, spreading along the
diagram's periphery. The dark blue plane contains the girdle of the $a$-axes. The red line represents a plane that includes both
the beam axis and the $c$-axes cluster. In this setup, setting the electric field vector approximately within directions of $E_1$ or $E_2$
splits the vector components into two directions. This allows derivation of two permittivity components, $\varepsilon_v$ and $\varepsilon_h$, as
permittivities in the two principal axes along the diagram's periphery. Figure A2b illustrates a scenario where the $c$-axes cluster
of single maximum fabric is inclined at an arbitrary horizontal orientation. The diagram's center is again the incident axis of
the microwave beam. The dark blue plane contains the girdle of the $a$-axes, while the red line is a plane containing both the
beam axis and the $c$-axes cluster. The dashed line represents a plane perpendicular to the red plane and containing the beam
axis. In this arrangement, setting the electric field vector approximately within rotated directions of $E_1$ or $E_2$ still splits the
vector components into two directions. However, it results in two non-principal components, $\varepsilon'_v$ and $\varepsilon'_h$, as permittivity.
Knowing the inclination angle of the $c$-axes cluster ($\theta$ in Figure A2b') and horizontal orientation of the $c$-axes cluster allows
for accurate derivation of $\Delta\varepsilon$ values by rotating around the core axis to align the $c$-axis maximum within plane of the electric
field vector (as shown in Figure A2b') and calculating the geometrical effects of the inclination angle of the $c$-axes cluster.

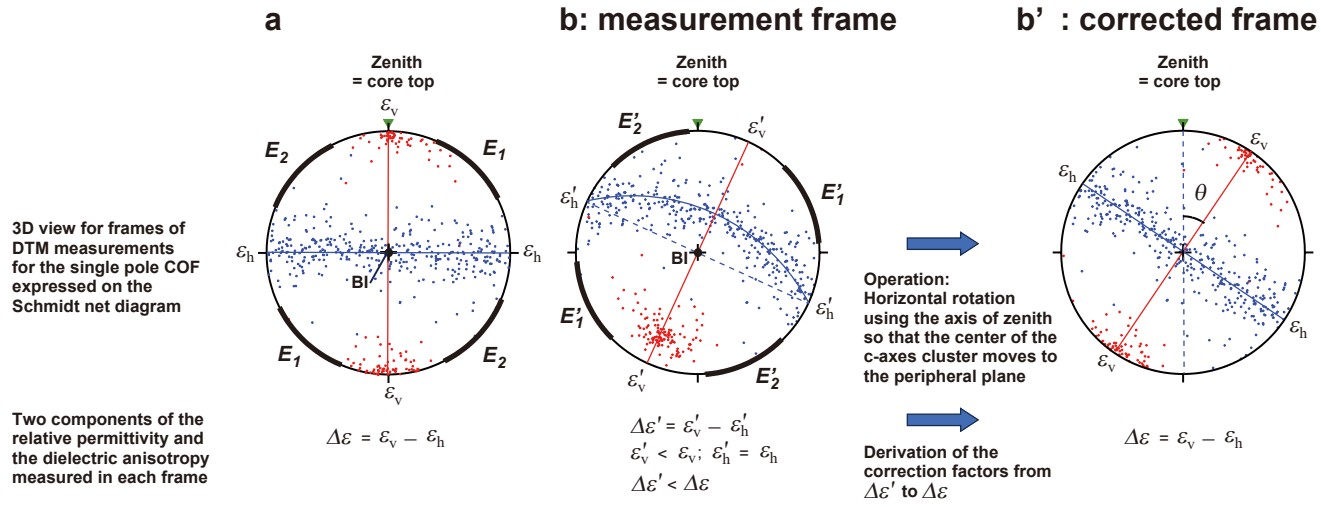


**Figure A2.** An explanation of the 3D geometry of systems composed of core samples, crystal axes, and applied electric fields, represented
using projections on the Schmidt net diagram. For three figures from (a) to (b'), both $c$-axes (red dots) and $a$-axes (dark blue dots) are
presented for an imaginary ice core sample. In (a) and (b), the center of the diagram represents the axis of incidence for the microwave beam,
with the peripheral circle forming a plane orthogonal to the beam. The top of this peripheral circle corresponds to the vertical orientation
within the ice sheet, which is also the top of each ice core. Case (a): The $c$-axes cluster of single pole fabric aligns with the vertical, and the
$a$-axes girdle lies on the horizontal plane. If the microwave beam's electric field vector falls within the orientation ranges of $E_1$ or $E_2$, it
induces birefringence, allowing us to derive two components of permittivity, $\varepsilon_v$ and $\varepsilon_h$, as principal components of the tensor. Consequently,
we can calculate the difference as $\Delta\varepsilon$. Case (b): Here, the $c$-axes cluster of single pole fabric is inclined to an arbitrary horizontal orientation.
The girdle of the $a$-axes deviates from the beam axis. In this scenario, using the electric field vectors from (a) within the orientation ranges
of $E_1$ or $E_2$ is ineffective for detecting birefringence, as the splitting of waves into two components is not equally balanced. However, if the
microwave beam's electric field vector is within inclined range from $E_1$ or $E_2$, namely $E'_1$ or $E'_2$, balanced birefringence is achieved, and
we can derive two non-principal components of permittivity, $\varepsilon'_v$ and $\varepsilon'_h$. This results in the calculation of the non-principal difference as $\Delta\varepsilon'$.
Case (b'): This is the same as (b), but the system is horizontally rotated so that the $c$-axes cluster aligns with the periphery of the diagram
where the electric field vector is located. By comparing permittivity values between cases (b) and (b') using the $c$-axis fabric data, we can
determine factors to adjust $\Delta\varepsilon'$ values to $\Delta\varepsilon$ values.

## A3 Detail of the layer inclination measurements

The inclination of the layers in the Dome Fuji ice core was measured using two methods, as follows: Thin, cloudy bands
are faint features with thicknesses ranging from about 10 mm to 1 mm. We observed the ice cores using a light stage,
approximately 250 mm by 600 mm in size, placed on a table. The ice cores, shaped as half vertical cuts of the original
cylindrical form, were visually inspected by observers who looked directly down at the core from above the light stage. The
ice cores were positioned between the observers' eyes and the light stage. The faint and thin cloudy layers are identified when
oriented vertically; only in this orientation do the observers recognize the layers as sharp lines. From angles deviating from
this, it is difficult to recognize such layers. By orienting the layers vertically and keeping them orthogonal to the core's
inclination on the light stage, the observers could measure and record the inclination angle of the ice cores using a large
protractor. With this procedure, layer inclination was measured in a 3D manner, and we always measured the maximum
inclination angle. In some cases, we employed another method where the observers used the coordinates of three or more
points in each layer within ice cores shaped as half of the core. Using these points, we measured the inclination angle of the
layers in 3D, ensuring the orientation was at the point of maximum layer inclination. At least two observers (in most cases,
Miyamoto and Fujita) measured the inclination angles for each layer as a cross-check. Additionally, we repeated measurements
several times to gain skills and confirm reproducibility. The observers estimated that the maximum errors in measurement
were about 5 degrees. The orientation of the inclination was not recorded because core orientation was not a topic of interest
at that time. However, we checked if these two are in the same vertical plane or not. We conformed that the horizontal
orientation of the $c$-axes cluster and the normal axis of the layer inclination are within the same vertical plain throughout the
LO20%.

## A4 Estimation of the error range of the normalized eigenvalues for the $c$-axis fabric data obtained through thin-section methods

Generally, if the total number of grain samples is N (in our Laue example, ranging from about 50 to about 250), the addition
of a grain means the normalized eigenvalue can vary by about 1/N at maximum. When $c$-axes are clustered, as in the Dome
Fuji core, the range of variation will be much smaller than this 1/N. Assuming each grain contributes an eigenvalue between
2/3 and 1 (see Figure 5a), the range of variation will be 1/(3N) or less. Therefore, the total number of grains is very important.
The error in the eigenvalue is on the order of 1/(3N) or less.

## A5 Microstructural observations

We analysed grain shape using ImageJ software. Table A4 shows the aspect ratio value defined as the ratio of the short and
long axis of a fitted ellipse. In the table, sample depths, concentration of dust particles, average and standard deviation of the
aspect ratio are listed. In some depths, we used several thin sections to obtain the number of crystal grains (indicated by
annotation). We found flattened (or elongated in 2D) grains in samples from depths of 2540, 2648, 2653.5, 2673.5, 2759.5 m.
(see brown shading in Figure 5). There, the aspect ratio ranges from 1.9 to 2.0. In contrast, the aspect ratio in deep impurity-
rich layers (samples from depths of 2907.5 and 2949 m) and impurity-poor layers ranges from 1.5 to 1.6. Shallower than a
depth of 2600 m, depths of 2518.5 and 2540 m are located in impurity-rich layer (see Figure 5). The concentration of dust
particles in 2518.5 m is not so high (~32 ppbv), so there is no evident elongation, and the aspect ratio is not large.
Table A4: Aspect ratio of grains analysed by Image J software using thin sections. Sample depths, concentration of dust
particles, grain number, average and standard deviation (SD) are listed. The columns with bold text correspond to the impurity-
rich layer indicating slanting and flattened (or elongated in 2D) grains.

| Depth [m] | 2470 | 2490 | 2500 | 2518.5 | **2540** | 2623.5 | **2648** | **2653.5** | **2673.5** |
|---|---|---|---|---|---|---|---|---|---|
| Concentration of dust particles [ppbv] | ~3 | ~1 | ~7 | ~32 | **~132** | ~4 | **~273** | **~221** | **~166** |
| Grain number | 122[a)] | 90[b)] | 44 | 121 | **167** | 43 | **335** | **238** | **244** |
| Average | 1.7 | 1.6 | 1.7 | 1.7 | **1.9** | 1.5 | **2.0** | **1.9** | **1.9** |
| SD | 0.6 | 0.5 | 0.5 | 0.6 | **0.5** | 0.3 | **0.7** | **0.6** | **0.6** |


| 2685 | 2730 | **2759.5** | 2807 | 2847.5 | 2868 | 2872 | 2887 | 2907.5 | 2949 |
|---|---|---|---|---|---|---|---|---|---|
| ~12 | ~10 | **~137** | ~4 | ~34 | ~2 | ~3 | ~10 | ~91 | ~51 |
| 55 | 96 | **262** | 44 | 107[a)] | 41 | 42[a)] | 45 | 56 | 48 |
| 1.6 | 1.6 | **1.9** | 1.5 | 1.5 | 1.6 | 1.6 | 1.5 | 1.6 | 1.5 |
| 0.5 | 0.4 | **0.7** | 0.5 | 0.4 | 0.5 | 0.5 | 0.6 | 0.3 | 0.3 |

985 a) Sum of two thin sections
986 b) Sum of three thin sections

987  Figure A3(a and b) shows closeup images of microstructures. Each panel corresponds to the top-right sections of Figures
988 5a and 5b images. Here, we illustrated grain boundaries, grain boundaries on the reverse side, and subgrain boundaries
989 speculated from geometry by colored shading.

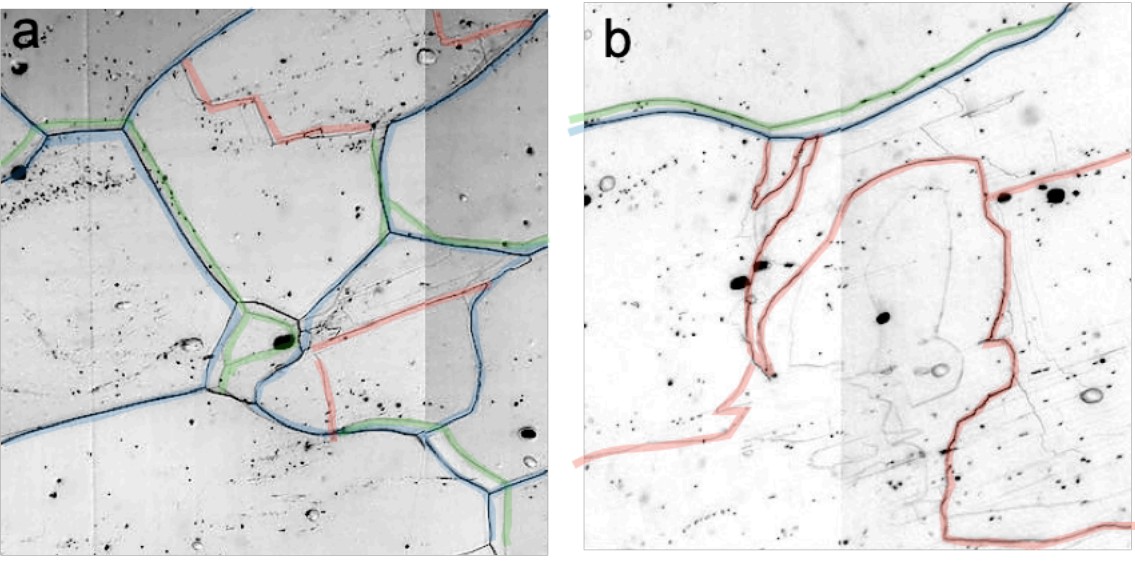

990
991 **Figure A3**. Closeup images of microstructures (samples from depths of 2648 and 2685 m). Each corresponds to the top-right
992 of the original images in Figures 5a and 5b. Grain boundaries (blue lines), grain boundaries on the reverse side (green lines)
993 and subgrain boundaries (red lines) speculated from geometry are illustrated. Scale bar: 2mm.

994
995

**Appendix B: Corrections from $\Delta\varepsilon'$ to $\Delta\varepsilon$**

Using data from the Laue X-ray diffraction measurement taken at 42 different depths, we initially examined both the inclination angle and the horizontal orientation of the $c$-axes cluster. We assumed that the inclination angle of the $c$-axes cluster consistently develops towards the same horizontal orientation within the ice sheet, a plausible assumption given the unlikelihood of horizontal rotation in ice flow under deep englacial conditions at the dome summit. In the ice sheet, there is a possibility that the inclination angle of the $c$-axis cluster may rotate with increasing depths if the dome summit position migrates in a complex manner. Another possibility is that when handling many cylindrical ice cores, sometimes the continuity of orientation between adjacent cores is lost, especially when irregular ice core breaks occur. When we perform the DTM measurements, we do not know the core orientation in advance. We can determine the orientation only from the Laue data (or orientation of layer inclination if they were recorded). Only with the assumptions we made were we able to apply corrections from $\Delta\varepsilon'$ to $\Delta\varepsilon$. The inclination angle of the $c$-axes cluster is presented in Figure 5f of the main text. We observed that the horizontal orientation of the $c$-axes cluster gradually rotates with increasing depth (Figure B1). To address the data scatter, we proposed a fitting curve as shown in the figure. Notably, some data scatter between approximately 2400 and 2700 m leads to significant deviations from the fitting curve. We calculated the tensorial components of $c$-axes projected onto the diagram's periphery and further computed eigenvalue anisotropy (the difference between the maximum and minimum) by combining the inclination angle of the $c$-axes cluster (Figure 5f) and the horizontal orientation of the $c$-axes cluster (Figure B1) for orientations where the $c$-axes cluster aligns with the periphery of the Schmidt net diagram (as shown in Figure A2b'). The results are presented in Figure B2. Using these data, we estimated coefficients for correcting $\Delta\varepsilon$ to $\Delta\varepsilon'$, detailed in Figure B3. We found that the necessary corrections are minor (up to several percent) at depths shallower than about 2730 m but become more significant at deeper depths (up to about 20 percent). At depths between approximately 2400 and 2730 m, where large deviations from the fitting curve were observed (Figure B1), the resulting errors in corrections are likely limited due to the inclination angle of the $c$-axes cluster remaining small, less than about 20 degrees, in this depth range (Figure 5c). Therefore, we estimate the errors in our correction coefficients to be less than 10%.

Figure B4 displays the $\Delta\varepsilon'$ and $\Delta\varepsilon$ values, along with eigenvalue anisotropy obtained from the Laue X-ray diffraction measurements. At depths exceeding 2900 m, the corrections incorporate errors due to continuity uncertainties between core samples, which arise from unpredictable core rotation. Consequently, sudden changes in $\Delta\varepsilon$ values may be erroneous. Nevertheless, we observed a good agreement between the $\Delta\varepsilon$ values from the two ice cores (DF and EDC) at ages older than approximately 630 kyr BP, corresponding to the ice age at 2900 m.

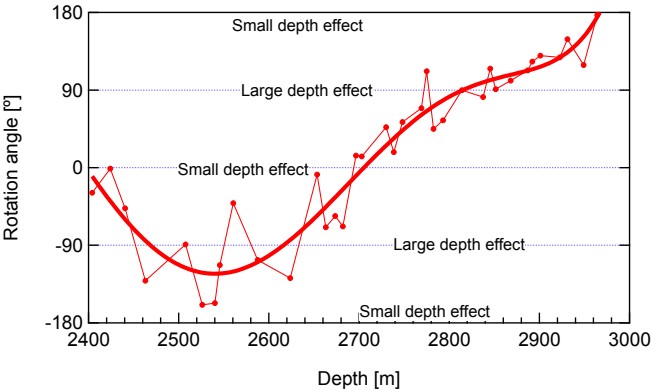


**Figure B1.** Horizontal orientation of the *c*-axes cluster (left axis) was derived from the *c*-axis fabric data obtained through the Laue X-ray
diffraction measurements. Angles of 0, 180, or –180 degrees indicate that the *c*-axes cluster is perpendicular to the observer's line of sight
and parallel to the electric field vector of the electromagnetic waves. Conversely, angles of 90 or –90 degrees indicate that the *c*-axes
cluster is parallel to the observer's line of sight and perpendicular to the electric field vector. When the horizontal orientation of the *c*-axes
cluster measures 0, 180, or –180 degrees, it aligns with the periphery of the Schmidt equal area projection.

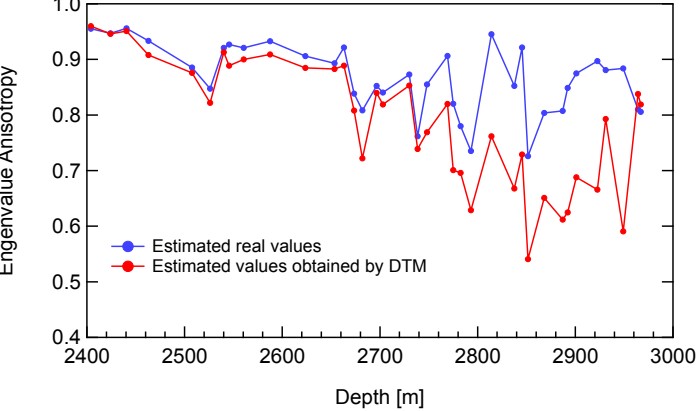


**Figure B2**. Comparison of eigenvalue anisotropy values between measurement frame (indicated in red) and corrected frame (indicated in
blue).

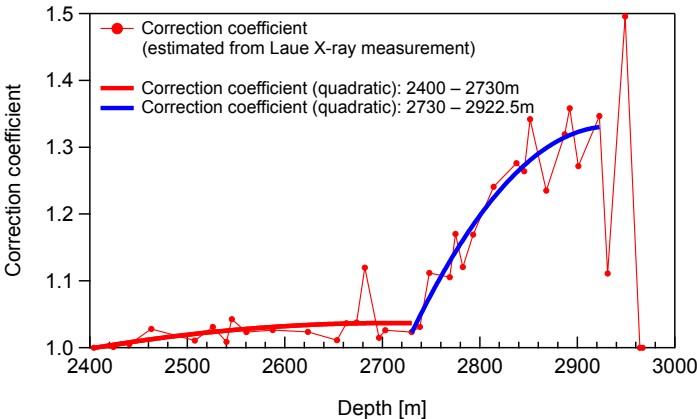


**Figure B3.** Correction coefficients for converting $\Delta\varepsilon'$ to $\Delta\varepsilon$. This figure presents proposed fitting curves for depths ranging from 2400 to 2730 m and for depths greater than 2730 m.


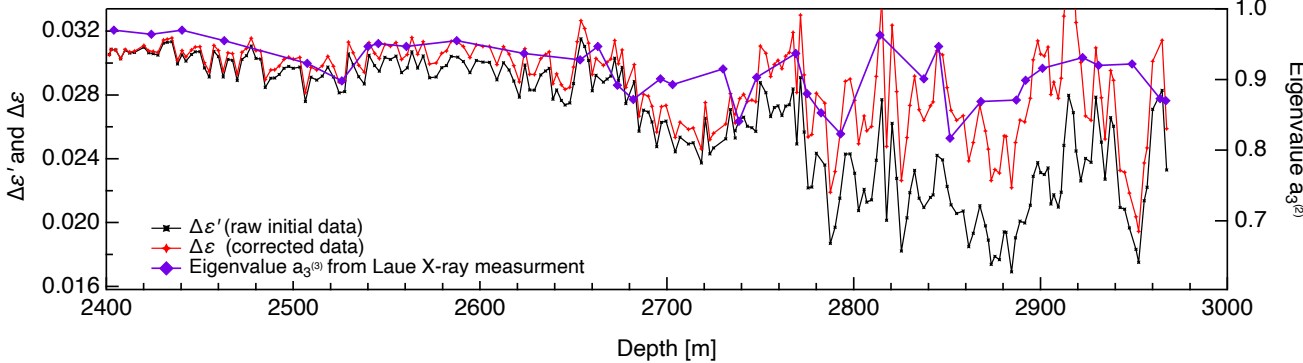


**Figure B4.** Correction of $\Delta\varepsilon'$ (depicted in black) to $\Delta\varepsilon$ (shown in red) and comparison with eigenvalues $a_3^{(2)}$ (illustrated in purple) as estimated from the Laue X-ray Diffraction Measurements. It is important to note that the DTM data represent average values for each 0.5 m section using thick-sections, while the Laue X-ray diffraction measurement data are derived from thin sections. The volume of ice represented by a single data point differs by approximately $10^2$ times. Additionally, the DTM method provides volume-weighted values of $c$-axis fabric, whereas the Laue X-ray diffraction measurements yield an average value across the total number of crystal grains, typically not accounting for the size (volume) of each individual crystal grain.


**Appendix C: Continuous Variations in $\Delta\varepsilon$ and Comparison with Microstructures**

Figure C1 showcases examples of the continuous variation of $\Delta\varepsilon$ across 0.5 m core samples at five depths (125.0–125.5,
1300.0–1300.5, 2648.0–2648.5, 2707.0–2707.5, and 2823.0–2823.5 m). The shallowest 2 depths were obtained from Saruya
et al. (2022b). The mean values and standard deviations for each 0.02 m segment were calculated using various TEM $_{0, 0, q}$
resonance modes in the open resonator method. For the data in Figure C1, the mean values (and SD) at these five depths are
0.0076 (0.0005), 0.0194 (0.0006), 0.0285 (0.0008), 0.0263 (0.0036), and 0.0282 (0.0043), respectively. Fluctuations within
the 0.5 m ice core samples become more pronounced at greater depths. However, in the core sample with a higher concentration
of dust particles (at 2648.0 m, represented by brown plots), the fluctuations are not as marked.

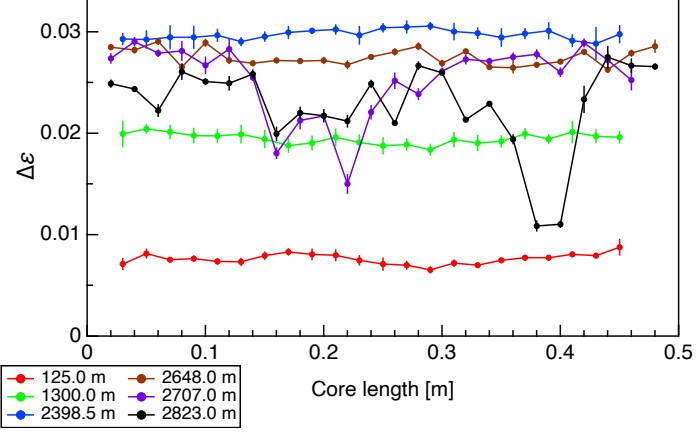


**Figure C1.** Examples of variations in $\Delta\varepsilon$ along 0.5 m sections of ice cores, as determined by continuous measurements. The bars display
standard deviations derived from various resonance modes in the open resonator.


Figure C2 presents an example of comparing the $\Delta\varepsilon$ profile with microstructures, specifically for the 2490.0–2490.5 m
sample. The upper and lower panels show fabric data and polarized images, respectively. Within the $\Delta\varepsilon$ profile, two locations
exhibit lower $\Delta\varepsilon$ values (at 0.11 and 0.27 m). At these depths, grains are observed with $c$-axis orientations differing from those
of the surrounding grains. These layers contribute to a decrease in the mean $\Delta\varepsilon'$ value and an increase in the SD value for this
0.5 m core section.

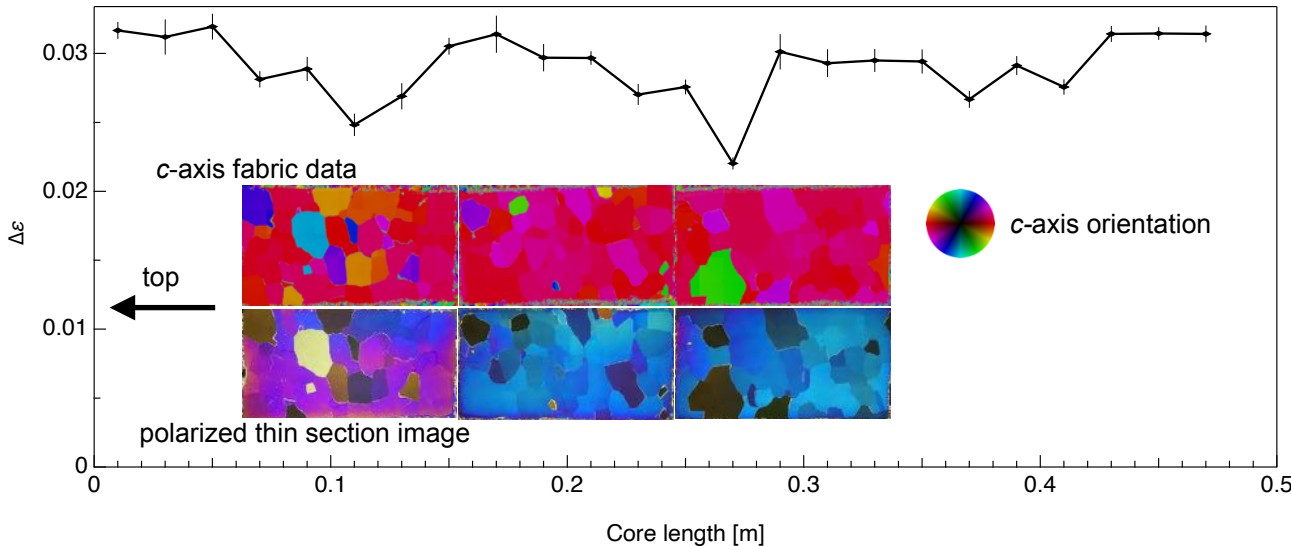


**Figure C2.** An example of comparing the Δε variation along a 0.5 m ice core section with microstructures (*c*-axis fabric data and polarized
images) obtained from thin-section measurements (2490.0–2490.5 m). The bars in the Δε variation graph represent standard deviations from
different resonance modes of the open resonator. The horizontal scales in both the Δε variation graph and the microstructural images are
aligned for consistency.

**Appendix D: Modified Figure 10 highlighting the Termination II and MIS 5**

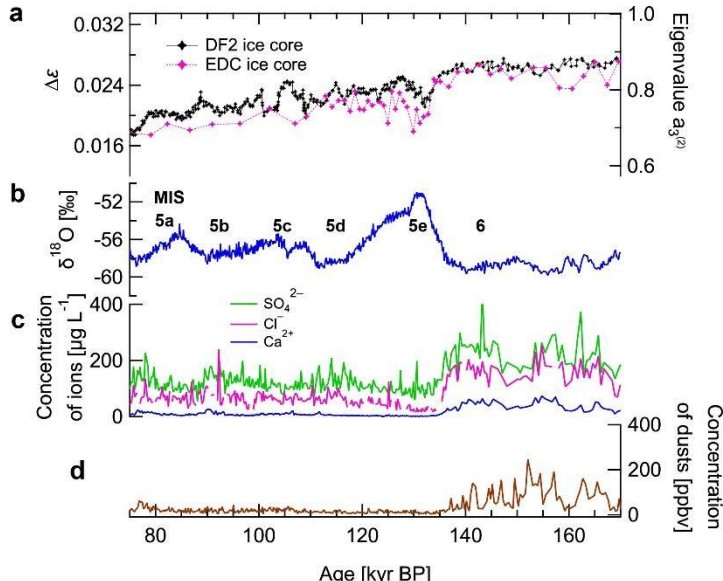

1075

**Figure D1**: Modification of the Figure 10. Crystal orientation fabric data from the DF core and the EDC core are compared using a common age scale of the DF2021 age scale. (a) $\Delta\varepsilon$ presented as mean values for each 0.5-m segment (black markers for DF) or for each thin section (magenta marker for EDC). The right axis provides a scale for normalized eigenvalues. (b) Oxygen isotope ratios ($\delta_{18}O$) in the DF ice core. (c) Concentrations of $Cl^-$, $SO_4^{2-}$, and $Ca^{2+}$ ions (Goto-Azuma et al., 2019). (d) Concentration of dust particles (DFICPM, 2017). For ice at EDC at MIS5, the normalized eigenvalues are smaller than those at DF.

1081

1082

1083

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
