# Peer review of "Development of crystal orientation fabric, microstructures and"

_EGUsphere, 2023_

## Referee Comment (RC2)

Review of Saruya et al.,: "Development of deformational regimes and microstructures in the deep sections and overall layered structures of the Dome Fuji ice core, Antarctica"
By David Prior, University of Otago

This paper contains some very interesting and potentially important new data and can make a significant contribution to ice core science. The use of the Dielectric Tensor method to get a much higher resolution data set for COF strength as a function of depth is particularly impressive. The paper also presents full crystal orientation data ([c] and <a> axes and this is an important development as c axes alone do not usually give the full information needed to infer deformation kinematics. Although the paper represents a substantial contribution to ice core science, it needs a lot of re-writing and possibly some more research work.

The paper is way too large and tries to cover too much. Any significant outcome is lost as the paper is poorly focused. I would focus on the orientation relationships of COF to layering and the evidence from the microstructure that helps understand deformation kinematics and mechanisms. The cross-correlation work linking COF to chemical impurities seems superfluous and could be thinned down and simplified. There is a lot of repetition in the figures and the paper can be shortened and improved by a redesign of the figure structure. For example figs 7 and 8 can be merged, figs 5 and 9 can be merged.

The really key new information is the relative orientation of layers and c-axis maxima. The analysis of these data is not robust as it does not consider the orientation relationships in 3 dimensions. I am pretty sure that the authors already have the information they need to do the analysis in 3D or can get this information with a little extra work and this should be done. If grain shapes were incorporated into this it would be even better. I start my review with this main point, followed by other key issues that I think the authors should address. Apologies I have run out of time on this so have not listed all the minor issues.

The paper is very difficult to read because of the ridiculous number of acronyms. Many of the acronyms represent long expressions that could easily be replaced with much shorter expressions, that could then be used as words so that the reader can read the paper without constant reference to an array of acronyms. This might make the paper a little longer, but it will take less time to read and the reader will understand it better. Some of the acronyms relate to words that do not clearly indicate meaning. I will send an annotated version of the acronym list later.

**Specific comments.**

**Major issue: Layering, c-axis inclination and core inclination.**
The relative orientation of layering and c-axis maxima is really important. In this paper it is presented in a way that is ambiguous and potentially misleading. "Layered structures" is in the paper title so it is crucial that there is no uncertainty as to the nature of the constraints on layer orientation.
Firstly, the description of how layering is measured in this manuscript and in the cited report (DFICPM: et al., 2017) is incomplete. It simply says it was measured with a protractor - it is not clear whether this is a measurement on an arbitrary cut surface (in

which case the true inclination is equal to or larger than the measured value) or whether it is a measure of the maximum inclination (true dip) - e.g. by cutting the core along a plane that contains the maximum inclination (perpendicular to strike). This needs to be clarified and the layering added schematically to figure 2.

There is no discussion of 3D geometrical relationships. The presentation of inclinations of c axis maxima and visible layers in figure 5 and the written interpretation suggests to the reader that the c-axis maximum and the pole to layering has the same azimuth and lie in a vertical plane. If this is the case, then the cartoon in fig 10c is valid. However, the inclination constraints as presented have an infinite set of possible orientations on cones around the core axis as shown on the stereonet below (Fig. R1). End member solutions where the c-axis maximum and the pole to layering are in the same plane as the core axis (see Fig R1. below) are quite different. An added complication is that the core axis is not vertical, with an inclination of 3 to 6 degrees in this part of the hole (Motoyama et al., 2021), although this angle is small and unlikely to cause significant complications in analysis.

[Figure]

Fig R1. Possible 3D relationships of c-axis maxima and poles to layers

I think the authors can do much better here. If there is visible layering in the core then you should be able to measure it in 3D; either by extracting the sine curve of the layer from the outside of the cylindrical core (although I realise that the cylinder probably does not exist now), by measuring the apparent inclination on two non-parallel surfaces or by cutting a plane that contains maximum inclination (maybe this is done, as I said before). A great circle (with a point for the pole) can then be plotted on each stereonet along with the COF data (fig 4) to show the orientation of layering to show the true 3D relative orientation of the c-axis maximum and layers in that sample. If layers are hard to see in the core, they are usually easy to see in a 5mm slice viewed in crossed polars.

With 3D data the relative orientations of layers, core axis and c-axis max can be shown for all samples in one stereonet. Below (Fig R2) I present three possible data outcomes (Fig R2 b,c,d) presented in a reference frame where the core axis is fixed (plotted

in the centre of the stereonet, as it is easier to see the patterns that way) and the c-axis maxima are all assigned to a common azimuth (arbitrary as you do not have azimuthal data).

[Figure]

Fig R2. Possible outcomes of measured 3D relationships of c-axis maxima and poles to layers

This figure could equally be plotted by assigning a common azimuth of the poles to layers - but as drawn it would match better with layers marked on individual sample data in figure 4. The outcome in Fig R2d would be consistent with the generalized cartoon in fig R1b and the geometrical interpretation provided in the paper in fig 10c. The outcome in Fig R2c would be consistent with the generalized cartoon in fig R1c. You cannot draw figure 10c in the paper, nor make any useful interpretation of the c-axis and layer inclinations without this analysis in 3D. Without this data are pretty meaningless.

A slightly simpler way of plotting Fig R2 is shown below in Fig R3. This may be easier to read.

[Figure]

Fig R3. Figure equivalent to b, c and d of fig R2. Maybe clearer like this.

**A few other things related to orientations**
In Appendix D there is a statement: "We assumed that the IACC consistently develops towards the same horizontal orientation within the ice sheet…". I agree that this is plausible, but I don't think you need to make this assumption. Fig C1b is in the measurement ref frame: the transformation of $\Delta\varepsilon'$ to $\Delta\varepsilon$ is simply a function of the vector orientation of the c-axis maximum relative the measurement reference frame. There should be no need to make any assumption about how this varies from one core piece to the next? Appendix C is easy to understand. (although line 801:"horizontally rotating the frame…" is not a good description: "rotating around the core axis to align the c-axis maximum within plane of the electric field vector" would be better). I really do not understand what Appendix D is saying. Is this about correcting measurements between different COF measurements??

It is unclear whether there are constraints independent of the core on the layering orientation in the DF hole. Are there televiewer data (Hubbard and Malone, 2013)? Are the radar data (Karlsson et al., 2018; Wang et al., 2023) good enough to extract the dip direction at the borehole site?  What are the constraints, other than from the core, on the layer structure shown in cartoon form in fig 10. Knowing the dip direction could turn the data into 3D in a geographic reference frame.

**Eigenvector analysis outcomes: error related to the number of grains.**

Robustness of the eigenvalues, eigenvector orientation (Inclination angle of c-axis cluster) and median cone half angle (median inclination of c-axis cluster) from the c-axis measurements depends on the number of grains measured. None of the five deepest samples (as shown in fig 4) have enough grains to provide robust eigenvector measurements and points related to these (eigenvalue, Inclination angle of c-axis cluster, median inclination of c-axis cluster) should not be shown on figure 5 or any other figure. I wonder whether other samples with large grain size, particularly between 2860 and 2890 m also have too few grains to make the eigenvector analysis robust.

Error analysis for eigenvectors is tricky - a simple way to do this is, is to recalculate the eigenvectors for different randomly selected subsets of the full data set. Since all of the c-axis patterns (where there are enough data) seem to be single maxima I suggest that you do this with one sample to give a representative value for errors of all the eigenvector derived numbers for all samples. Pick a data set with many grains (e.g. the data in fig 4 b). Compare the eigenvector results for ten randomly selected sets of n grains from this larger data set. Use n values from 10 to 200 (in increments of 10?) to get an idea of the errors for the parameters you show, as a function of the number of grains in the data set. Then all the parameters derived from eigenvector analysis can she shown with error bars corresponding to the number of grains measured.

You should show all of the stereonets of c-axis and <a>-axis data (in an appendix?) and tabulate the measurements (e.g. eigenvalues) for all the samples. The paper refers to fabric analyser data as well as Laue data. Where are these data? They should be included.

**Elongate [c] axis patterns**

The c-axis maxima in fig 5 are not all circularly symmetric. Subfigs a,b,d,e,g, and h are all elongate. You should highlight this observation. This is important because all laboratory experiments in shear give elongate c-axis maxima (Bouchez and Duval, 1982; Journaux et al., 2019; Kamb, 1972; Li et al., 2000; Qi et al., 2019; Wang et al., 2024) - with elongation perpendicular to the shear direction. The elongation is less clear in natural samples from shear zones: it is present in the Whillans shear margin (Jackson and Kamb, 1997) but absent in other studies (Disbrow-Monz et al., 2024; Monz et al., 2021; Thomas et al., 2021).

It would be good to know whether this is a parameter that changes between the UP80% and the LO20%. I cannot find any c-axis stereonets related to UP80% (Saruya et al., 2022) in although I note that samples 1720m and deeper in DF1 (Azuma et al., 1999) have elongate c axis clusters.

**<a> axis data**

The <a> axis data are not well used. They are, in part, hard to evaluate because the quality of figure 4 is poor (this figure needs to be better). You say in the text that the <a>-axes girdles have no maxima - I don't believe that, it looks to me like some of the girdles contain maxima. Fig 5 g, h and I all look to have maxima in the middle of the stereonets in the corrected view. Remember that the maximium value for any measure of <a> axis orientation density must be less than 1/3 the maximum value for the [c] axis maximum, so <a> axis patterns will always look more subtle than [c] axis patterns. The hexagonal repeat of <a>-axes renders an eigenvector analysis uninformative. You need to present <a>-axes in a contoured form to evaluate them. To compare <a> axis and [c] axis patterns they need to be contoured independently, so that both are scaled to a maximum value for that crystal direction.

This is pretty important as a distinct <a> axis maximum would add strength to the inference that these are COFs related to shear. All shear experiments where <a>-axes have been measured have a preferred <a>-axis maximum (Journaux et al., 2019; Qi et al., 2019; Wang et al., 2024). Natural examples from sheared ice (Disbrow-Monz et al., 2024; Monz et al., 2021; Thomas et al., 2021) also have a preferred <a>-axis maximum, although in the case of Thomas et al it is variably developed.

If you do have a -axis maxima they could also be plotted on a figure like Fig R2 or Fig R3. How to do this would depend on the nature of the data, but I am certain that they would be useful.

I think your Laue data are one point per grain? In this case there will be insufficient data to calculate crystal vorticity axes (Kruckenberg et al., 2019; Michels et al., 2015). It would be worth thinking about collecting full crystal orientation data at higher density (EBSD data) to enable crystal vorticity axis (CVA) analysis. Obviously I'm not suggesting you need to do this for this paper- an idea for the future. (Thomas et al., 2021) have very clear CVA maxima that are consistent with the dominant simple shear that is also constrained from other data- with the very strong single point maxima you have a CVA analysis may provide excellent constraint on deformation kinematics.

**Relevant comparative data from the literature.**

A key part of the paper is about what might be controlling the COF, including kinematics mechanisms and conditions. Excellent constraints on these come from laboratory experiments and field studies where deformation kinematics and/or conditions are constrained. There is virtually no reference to this extensive literature. I have already related (in the previous two sections) a couple of your observations to the literature for COFs from experiments and kinematically constrained. This is sorely needed if we are to understand ice cores such as DF where there are no measured constraints on deformation kinematics (the thinning model is not a measured constraint- it is a model with imposed kinematic assumptions).

A good starting point is the fact that your c-axis patterns are very tight single maxima, with most of them close to vertical. This is common in palaeo-climate focused ice cores (Faria et al., 2014). Many of the papers that describe these data infer that the primary cause of the vertical tight c axis maximum is vertical uniaxial compression due to lattice rotation. This is entirely at odds with the experimental data. There are no uniaxial experiments published that have tight c-axis maxima parallel to compression. Virtually all

uniaxial experiments (my list of refs is just a subset) have open cones (small circle distributions) with the cone axis parallel to compression (Budd and Jacka, 1989; Fan et al., 2020; Hunter et al., 2023; Jacka and LI, 2000; Jacka and Maccagnan, 1984; Montagnat et al., 2015; Qi et al., 2017; Vaughan et al., 2017). Single maxima can form in uniaxial compression at high stress corresponding to high strain rates and or low temperatures (Fan et al., 2020; Qi et al., 2017) but these are weak maxima, tight maxima never form under these conditions. Lower stresses, as expected in nature, would tend towards open cones. Tight single maxima are observed in experiments, but only in shear, where they form normal to the shear plane (Bouchez and Duval, 1982; Journaux et al., 2019; Kamb, 1972; Qi et al., 2019; Wang et al., 2024).

I think that some general discussion as to why you have tight single maxima, with reference to the experimental literature is needed. I suspect it is dominance of shear on a shallow shear plane at all depths. I don't think there are many direct measurements of deformation kinematics in deep ice boreholes: (Treverrow et al., 2015) show that shear strain rate (on shallow shear plane) is much vertical shortening at all depths in the Law Dome boreholes.

My knowledge of the geometry of c-axis maxima relative to shear kinematics together with some basic knowledge of structural geology leads me to suggest that he cartoon in figure 10c (if it is correct: see section on layering) requires a sense of shear opposite to what is alluded to by the authors. In shear, the c-axis maximum will remain perpendicular to the shear plane as seen in all ice experiments (Bouchez and Duval, 1982; Journaux et al., 2019; Kamb, 1972; Li et al., 2000; Qi et al., 2019; Wang et al., 2024), with no significant rotation as a function of shear strain. In shear, layering will rotate so that the pole to layering will be aligned with the short axis of the finite strain ellipsoid (Fossen and Cavalcante, 2017; Hudleston, 2015; Jennings and Hambrey, 2021). These two relationships, shown on the left side of the Fig R4, are consistent with observations from ice shear zones (Hudleston, 1977; Thomas et al., 2021). On the right side of the figure I have rotated the picture to match the reference frame of fig 10c.

[Figure]

Shear relations of c axis max to layers    Rotated into ref frame equivalent to Fig 10c

**Microstructures and their interpretation**

Section 4.5 is not too bad, although there are some unintelligible bits (It is noteworthy that the concentration of the less impure ice is markedly smaller than in impure ice??). The key issue in this section is to make clear in Fig. 6 what are grain boundaries and

what are sub-grain boundaries. The arrows are not good enough. I would suggest that an additional column is added with the photo from column 3 repeated, but with an overlay with three different coloured lines for boundaries on the top side, boundaries on the bottom side and subgrain boundaries. The description of boundaries being concave towards complex subgrain boundaries is not correct, wishful thinking I suspect. In both examples cited in the text (a and b in Fig 6) there are both concave and convex boundary segments adjacent to the subgrain structures.

I think you need to locate the micrographs in column 3 of Fig 6. on the micrographs in a and b. This is important for the reader to assess whether boundaries are grain or subgrain boundaries. The only one I can locate myself is b. The "subgrain" structures highlighted in b are weird. I've never seen subgrains like this in any naturally or experimentally deformed ice. I have seen grain boundaries like this and this could be a grain boundary with c-axes closely aligned but a-axes misaligned by 10-15 degrees?

Observations from sections 5.3.1 and 5.3.2. should be incorporated into the descriptions in section 4.5, so all the microstructural observations are together. Section 5.3.1 and 5.3.2. are both really poor: microstructural observations and interpretations are often intermixed and many features are described in interpretive terms (e.g. "migrating boundary" used as a description). This is poor science as the paper omits clear microstructural observations (facts that will never be wrong) that enable future researchers to build new interpretations on these.  Sorry I have run out of time to make my comments clearly structured so the following sections might be a bit of a mess. There is a lot to comment on as these microstructural sections need extensive re-evaluation and re-writing for both the observations and the interpretations.

I fail, to see the "brick wall" patterns you describe. You need to show a lower magnification microstructure, with many more grains for the brick wall structure to be convincing: I doubt it will be from what I can see. None of the micrographs in Fig 6 or in Fig 12 look like Fig. 9 in Faria et al, 2009. (Weikusat et al., 2017) does not show "brick wall" patterns but infers their possibility based on very high grain elongation data: I can't see mean elongations that compare with Weikusat f(ig 4c) or individual grain elongations that compare with Weikusat (fig 4) that justify a comparison. The other cited reference (Kuiper et al., 2020b) does not use the term "brick wall" (nor does (Kuiper et al., 2020a)- this is a better comparison to your data as it is the deep part of NEEM).

The inference of microshear processes based on the description provided is unjustified. What is the evidence for shear being localized? I can believe, as a general assumption, that it may have been localized but you show no clear evidence. A shape fabric does not mean localization unless it varies in orientation reflecting variable strain (Hudleston, 1977, 2015; Jennings and Hambrey, 2021; Ramsay, 1980). The grain shape fabric provides really useful information about deformation kinematics – you should use it. Grain long axes at 2648m (fig 6a) are ~ 60-70 degrees clockwise of the vertical on the picture. If the c-axis max is vertical in this pic, then this would imply shear with top to the right (Bouchez and Duval, 1982; Fossen and Cavalcante, 2017; Hudleston, 1977). The shape fabric is another thing that would be very usefully oriented relative to layering and the COF.

The paper by Faria et al, 2009 predates a lot of work that relates to the GBS process in rocks and ice. Rock deformation studies report small recrystallized grains have CPOs that are randomly dispersed equivalents of the stronger parent grain COFs (Bestmann and Prior, 2003; Jiang et al., 2000; Warren and Hirth, 2006) (and many more recent papers). These observations were interpreted as the result of an increase in the contribution of GBS in fine grains. (Craw et al., 2018) and (Fan et al., 2020) reported similar observations in uniaxially deformed Antarctic ice and synthetic ice respectively, and the reduction of COF intensity in grains with finer sizes was attributed to GBS. (Fan et al., 2020; Qi et al., 2019; Wang et al., 2024) all infer that strength of COF in experiments is a competition between grain boundary migration (strengthening COF) and "rotation" processes, where those rotation processes include lattice rotation related to dislocation creep and recover and grain boundary sliding: following broadly ideas outlined in (Alley, 1992).

The listed controls on GBS are not the most important. The prime drivers of whether GBS is important are likely to be grain size and stress; if you scale grain size sensitivity from experimental data (Goldsby and Kohlstedt, 1997, 2001; Qi and Goldsby, 2021) to natural grain sizes and stress (Kuiper et al., 2020a; Kuiper et al., 2020b) then a significant GBS contribution is predicted. Impurities, in small volume fractions, likely have a secondary effect through restricting grain growth (Fan et al., 2023; Qi et al., 2018), keeping grain sizes small.

Nucleation. We only know that nucleation must occur by analogy with experiments where the number of grains increase with strain (Fan et al., 2020) and in zones of localized deformation where grain size reduces (common in silicates and carbonates: few observations in ice). We know virtually nothing about nucleation: two possible mechanisms are proposed: sub grain rotation recrystallisation and bulging nucleation and inferring these processes are difficult (Craw et al., 2018; Fan et al., 2020; Urai et al., 1986). Spontaneous nucleation in random orientations has been suggested (Chauve et al., 2017; Falus et al., 2011) but the physics of this process remains unconvincing to me. The evidence from experiments is that grains with orientations where the shear stress is high on the basal plane will grow (Fan et al., 2020; Qi et al., 2017; Qi et al., 2019) during strain induced GBM at the expense of other grain orientations. The grains that grow are already established and their nucleation is irrelevant to this process. Identifying nucleated grains is very hard: we can't do it in a natural samples.

Lobate and highly irregular boundaries are the observation often used to infer that grain boundary migration has been an important process (Rollett et al., 2017; Urai et al., 1986). The micrographs you present do not have strongly curved/ irregular/ lobate boundaries. Compare them for example with boundaries of grains in micrographs of the NEEM core (Weikusat et al., 2017). I see no highly lobate boundaries that would suggest that strain induced grain boundary migration was a dominant process. The grain boundaries are slightly curved. This needs to be the basis for this discussion. Telling the past direction of grain boundary migration is fraught with difficulties (Jessell, 1986, 1987). You have not presented what the observations are that allow you to infer migration directions. I don't think these inferences are robust.

The photos of particles on grain boundaries are very nice. I'm not sure they help particularly in the analysis- they are expected. They might be better in a short paper focused on that topic.

A key set of conclusions relate processes that control microstructures and COF through the whole core, with emphasis on the difference of the bottom 20% from the upper 80%. I cannot find any microstructural descriptions or micrographs of the upper 80% in this or any other paper. Similarly, I can find no directly measured COFs (Fabric Analyser, Laue, EBSD) for the upper 80% that allows the inclusion of COF shapes in the discussion. The dielectric tensor data in (Saruya et al., 2022) reduces the proxy COF to eigenvectors and loses shape and symmetry information.

Section 5.4 is not very good. You cannot have dislocation creep without both dynamic recovery and dynamic recrystallisation mechanisms. At the high homologous temperatures throughout the ice core ($T_H$ >0.8) it is inconceivable to have no dynamic recrystallisation. The key factor is the relative contribution the sub processes that are all needed for dynamic recrystallisation to occur: sub-grain rotation recrystallisation (following from recovery and sub-grain rotation: see nomenclature in (Trimby et al., 1998)), bulge nucleation, and strain induced grain boundary migration. See interpretive sections of (Fan et al., 2020; Qi et al., 2017) for example.

**Temperature**
It is clear that temperature is an important parameter in controlling ice behaviour. Fig 5 needs temperature data from the hole and any figure covering the whole hole depth (e.g. fig 9) needs this as well. I see that there are measurements in (Motoyama et al., 2021) and broader modeling by (Obase et al., 2023).

Sorry out of time here!!!

Alley, R. B., 1992, Flow-law hypotheses for ice-sheet modeling: Journal of Glaciology, v. 38, no. 129, p. 245-256.

Azuma, N., Wang, Y., Mori, K., Narita, H., Hondoh, T., Shoji, H., and Watanabe, O., 1999, Textures and fabrics in the Dome F (Antarctica) ice core, *in* Jacka, T. H., ed., Annals of Glaciology, Vol 29, 1999, Volume 29, p. 163-168.

Bestmann, M., and Prior, D. J., 2003, Intragranular dynamic recrystallization in naturally deformed calcite marble: diffusion accommodated grain boundary sliding as a result of subgrain rotation recrystallization: Journal of Structural Geology, v. 25, no. 10, p. 1597-1613.

Bouchez, J. L., and Duval, P., 1982, The fabric of polycrystalline ice deformed in simple shear - experiments in torsion, natural deformation and geometrical interpretation: Textures and Microstructures, v. 5, no. 3, p. 171-190.

Budd, W. F., and Jacka, T. H., 1989, A review of ice rheology for ice-sheet modeling: Cold Regions Science and Technology, v. 16, no. 2, p. 107-144.

Chauve, T., Montagnat, M., Barou, F., Hidas, K., Tommasi, A., and Mainprice, D., 2017, Investigation of nucleation processes during dynamic recrystallization of ice using cryo-EBSD: Philosophical Transactions of the Royal Society a-Mathematical Physical and Engineering Sciences, v. 375, no. 2086.

Craw, L., Qi, C., Prior, D. J., Goldsby, D. L., and Kim, D., 2018, Mechanics and microstructure of deformed natural anisotropic ice: Journal of Structural Geology, v. 115, p. 152-166.

DFICPM:, Kawamura, K., Abe-Ouchi, A., Motoyama, H., Ageta, Y., Aoki, S., Azuma, N., Fujii, Y., Fujita, K., Fujita, S., Fukui, K., Furukawa, T., Furusaki, A., Goto-Azuma, K., Greve, R., Hirabayashi, M., Hondoh, T., Hori, A., Horikawa, S., Horiuchi, K., Igarashi, M., Iizuka, Y., Kameda, T., Kanda, H., Kohno, M., Kuramoto, T., Matsushi, Y., Miyahara, M., Miyake, T., Miyamoto, A., Nagashima, Y., Nakayama, Y., Nakazawa, T., Nakazawa, F., Nishio, F., Obinata, I., Ohgaito, R., Oka, A., Okuno, J. i., Okuyama, J., Oyabu, I., Parrenin, F., Pattyn, F., Saito, F., Saito, T., Saito, T., Sakurai, T., Sasa, K., Seddik, H., Shibata, Y., Shinbori, K., Suzuki, K., Suzuki, T., Takahashi, A., Takahashi, K., Takahashi, S., Takata, M., Tanaka, Y., Uemura, R., Watanabe, G., Watanabe, O., Yamasaki, T., Yokoyama, K., Yoshimori, M., and Yoshimoto, T., 2017, State dependence of climatic instability over the past 720,000 years from Antarctic ice cores and climate modeling: Science Advances, v. 3, no. 2, p. e1600446.

Disbrow-Monz, M. E., Hudleston, P. J., and Prior, D. J., 2024, Multimaxima crystallographic fabrics (CPO) in warm, coarse-grained ice: a case study: Journal of Structural Geology, v. In Press (SG-D-23-00145R2).

Falus, G., Tommasi, A., and Soustelle, V., 2011, The effect of dynamic recrystallization on olivine crystal preferred orientations in mantle xenoliths deformed under varied stress conditions: Journal of Structural Geology, v. 33, no. 11, p. 1528-1540.

Fan, S., Hager, T. F., Prior, D. J., Cross, A. J., Goldsby, D. L., Qi, C., Negrini, M., and Wheeler, J., 2020, Temperature and strain controls on ice deformation mechanisms: insights from the microstructures of samples deformed to progressively higher strains at −10, −20 and −30°C: The Cryosphere, v. 14, no. 11, p. 3875-3905.

Fan, S., Prior, D. J., Pooley, B., Bowman, H., Davidson, L., Wallis, D., Piazolo, S., Qi, C., Goldsby, D. L., and Hager, T. F., 2023, Grain growth of natural and synthetic ice at 0 degrees C: Cryosphere, v. 17, no. 8, p. 3443-3459.

Faria, S. H., Weikusat, I., and Azuma, N., 2014, The microstructure of polar ice. Part I: Highlights from ice core research: Journal Of Structural Geology, v. 61, p. 2-20.

Fossen, H., and Cavalcante, G. C. G., 2017, Shear zones - A review: Earth-Science Reviews, v. 171, p. 434-455.

Goldsby, D. L., and Kohlstedt, D. L., 1997, Grain boundary sliding in fine-grained Ice I: Scripta Materialia, v. 37, no. 9, p. 1399-1406.

-, 2001, Superplastic deformation of ice: Experimental observations: Journal of Geophysical Research-Solid Earth, v. 106, no. B6, p. 11017-11030.

Hubbard, B., and Malone, T., 2013, Optical-televiewer-based logging of the uppermost 630 m of the NEEM deep ice borehole, Greenland: Annals of Glaciology, v. 54, no. 64, p. 83-89.

Hudleston, P. J., 1977, Progressive Deformation and Development of Fabric Across Zones of Shear in Glacial Ice, in Saxena, S. K., Bhattacharji, S., Annersten, H., and Stephansson, O., eds., Energetics of Geological Processes: Hans Ramberg on his 60th birthday: Berlin, Heidelberg, Springer Berlin Heidelberg, p. 121-150.

-, 2015, Structures and fabrics in glacial ice: A review: Journal Of Structural Geology, v. 81, p. 1-27.

Hunter, N. J. R., Wilson, C. J. L., and Luzin, V., 2023, Crystallographic preferred orientation (CPO) patterns in uniaxially compressed deuterated ice: quantitative analysis of historical data: Journal of Glaciology, v. 69, no. 276, p. 737-748.

Jacka, T. H., and LI, J., 2000, Flow rates and crystal orientation fabrics in compression of polycrystalline ice at low temperatures and stresses: Physics of ice core records. Contrib. Inst. Low Temp. Sci. Hokkaido Univ., Ser. A,, p. 83-102.

Jacka, T. H., and Maccagnan, M., 1984, Ice crystallographic and strain rate changes with strain in compression and extension: Cold Regions Science and Technology, v. 8, no. 3, p. 269-286.

Jackson, M., and Kamb, B., 1997, The marginal shear stress of Ice Stream B, West Antarctica: Journal of Glaciology, v. 43, no. 145, p. 415-426.

Jennings, S. J. A., and Hambrey, M. J., 2021, Structures and Deformation in Glaciers and Ice Sheets: Reviews of Geophysics, v. 59, no. 3.

Jessell, M. W., 1986, Grain-Boundary Migration and Fabric Development in Experimentally Deformed Octachloropropane: Journal of Structural Geology, v. 8, no. 5, p. 527-542.

-, 1987, Grain-Boundary Migration Microstructures in a Naturally Deformed Quartzite: Journal of Structural Geology, v. 9, no. 8, p. 1007-1014.

Jiang, Z. T., Prior, D. J., and Wheeler, J., 2000, Albite crystallographic preferred orientation and grain misorientation distribution in a low-grade mylonite: implications for granular flow: Journal of Structural Geology, v. 22, no. 11-12, p. 1663-1674.

Journaux, B., Chauve, T., Montagnat, M., Tommasi, A., Barou, F., Mainprice, D., and Gest, L., 2019, Recrystallization processes, microstructure and crystallographic preferred orientation evolution in polycrystalline ice during high-temperature simple shear: The Cryosphere, v. 13, no. 5, p. 1495-1511.

Kamb, W. B., 1972, Experimental recrystallization of ice under stress, *in* Heard, H. C., Borg, I. Y., Carter, N. L., and Rayleigh, C. B., eds., Flow and Fracture of Rocks, American Geophysical Union, p. 211-242.

Karlsson, N. B., Binder, T., Eagles, G., Helm, V., Pattyn, F., Van Liefferinge, B., and Eisen, O., 2018, Glaciological characteristics in the Dome Fuji region and new assessment for "Oldest Ice": Cryosphere, v. 12, no. 7, p. 2413-2424.

Kruckenberg, S. C., Michels, Z. D., and Parsons, M. M., 2019, From intracrystalline distortion to plate motion: Unifying structural, kinematic, and textural analysis in heterogeneous shear zones through crystallographic orientation-dispersion methods: Geosphere, v. 15, no. 2, p. 357-381.

Kuiper, E. J. N., de Bresser, J. H. P., Drury, M. R., Eichler, J., Pennock, G. M., and Weikusat, I., 2020a, Using a composite flow law to model deformation in the NEEM deep ice core, Greenland – Part 2: The role of grain size and premelting on ice deformation at high homologous temperature: The Cryosphere, v. 14, no. 7, p. 2449-2467.

Kuiper, E. J. N., Weikusat, I., de Bresser, J. H. P., Jansen, D., Pennock, G. M., and Drury, M. R., 2020b, Using a composite flow law to model deformation in the NEEM deep ice core, Greenland – Part 1: The role of grain size and grain size distribution on deformation of the upper 2207 m: The Cryosphere, v. 14, no. 7, p. 2429-2448.

Li, J., Jacka, T. H., and Budd, W. F., 2000, Strong single-maximum crystal fabrics developed in ice undergoing shear with unconstrained normal deformation, *in* Hutter, K., ed., Annals of Glaciology, Vol 30, 2000, Volume 30, p. 88-92.

Michels, Z. D., Kruckenberg, S. C., Davis, J. R., and Tikoff, B., 2015, Determining vorticity axes from grain-scale dispersion of crystallographic orientations: Geology, v. 43, no. 9, p. 803-806.

Montagnat, M., Chauve, T., Barou, F., Tommasi, A., Beausir, B., and Frassengeas, C., 2015, Analysis of dynamic recrystallisation of ice from EBSD orientation mapping: Frontiers of Earth Science, v. 3, p. 13.

Monz, M. E., Hudleston, P. J., Prior, D. J., Michels, Z., Fan, S., Negrini, M., Langhorne, P. J., and Qi, C., 2021, Full crystallographic orientation (c and a axes) of warm, coarse-grained ice in a shear-dominated setting: a case study, Storglaciären, Sweden: The Cryosphere, v. 15, no. 1, p. 303-324.

Motoyama, H., Takahashi, A., Tanaka, Y., Shinbori, K., Miyahara, M., Yoshimoto, T., Fujii, Y., Furusaki, A., Azuma, N., Ozawa, Y., Kobayashi, A., and Yoshise, Y., 2021, Deep ice core drilling to a depth of 3035.22 m at Dome Fuji, Antarctica in 2001-07: Annals of Glaciology, v. 62, no. 85-86, p. 212-222.

Obase, T., Abe-Ouchi, A., Saito, F., Tsutaki, S., Fujita, S., Kawamura, K., and Motoyama, H., 2023, A one-dimensional temperature and age modeling study for selecting the drill site of the oldest ice core near Dome Fuji, Antarctica: Cryosphere, v. 17, no. 6, p. 2543-2562.

Qi, C., and Goldsby, D. L., 2021, An Experimental Investigation of the Effect of Grain Size on "Dislocation Creep" of Ice: Journal of Geophysical Research-Solid Earth, v. 126, no. 9.

Qi, C., Goldsby, D. L., and Prior, D. J., 2017, The down-stress transition from cluster to cone fabrics in experimentally deformed ice: Earth and Planetary Science Letters, v. 471, p. 136-147.

Qi, C., Prior, D. J., Craw, L., Fan, S., Llorens, M. G., Griera, A., Negrini, M., Bons, P. D., and Goldsby, D. L., 2019, Crystallographic preferred orientations of ice deformed in direct-shear experiments at low temperatures: The Cryosphere, v. 13, no. 1, p. 351-371.

Qi, C., Stern, L. A., Pathare, A., Durham, W. B., and Goldsby, D. L., 2018, Inhibition of Grain Boundary Sliding in Fine-Grained Ice by Intergranular Particles: Implications for Planetary Ice Masses: Geophysical Research Letters, v. 45, no. 23, p. 12757-12765.

Ramsay, J. G., 1980, Shear zone geometry - a review: Journal of Structural Geology, v. 2, no. 1-2, p. 83-99.

Rollett, A. D., Rohrer, G. S., and Humphreys, F. J., 2017, Recrystallization and Related Annealing Phenomena, Elsevier Science.

Saruya, T., Fujita, S., Iizuka, Y., Miyamoto, A., Ohno, H., Hori, A., Shigeyama, W., Hirabayashi, M., and Goto-Azuma, K., 2022, Development of crystal orientation fabric in the Dome Fuji ice core in East Antarctica: implications for the deformation regime in ice sheets: The Cryosphere, v. 16, no. 7, p. 2985-3003.

Thomas, R. E., Negrini, M., Prior, D. J., Mulvaney, R., Still, H., Bowman, M. H., Craw, L., Fan, S., Hubbard, B., Hulbe, C., Kim, D., and Lutz, F., 2021, Microstructure and Crystallographic Preferred Orientations of an Azimuthally Oriented Ice Core from a Lateral Shear Margin: Priestley Glacier, Antarctica: Frontiers in Earth Science, v. 9, no. 1084.

Treverrow, A., Warner, R. C., Budd, W. F., Jacka, T. H., and Roberts, J. L., 2015, Modelled stress distributions at the Dome Summit South borehole, Law Dome, East Antarctica: a comparison of anisotropic ice flow relations: Journal of Glaciology, v. 61, no. 229, p. 987-1004.

Trimby, P. W., Prior, D. J., and Wheeler, J., 1998, Grain boundary hierarchy development in a quartz mylonite: Journal of Structural Geology, v. 20, no. 7, p. 917-935.

Urai, J. L., Means, W. D., and Lister, G. S., 1986, Dynamic recrystallization of Minerals, *in* Hobbs, B. E., and Heard, H. C., eds., Mineral and Rock Deformation (Laboratory Studies), Volume 36, p. 161-200.

Vaughan, M. J., Prior, D. J., Jefferd, M., Brantut, N., Mitchell, T. M., and Seidemann, M., 2017, Insights into anisotropy development and weakening of ice from in situ P wave velocity monitoring during laboratory creep: Journal of Geophysical Research-Solid Earth, v. 122, no. 9, p. 7076-7089.

Wang, Q., Fan, S., Richards, D. H., Worthington, R., Prior, D. J., and Qi, C., 2024, Evolution of crystallographic preferred orientations of ice sheared to high strains by equal-channel angular pressing: EGUsphere, v. 2024, p. 1-34.

Wang, Z., Chung, A. L., Steinhage, D., Parrenin, F., Freitag, J., and Eisen, O., 2023, Mapping age and basal conditions of ice in the Dome Fuji region, Antarctica, by combining radar internal layer stratigraphy and flow modeling: Cryosphere, v. 17, no. 10, p. 4297-4314.

Warren, J. M., and Hirth, G., 2006, Grain size sensitive deformation mechanisms in naturally deformed peridotites: Earth And Planetary Science Letters, v. 248, no. 1-2, p. 438-450.

Weikusat, I., Jansen, D., Binder, T., Eichler, J., Faria, S. H., Wilhelms, F., Kipfstuhl, S., Sheldon, S., Miller, H., Dahl-Jensen, D., and Kleiner, T., 2017, Physical analysis of an Antarctic ice core-towards an integration of micro- and macrodynamics of polar ice: Philosophical Transactions of the Royal Society a-Mathematical Physical and Engineering Sciences, v. 375, no. 2086.

---

## Author Comment (AC1)

[Figure]

**Figure 8.** COF data from the DF core and the EDC core are compared using common age scales. Here, we applied the DF2021 age scale (Oyabu et al., 2022) for ages younger than 216 thousand years BP and the AICC2012 age scale (Bazin et al., 2013) for older ages. Light-gray shading indicates the interglacial periods. (a) $\Delta\varepsilon$ presented as raw data (dots) and as mean values for each 0.5-m segment (black markers for DF) or for each thin section (red marker for EDC). The right axis provides a scale for normalized eigenvalues. (b) SD for $\Delta\varepsilon$ in the DF core. (c) Detrended $\Delta\varepsilon$ values in the DF core. (d) Oxygen isotope ratios ($\delta^{18}O$) in the DF ice core. (e) Grain size in the DF cores (Azuma et al., 2000 and this study) and the EDC core (Durand et al., 2009). (f) Concentrations of $Cl^-$, $SO_4^{2-}$, and $Ca^{2+}$ ions (Goto-Azuma et al., 2019). (g) Concentration of dust particles (DFICPM, 2017).

---

## Author Response (AR1)

Dear Reviewers and Editors,

We sincerely appreciate your tremendous efforts in handling and reviewing our manuscript. In addition to our responses addressing the comments from RC1 and RC2, we would like to offer the following statement:

This paper has generated a wealth of new insights across the deep and full layers of the Antarctic ice sheet by applying several innovative analytical methods, which have rarely been used on such high-quality and unique deep ice cores. During the revision process, we made bold attempts to simplify the content, but this resulted in a version that conveyed only a rough outline of our substantial research findings.

We believe that upon reviewing the revised abstract, introduction, and the more structured conclusion, you will understand that this paper cannot be confined to a single or limited narrative, as is often the case in other work. This paper has become particularly broad and significant in scope, even within our research careers. Compared to the many papers we have written, we are confident that the findings presented here are of great importance.

Over-simplifying this paper would compromise its scientific message. Writing numerous papers with simple, single-narrative structures focusing on "a story" would be a disservice to the scientific community. It is essential to present the overall picture comprehensively, rather than fragmenting it into piecemeal communications.

We sincerely hope you understand that the substantial volume of this paper is necessary to convey the facts and discussions thoroughly and carefully to the readers.

With many thanks and kind regards,
Tomotaka Saruya, Atsushi Miyamoto, and Shuji Fujita
on behalf of all authors

**How the authors addressed the comments by RC1 in their responses to the revised manuscript**

Tomotaka Saruya[1], Atsushi Miyamoto[2], Shuji Fujita[1,3], Kumiko Goto-Azuma[1,3], Motohiro Hirabayashi[1], Akira Hori[4], Makoto Igarashi[1], Yoshinori Iizuka[5], Takao Kameda[4], Hiroshi Ohno[4], Wataru Shigeyama[3*], Shun Tsutaki[1,3]

[1] National Institute of Polar Research, Tokyo 190-8518, Japan
[2] Institute for the Advancement of Graduate Education, Hokkaido University, Sapporo 060-0817, Japan
[3] Polar Science Program, Graduate Institute for Advanced Studies, SOKENDAI, Tokyo 190-8518, Japan
[4] Kitami Institute of Technology, Kitami 090-8507, Japan
[5] Institute of Low Temperature Science, Hokkaido University, Sapporo 060-0819, Japan

[*]Currently at: JEOL Ltd., Tokyo, Japan

**Correspondence**: Tomotaka Saruya (saruya.tomotaka@nipr.ac.jp), Atsushi Miyamoto (miyamoto@high.hokudai.ac.jp), Shuji Fujita (sfujita@nipr.ac.jp)
* * *
**Explanation of the text colors:**

➤ *Black Italic text: Comments from Reviewer 1 dated February 13, 2024*

➤ Blue text: Authors' reply to RC1 on April 10, 2024

➤ Brown text: Explanations of how the authors addressed RC1's comments in their revised manuscript, submitted on August 17, 2024
* * *
In this document, we explain how we addressed the comments provided by RC1 on the revised manuscript. We will use the identifier [Rev1_#] for our explanations.

General comments

In the revised manuscript, we significantly revised the discussions based on the reviewers' comments. Some detailed discussions on specific topics, such as ions and dust particles, were removed or condensed to reduce the overall volume. Following Reviewer 2's comments, we investigated and expanded the discussions on *a*-axis fabric. This led to the discovery of a new and important aspect of *a*-axis organization and grain shape information (aspect ratio). Consequently, we needed to discuss the possible causes of *a*-axis organization within the context of dynamic recrystallization. Possibility of the crystal twinning needed to be speculated. In the discussion sections, we focused more on the contrast between impure and less-impure layers. Microstructures are now positioned as supporting information for the discussion on crystal orientation fabric, layered structures, and their relationships. We reorganized sections, combining previous sections 5.3.1 and 5.3.2 with the former section 4.5. Additionally, we redesigned the figures throughout the paper. Including items above, the major modifications are as follows:

➤ The introduction was expanded and structured, with additional references on the background of the subjects.

➤ Descriptions of microstructures were reduced and merged into Section 4.5.

➤ Microstructural images in Figures 6 and 7 were redesigned.

➤ Detailed descriptions of chemical impurities were significantly reduced.

➤ The order of discussions within the Discussion section was rearranged (1st: deep sections – 2nd: entire).

➤ Discussions on a-axis distribution were added as Section 5.1.5.

➤ The description of zoning (previously Section 5.4) was removed.

➤ Acronyms were significantly reduced.

[Authors' reply to RC1 on April 10, 2024]

We thank the reviewer for the careful and critical review of the manuscript, as well as for the thoughtful comments aimed at improving it. We also extend our gratitude to all individuals involved (reviewers and editors) for dedicating their precious time to this endeavor. Below, we outline our responses to each of the reviewer's comments, denoted in blue text, while the original comments are presented in black. For ease of reference, we have assigned ID numbers to each of the reviewer's comments and to each of our replies. To begin, we offer several key statements as an introduction to our responses.

**Key statements**

**[A0_1] Regarding the reviewer's suggestion on reorganization and convergence into a coherent story**

In both the global statements and detailed points, the reviewer has frequently commented that the discussions need to be re-organized to have a clearer focus on selected results and observations that contribute something new to the "story." The reviewer suggests that many results and ideas should converge into a "story" with more precise comparisons with existing work.

We appreciate the reviewer's feedback and agree to attempt a reorganization to better emphasize some of the results. However, we also express a strong concern that focusing overly on a single narrative, or splitting the paper into multiple narratives, might not be the best approach for the current work. This is due to its numerous results and insights, as listed in the conclusion section of the preprint. We have strong concerns that this approach might lead to the omission of significant findings, given the novelty and breadth of our research, which relies on innovative methods or discoveries in our present work.

A possible middle ground could be to structure the discussion around a few central themes or questions that the research addresses, organizing related findings and insights under these themes. This approach can help achieve both clarity and comprehensiveness, allowing us to present a coherent narrative without omitting significant results. It also provides a framework that can help readers understand how different parts of the research contribute to the overall story.

For this possible middle ground, in the revision, we will focus on at least the following topics:
(i) Crystal anisotropy and ice sheet dynamics
(ii) Advantages of ice cores from dome regions in the central plateau area of ice sheets
(iii) Exploring deep ice rheology and microstructure
(iv) Advancing ice sheet dynamics: The development and role of crystal orientation fabric
(v) Contrasting mechanisms of ice deformation across depths: revisiting shear impact and COF variations in ice cores such as EDC and DF.
Readers will discover that information itemized from (i) to (iv) will be very informative to better understand the important topic (v).

[Rev0_1] In the revised version, we focused on the items listed above. The introduction section was structured focusing on these items. Readers can better understand what background are and what are already known, including many earlier papers. In addition, we put a focus on presence/absence of simple shear. Based on the suggestion by the reviewer 1, we focused on it in the abstract, introduction and conclusion. Another important focus was that we summarized our improved view on formation processes of COF, in the abstract and in discussions.

**[A0_2] The position of this work in terms of ice sheet dynamics: Advantages of ice cores from dome regions in the central plateau area of the ice sheets**

The ice in the dome summit regions of central plateau areas on ice sheets offers an opportunity for studying ice deformation processes. In these regions, conditions are relatively simple because they are presumed to be least affected by simple shear deformation, typically caused by surface slope and gravity. Due to this simplicity, dome regions in the central area of the ice sheets

are ideal for investigating how processes related to deformational progress, such as dislocation creep and recrystallization, develop in a relatively simplified stress-strain environment and under moderate temperature gradients. Among the drilling sites in Antarctica, it is presumed that, so far, only the DF and EDC sites meet these conditions. In fact, in the case of inland domes, ice deforms mainly through compression along the vertical axis. This has been confirmed by the COF measured along cores drilled at these sites, which exhibits a single-pole distribution of the COF. Similarly, this was confirmed in the Greenland ice sheet with the GRIP core (Thorsteinsson et al., 1997). It is noted that there are also local domes at the edge of the ice sheet plateau, for example, Talos Dome (Urbini et al., 2008). Such a dome is characterized by a lower elevation (of about 2300 m), a location facing the coast, an annual accumulation rate much larger than those of the central plateau (by 3 to 4 times), and suggested migration at least during the last few centuries. Such a dome has higher instability and is suitable for investigating the impact of more shear, in contrast to the domes in the central plateau area. The COF evolution for the Talos Dome core was presented by Montagnat et al. (2012) as mentioned by the reviewer.

[Rev 0_2] These points were explained in the new section 1.2.

**[A0_3] On the claims of the presence of simple shear in Termination II within the ice sheet, as suggested by Durand et al.**

In the present study, including that by Saruya et al. (2022b), we did not identify any anomalous steps of c-axes clustering with increasing depth (as illustrated in Figures 7 and 8 in the preprint and Figure A0_3 in this document below). Instead, the COF exhibits fluctuations that are already apparent at a few hundred meters depth at Termination I. Thus, we found no indication attributable to the presence of simple shear. This argument differs, at least in part, from the scientific claims made by Durand et al. (2007; 2009), who suggested that the impact of shear was significant in explaining the anomalous strengthening of the c-axes cluster, especially at Termination II (refer to Figure 8). In contrast to shear, Saruya et al. (2022b) attribute small depressions of $\Delta\varepsilon$ values at MIS5 to the very low concentration of resolved $Cl^-$ ions (which can substitute for the location of $H_2O$) in the crystal lattice, indicating that ice is harder. This explanation might apply to the EDC case because, at EDC, most $Cl^-$ ions are lost to the atmosphere from snow compared to DF during the Holocene (i.e., the interglacial period), while at Dome Fuji, they are preserved as NaCl and in solid solution (Oyabu et al., 2020). These conditions suggest that ice becomes harder, preventing the COF cluster from strengthening at MIS5 as well. In other words, this could explain the significant depression of COF at MIS5 and a notable jump in the COF cluster in adjacent older ice at EDC. Additionally, further complexity arises because, at EDC, the surface mass balance (SMB) contrast between glacial and interglacial periods is approximately 20% larger than that of DF (Fujita et al., 2015; Parrenin et al., 2016), which will dilute $Cl^-$ ions more at EDC and also complicate ice flow models.

[Figure]

**Figure A0_3: Modification of the Figure 8 in the preprint. COF data from the DF core and the EDC core are compared using a common age scale of the DF2021 age scale. (a) $\Delta\varepsilon$ presented as mean values for each 0.5-m segment (black markers for DF) or for each thin section (magenta marker for EDC). The right axis provides a scale for normalized eigenvalues. (b) Oxygen isotope ratios ($\delta18O$) in the DF ice**

core. (c) Concentrations of Cl⁻, SO₄²⁻, and Ca²⁺ ions (Goto-Azuma et al., 2019). (d) Concentration of dust particles (DFICPM, 2017). For ice at EDC at MIS5, the normalized eigenvalues are smaller than those at DF.

[Rev0_3] Based on our study, we suggest that the claims for presence of simple shear in the termination II within the ice core, given by Durand and others, is not supported by our data. This point is clearly written in the abstract, section 5.2.2, and conclusion. This is one of the important stories that the reviewer requested us to provide, to which we addressed.

**[A0_4] On the roles of different methods in investigating COF**

We utilized the thick-section-based Dielectric Tensor Method (DTM), as well as the thin-section-based methods of Laue X-ray diffraction and the automatic ice fabric analyzer. The roles and characteristics of each method were presented in Saruya et al. (2021, 2022). Using DTM, COF data are provided as eigenvalues with a high sampling frequency, high spatial resolution, and continuity, thereby offering statistical significance. However, this method does not yield data on the crystal axes of individual grains. Additionally, if a cluster of c-axes is significantly inclined from the vertical, deriving correct eigenvalues is challenging without knowledge of both the inclination angle of the c-axis cluster and its horizontal orientation. Moreover, the Laue X-ray diffraction method enables the clarification of detailed information about not only the c-axes but also the a-axes of each crystal grain. Furthermore, the automatic fabric analyzer, model G50, originally manufactured by Russel–Head Instruments, proves very useful for demonstrating how crystal orientation differs from one grain to another. The judicious use of these methods provides unprecedentedly rich information set on COF along ice cores.

In the revised manuscript, we will explain the role and complementarity of each method.

[Rev0_4] We addressed it in the abstract, introduction and conclusions as one of the most important messages in this paper.

**Reply point-by-point to each of reviewer's comment**

**General points**

[R1_1] *This paper provides analyses of physical properties (grain size and crystallographic orientations) measured along the deepest part of the Dome Fuji ice core, between 2400 and 3035 m depth. Measurements were done using the dielectric tensor method (DTM) on thick sections of ice, the Laue X-ray diffraction method and the automatic ice fabric analyser on thin sections. Based on these measurements, the authors provide an analysis of the deformation and recrystallization processes likely to take place in the deepest part of the ice core. They also observe and somehow quantify the impact of the shear component on the flow of ice below Dome Fuji.*

*Global comments:*
[R1_2] *This paper presents some valuable high resolution data of crystal orientation fabric (COF) in the deep part of the core where deformation heterogeneities are supposed to occur that could disrupt the dating signal and the radar echo sounding in the area.*
[R1_3] *They also provide some comparisons between three different means of measuring the COF that can be of interest.*

[A1_1, A1_2, A1_3] Thank you for the comments.

[R4] *Nevertheless, I did not clearly understand the necessity to have the three different types of measurement, especially for Laue X-ray diffraction and fabric analyser that are strongly redundant (except that X-ray diffraction provide the a-axes orientations that are known to be isotropic, so yes, it is interesting to double check but it is not used in the analyses).*

[A1_4] Please refer to [A0_4].

[Rev1_4] Please refer to [Rev 0_4].

[R1_5] *The DTM method provides a higher resolution than classical diffraction or optical measurements nevertheless the method still requires making sections (thick ones), and, in order to be statistically representative, it must integrate a minimum number of grains, therefore it may be viewed as a moving average of the COF obtained from the other means from thin section?*

[A1_5] We plan to address it in the revised manuscript. As for the statistical significance, depending on number of crystal grains in the beam, statistical significance is different. In some cases, a measurement at a point contains hundreds of grains, in another case, a beam contains smaller numbers (10 to $10^2$). Then, moving averaging along the core is useful to gain more significance. A table below will be added to revised manuscript.

**Table A5: Grain numbers included in a Gaussian beam or in a thin section**

| Measuring size and grain area | Dimension (mm) | 0.1 cm$^2$ | 0.3 cm$^2$ | 0.5 cm$^2$ | 1 cm$^2$ | 2 cm$^2$ | 4 cm$^2$ |
|---|---|---|---|---|---|---|---|
| Thick-section-based method | | | | | | | |
| DTM No.1 | ~16φ × ~40 | 353 | 68 | 32 | 11 | 4 | 1 |
| DTM No.2 | ~38φ× ~70 | 3490 | 672 | 312 | 110 | 39 | 14 |
| Thin-section-based methods | | | | | | | |
| Laue X-ray method | 100 × 45 | | | | | | |
| Optical method of microscopy and G50 | 90 × 50 | 450 | 150 | 90 | 45 | 23 | 12 |

[Rev1_5] This table was inserted as Table A3 in Appendix A1.

[R1_6] *Overall, the part that analyses and discusses the results should be re-organised and strengthen by some more precise references to previous works.*

[A1_6] Re-organization and more referencing will be considered one by one addressing each of specific comments.

[Rev1_6] We reorganized through the revised manuscript (particularly Introduction and Discussion Sections) and added references.

[R1_7] *This part appears as a "brainstorming", with many results and ideas mentioned, but they need to be converged into a "story" that shows what comes out of the results, what it brings to the existing story about ice core analyses (with more precise comparisons with existing work), and/or what new story it tells (although I didn't see so many new results or observations in this work).*

[A1_7] Thank you for the comment. Please refer to [A0_1] above.

[Rev1_7] Please refer to [Rev0_1].

[R1_8] *In particular, dynamic recrystallization along ice cores has been studied for a long time, both in terms of basis mechanisms that come into play and in terms of impact on the COF evolution. I think that COF evolution along Dome F ice core should be*

*analysed regarding this existing frame. I provide some references in the detailed comments below but many other exist that the authors can refer to (since most of the references I provide are from my team… as it is more straightforward for me).*

[A1_8] We agree that dynamic recrystallization has been investigated widely, as reviewed in papers or textbooks (e.g. Faria et al., 2014a, 2014b; Humphreys & Hatherly, 2004; Poirier, 1985) or individual papers (e.g., De La Chapelle et al., 1998; Kipfstuhl et al., 2009; Montagnat et al., 2012; 2014; 2015 Stoll et al., 2021a, b; Weikusat et al., 2009). Exploring deep ice rheology and crystal properties, dynamic recrystallization is one of main focuses in this paper.
In the revised manuscript, we will add references widely including review paper, individual paper, and textbook.

[Rev1_8] This point was addressed in the second paragraph of Section 1.4, citing most of the papers listed above.

[R1_9] *Some interpretations are not rigorous enough and should be strengthen by some physical concepts or adapted references. This is particularly the case when are mentioned grain boundary sliding and "microshear". Grain boundary sliding is a specific mechanism that occurs under specific circumstances that are clearly not encountered here (small grains, high level of strain, accommodating mechanisms at grain boundary, etc.). The only observation of a few grain boundaries with specific shapes is not enough to support grain boundary sliding (and in 2D!). See for instance the works done in the metallurgy community (e.g. Doquet & Barkia, 2016, Mechanics of Material, Linne et al., 2020, Int. J. Plasticity). I think this is not positive for the ice community to keep on mentioning GBS as an important mechanism although there exist so few evidence of it along ice cores. Especially since natural ice is characterized by very large grains, with very efficient accommodation processes that are grain boundary migration and dynamic recrystallization, there is no requirement of GBS to explain ice deformation along ice cores (even if it might take place in the specific conditions of the Goldsby & Kohlstedt, 1997 experiments on very fine-grained ice).*

[A1_9] As indicated by the reviewer, there is no significant evidence to suggest that microscopic GBS via microshear is occurring. We understand that the occurrence of GBS under the conditions present in ice sheet ice (e.g., large grain size) is far removed from common knowledge in metallurgy. Even if the GBS were to occur, it would likely be limited to special areas where the conditions are met. In the revised manuscript, we will emphasize that the GBS via microshear is one of the potential mechanisms that can explain the observed microstructure (brick-wall pattern) in the impure layers. Since the microscopic GBS is not the main focus of the present paper, we will provide a summary of this mechanism. It is worth mentioning that the brick-wall patterns were observed exclusively in impure layers, and microscopic GBS via microshear is a possible explanatory mechanism. Additionally, we will include quantitative discussions on grain shape (e.g., aspect ratio of grains). In the revised manuscript, we plan to revise Section 5.3.1 as follows:

**5.3.1 Microstructures in the impure layer: brick-wall pattern and steady-state grain size**

In this section, we discuss the $\Delta\varepsilon$ value variations, focusing on microstructures in the impure layer and dynamic recrystallization process. From microstructural observations, we identified interesting features in the impure layers (Figure 6): the crystal grains elongate, and their major axes incline away from horizontal directions. These features were not observed in surrounding less-impure (interglacial) layers. The aspect ratio of the short and long axis of a fitted ellipse in the impure layers is 1.9–2.0 (three thin sections), while that in less-impure layers is 1.5–1.7 (thirteen thin sections). Detailed microstructures are shown in Figure 12a. The same feature, known as the "brick-wall pattern" has been observed at the high-impurity ice layers of the EDML ice core in Antarctica (see Figure 1a for location) (Faria, 2009; Weikusat et al., 2017) and the NEEM ice core in Greenland (Kuiper et al., 2020), but it has not been reported in the EDC ice core. Weikusatl et al. (2017) mentioned that shear deformation is responsible for the brick-wall pattern in the EDML ice core. However, observations of the brick-wall pattern in ice cores are few and the mechanism is not clear. Faria et al. (2009) proposed microscopic grain boundary sliding (GBS) via microshear as a deformation mechanism for making the brick-wall pattern. Their idea was based on analogue experiments by Bons and Jessel (1999) using octachloropropane. Llorens et al. (2016) also discussed it based on the numerical simulation. According to Faria et al., (2009), a combination of high impurity content, temperature, and moderate stress causes microscopic GBS. These conditions could be achieved in the bottom part

of the ice sheets (Faria et al. 2009). The formation of microshear bands creates new, flat GBs parallel to the shear plane, resulting in the brick-wall pattern. It is no clear that GBS occurs in ice-sheet ice, but this mechanism can explain the observations of microstructures. Regarding the influence of GBS on the COF development, Faria et al. (2009) suggested that GBS has no severe impact on COF. In our study, we observed significant variations in $\Delta\varepsilon$ values within the impure layers. Furthermore, the $\Delta\varepsilon$ value has an inverse correlation with the dust concentration within the impure layer. Assuming COF remains unchanged by GBS, the variations in COF observed in the impure layers at greater depths would have originated at shallower depths before the onset of GBS. We observe an inverse correlation between the $\Delta\varepsilon$ value and dust concentration even at depths shallower than 2400 m.

The brick-wall pattern appears to be less pronounced in deeper sections as shown in Figure 6e (sample from a depth 2909 m). Despite the high concentration of dust particles (although not as high as 2648 and 2759 m), there are no evident brick-wall patterns. Due to high temperature close to melting point, the pattern could be lost by dynamic recovery and recrystallization. The brick-wall pattern will change with temperature and impurity concentration. Further investigations are required to clarify the mechanism of brick-wall pattern formation and development, and relation with temperature and impurity concentration. Although microscopic GBS can explain the microstructures in impure layers, it is not possible to determine from our observations whether microscopic GBS via microshear is really occurring or not. Further detailed observations and experimental verification are needed in future. Another feature in the impure layer is approximately constant (or small fluctuation) grain size (Figure 9d). In the DF ice core, a constant grain size is observed between depths of 2640 and 2675 m, where the concentration of dust particles is extremely high. This constant grain size in ice core is known as "steady state grain size" (e.g., Steinbach et al., 2017). It is believed that steady state grain size is achieved when normal grain growth counteracts rotation recrystallization regardless of the initial grain size (Jacka and Li, 1994). We believe that the steady grain sizes in the impure layers established themselves after these layers had reached deeper depths. When the impure layers were at shallower depths in the past, grain size would have been inversely correlated with dust concentration.

[Rev1_9] The sentences above were provided as section 4.5.2, with a more concise manner. Previous Section 5.3 (Microstructures at the deeper part of the DF ice core) was merged into Section 4.5. Microstructural observations are located as supporting information for the discussions of COF and layered structures. So, we reduced the reference to microstructures.

[R1_10] *Many repetitions exist all along section 5. Maybe the authors should be more focused on the main message they want to provide, in order to avoid dispersion in the interpretations and some lack of rigour as just mentioned, but also to enable more focused and deeper comparisons with existing works.*

[A1_10] Thank you for the comments. Please refer to [A0_1] above. This paper, as a result of applying innovative methods, has reached a wide range of conclusions, and in the process of its discussion, it inevitably has to mention the same thing occasionally, even within different sections, in an introductory manner. However, of course, the authors will attempt to improve it.

[Rev1_10] By removing a few sections, for example, previous section "5.4 Zoning of COF development in the DF ice core", we attempted to improve on this point.

[R1_11] *In general, to make reading easier, please avoid acronyms in section titles, abstract and conclusion.*

[A1_11] Our views are as follows: For all acronyms, we have provided clear explanations when each abbreviation first appears in the manuscript. This approach is intended to avoid the repetition of lengthy expressions in the abstract. Examples include using "COF" for crystal orientation fabrics, "UP80%" for the uppermost approximately 80% thickness zone, and "LO20%" for the lowermost approximately 20% thickness zone. Without employing these acronyms, the readability of the abstract would suffer due to the repetition of lengthy terms. Regarding for the section titles in the paper, we used "COF" 9 times. This strategy allows readers to intuitively understand the abbreviation each time it appears, rather than navigating through lengthy expressions repeatedly. We ask readers to recognize the clear benefits of this approach.

[Rev1_11] We reduced number of acronyms dramatically.

**Specific comments along the text**

[R1_12] *l. 24-25 (abstract): dynamic recrystallization is characterized by nucleation of new grains and grain boundary migration. The main basis processes have been described for a long time (see e.g. Poirier 1985, Humphreys and Haterly 2004) and please refer to the existing frame. Therefore this sentence is not clear "dynamic recrystallization related to this emergence".*

[A1_12] We will rephrase it to "migration recrystallization".

[Rev1_12] We rewrote this part as a sentence "Impurity-rich layers maintained stronger clustering, while impurity-poor layers showed relaxation, likely due to the emergence of new grains with $c$-axis orientations offset from the existing cluster and migration recrystallization." As for the citation, the suggested papers were referred to, in the main text.

[R1_13] *l. 43: what is meant by "dynamical analysis"?*

[A1_13] We will rephrase it to "analysis of ice cores in terms of ice dynamics".

[Rev1_13] We rephrased it at the last sentence in Section 1.1.

[R1_14] *l. 79: "remains" to be done?*

[A1_14] We will add "to be done".

[Rev1_14] We addressed it.

*Part 3.1:*
[R15] *What about measurement of fabrics that depart from cluster-shape with the DTM technique? We have observed some, for instance along the NEEM ice core (Montagnat et al. 2014, The Cryosphere), likely resulting from a tension component of the stress field.*

[A1_15] Thanks for the comment. DTM is useful to detect $\Delta\varepsilon$ values in the girdle type ice fabrics. We can refer to basic principle in Figure C1 in Appendix C, as girdle distribution of a-axes in the figure. This will be mentioned in the revised version.

[Rev1_15] We added it in the middle of Section 3.1.2.

*Part 3.2 and 3.3:*
[R1_16] *Why is it necessary to perform all these different measurements? How complementary are they?*

[A1_16] Please refer to [A0_4] above.

[Rev1_16] As answered in [Rev0_4], they are crucial.

[R1_17] *Do you have enough grains in the X-ray thin sections?*

[A1_17] Please refer to number of grains extracted from each thin section in Figure 4. "Enough grains or not" will depend on requirement for statistical analysis. Also, please refer to [A5] above.

[Rev1_17] As answered above.

[R1_18] *Why don't you measure the grain size from the automatic fabric analyser?*

[A1_18] The grain size data were established immediately after the drilling based on observations of the ice core surfaces.

[Rev1_18] As answered above.

*Part 4:*
[R1_19] *Figure 4 is of very low quality. Please improve it in the final version.*

[A1_19] In revision, better quality file will be provided.

[Rev1_19] In revision, a better quality file was provided.

[R1_20] *Content of appendix C is very often referred to and is highly necessary to understand the data. Maybe it should be put back in the main text.*

[A1_20] Though there is this possibility, this is technical detail for correction. We will keep it in Appendix because more focus on science is the priority in the main text.

[Rev1_20] We kept it in the Appendix.

[R1_21] *Part 4.4: the quality of the writing should be improved in this part.*

[A1_21] We will attempt to improve.

[Rev1_21] We improved this section by removing acronyms.

[R1_22] *What does "an axis orthogonal to the shear plane and the IACC will deviate" means?*

[A1_22] It means that misorientation angle between an axis orthogonal to the shear plane and the IACC will appear and increase. Also, please refer to the explanations in Figure 9. In principle, it means the deviation between the IACC and the IAVL.

[Rev1_22] As answered above.

[R1_23] *Part 4.5: to my point of view, the microstructure features analyses are based on too few grains, and on 2D observations of 3D mechanisms, and must therefore be taken with care unless they are statistically significant (for instance if the shape of grains have been characterized over a large enough number of grains and sections, etc.).*

[A1_23] We consider the issue of statistical significance to be unavoidable in thin-section measurements for COF and microstructures. In the current manuscript, we present examples of noteworthy microstructures. Between depths of 2400 and 3000 m, we observed 15 thin sections using optical microscopy (and G50) and analyzed 43 polarized images (from Laue X-ray

measurements). We identified brick-wall patterns in all impure layers, except in very deep sections. In the revised manuscript, we will include quantitative discussions on the grain shape in both impure and less-impure layers, such as the aspect ratio of grains. Furthermore, we will add the following sentence:

*In the present study, we provide examples of noteworthy microstructures. Further investigations are necessary to thoroughly discuss the detailed evolution of these microstructures. Such analyses could include examining the misorientation angles of nucleated grains or sGBs, investigating the depth variation in the frequency and contribution of dynamic recrystallization, and exploring the mechanisms behind the brick-wall pattern observed in impure layers.*

[Rev1_23] We used the sentences above in Section 4.5.1.

[R1_24] *Moreover, extraction of the ice core leads to stress relaxations, especially around bubbles or clathrates in the deeper part of the core, that induce a lot of dislocations substructures (like subgrains). Care must be therefore taken when analyzing subgrain-scale features from these samples.*

[A1_24] Thank you for the general caution regarding the handling of ice cores. We agree with this point. Care must be taken to minimize post-drilling relaxation, or, should it occur, we need to detect it. For the Dome Fuji ice core, the cores have been preserved at a very cold temperature of -50°C at Hokkaido University, and later at the same temperature at NIPR. Under these cold conditions, we attempt to suppress (or slow down) the occurrence of relaxation phenomena. What we can do is maintain the ice under very cold conditions. To our knowledge, this is the best practice for handling ice cores in terms of temperature management within our community.

[Rev1_24] As answered above.

[R1_25] *When recrystallization is dominated by grain boundary migration (such as in the bottom of the GRIP core, de La Chapelle et al. 1998 (J. Geophys. Res) or along the NEEM core, Montagnat et al. 2014, for instance), the grain boundaries are highly serrated, and it can be observed on a large number of grains, therefore with a statistical value that overcome the 2D sectioning effect, or the impact of stress relaxation.*

[A1_25] In the DF core, we did not find such serrated features. Perhaps it is related to difference in conditions such as ice age and/or stress strain conditions.

[Rev1_25] As answered above.

*Part 5:*

[R1_26] *As mentioned in the global statements, the discussion would need being re-organized, with a clearer focus on the selected results and observations that bring something new to the "story".*

[A1_26] Please refer to [A0_1].

[Rev1_26] We reorganized through the revised manuscript. Please also refer to [Rev0_1]

[R1_27] *Overall, the authors should bring on numerous existing observations and analyses, especially for dynamic recrystallization and its impact on COF and microstructures, and its interactions with impurity contents. Most analyses and interpretations remain vague and not precise enough, some are not far from being wrong. See below for details.*

[A1_27] We will provide our reply point by point below.

[Rev1_27] We modified the following answers.

[R1_28_1] *Part 5.1.1: a highly concentrated clustered COF is related either to deformation by compression, with no dynamic recrystallization (since dynamic recrystallization opens the texture close to 45° from the compression direction, see e.g. work of Jacka's team in the 80's and 90's or more recently Montagnat et al. 2009 PICR2, Montagnat et al. 2015, Frontiers in Earth Sciences), or to simple shear.*

[A1_28_1] We agree that a highly-clustered COF is related either to deformation by compression, with no dynamic recrystallization, or to simple shear. We also agree that this concept was already established in late 80's. Related papers at this stage include, Azuma & Higashi, 1985; Budd & Jacka, 1989; Jacka & Budd, 1989. Additionally, Alley, 1988 compiled ideas at that time and gave a synthesis. There were many other papers related to this subject at that time and later time. The reviewer suggested us two papers, one entitled as "Recrystallization Processes in Granular Ice" and another "Analysis of Dynamic Recrystallization of Ice from EBSD Orientation Mapping". Our view is that these papers on laboratory measurements are surely related to this subject, but perhaps slightly far to use as key reference papers here.

[Rev1_28_1] Reference papers listed in the first three lines above were cited in Section 1.2 in the revised manuscript.

[R1_28_2] *In Montagnat et al. 2012 (Talos Dome ice core, EPSL), we have shown that compression alone could not explain a strong COF of the type the authors are measuring here, but that some simple shear is necessary. It seems to be the case here too.*

[A1_28_2] Please refer to [A0_2] and [A0_1]. It is one of points which we will develop explicitly in the revised version.

[Rev1_28_2] Talos Dome ice core was referred to in the end of section 1.3 explicitly.

[R1_29] *Simple shear could be assisted by dynamic recrystallization (dynamic recrystallization) but it is less straightforward to distinguish since dynamic recrystallization strengthens simple shear textures (see Bouchez and Duval 1982, in Textures and Microstructures, or Journaux et al. 2019, The Cryosphere).*

[A1_29] In the suggested papers of laboratory experiments under temperatures near the melting point, strain rate is larger than those in the Antarctic ice sheet by about $10^6$ times or more. To discuss analogy with dome ice cores, obviously extensive discussions (even papers) are necessary, which is beyond the scope of the present paper.

[Rev1_29] In Section 1.2 of the revised paper, we expressed the issue.
*These laboratory-based studies are characterized by experimental setting of strain rate by more than several orders of magnitude, and under temperature close to melting point, typically between -20 ℃ and 0 ℃. Therefore, these laboratory-based knowledge can be a valuable reference mainly for such conditions. To better understand rheology of polar ice sheets, analyses of ice texture sampled from polar ice sheets play essential roles.*

[R1_30] *L. 287: dynamic recrystallization is not a deformation process! This is an accommodation process that reduces the local strain and stress heterogeneities and facilitated further deformation by dislocation creep.*

[A1_30] We plan to repair expressions so that the paper gives no misunderstanding to readers.

[Rev1_30] The relevant section has been deleted.

[R1_31] *Diffusional creep is very unlikely to occur along deep ice cores, first because the required densities of vacancies or interstitials are not met and also because it would not lead to the observed textures and dynamic recrystallization mechanisms.*

[A1_31] We understand the reviewer's view. However, it seems also impossible to rule out this possibility conclusively.

[Rev1_31] As answered above.

[R1_32] *L. 290-294: the relationship between climatic period (and impurity content) and microstructure has been studied a long time ago, and should be referred to here.*

*See for instance Duval and Lorius 1980 (EPSL), Weiss et al. 2002 (Ann. Glaciol, at Dome Concordia), Durand et al. 2006 (JGR). The role of GB pinning on GB migration and therefore on the dynamic recrystallization mechanisms has been widely documented and even modeled in some of these articles.*

[A1_32] In Section 5.1.1 containing the L.290-294 in it, nothing was discussed on the relationship between climatic period (and impurity content) and microstructure. Relationship between climatic period (impurity content) and microstructure is discussed in other section (e.g., 5.2.3).

[Rev1_32] As answered above.

[R1_33] *The sentence "We interpret the wide scatter as indicative of grain nucleation and subsequent growth" is very vague. How is this interpretation related to already known mechanisms of dynamic recrystallization?*

[A1_33] We cite two sentences here. "*We suggest that evidence includes a wide scatter of dots with low values of Δε in figures, leading to smaller average Δε values in interglacial periods, accompanied by large SD values. We interpret the wide scatter as indicative of migration recrystallization, and grain nucleation and subsequent growth.*"
We meant simply here that the wide scatter was not caused by errors but this physical mechanism. Our views for recrystallization are given in several sections within this preprint, citing important papers. This location of the paper is not suitable for developing interpretation related to already known mechanisms of dynamic recrystallization.
In the revised version, we will attempt to make intended meaning clearer. In addition, along with [R32], we need to seek for a proper location to mention already known mechanisms of dynamic recrystallization.

[Rev1_33] The relevant section has been deleted.

[R1_34] *About the scatter of the permittivity signal, I could also interpret it as being related to the small measurement step regarding the size of the grains, that is the characteristic scale of the COF pattern. I would expect a measuring step lower that this grain size to create this variability as each measurement only account for a low number of grains. Then, the effect of the variability is reduced by the moving averaging.*

[A1_34] In the paper, we deliberately present raw data to show variability. Bulk values are provided as averages over 50 cm. Please refer to [A1_5], too.

[Rev1_34] As answered above.

*- Part 5.1.2:*

[R1_35] *l. 315-316. It is very well know that dislocation creep and dynamic recrystallization mechanisms are temperature-controled mechanisms! No need to "speculate" it. Indeed, both dislocation velocity and grain boundary migration are temperature-dependent through Arhenius type of law…*

[A1_35] We will repair this point in revision.

[Rev1_35] We rewrote the sentence at this point, to avoid readers' misunderstandings in Section 5.2.2.
*We hypothesize that the temperature environment within the ice sheet, through mechanisms such as dislocation creep and recrystallization, might have resulted in the commonality of COF at three points in the ice sheets.*

[R1_36] *I don't see why molecular diffusion is mentioned here?*

[A1_36] We agree that our expression was not good enough. We intended to express thermal activation of molecular processes as follows. "In ice physics, it is well-accepted that the mass transport phenomena such as plastic deformation, the molecular diffusion process and recrystallization are highly temperature-dependent."

[Rev1_36] Please refer to [Rev1_35].

[R1_37] *l. 321-322. What is meant by "more effective as a condition to trigger active emergence of recrystallization"? A strong clustered texture will be hard to compress, and therefore dynamic recrystallization is necessary as an accommodation process (or fracturing at higher strain rates) to enable further deformation, but such a clustered texture is very easy to shear (when the cluster orientation is normal to the shear plane, which is the case here), and therefore deformation can occur without or with a limited amount of dynamic recrystallization. This sentence is too vague, and it strengthens the confusion mentioned before about the role of simple shear in the origin of the highly clustered texture.*

[A1_37] We were incorrect on temperature of GRIP core ice. Thus, we need to correct statements at this part of the paper. As for l. 319-322 in the preprint, we will rewrite this part as,

*"In this hypothesis, at the bottom of the UP80%, the thickness of ice is approximately 10% of the original ice equivalent thickness at the time of deposition (Figure 9f). At this depth, the eigenvalue $a_3^{(2)}$ reaches about 0.93. In the absence (or faint presence) of shear stress, a strongly clustered texture will be difficult to compress or shear, thereby necessitating dynamic recrystallization as an accommodation process. This state of saturation of the c-axis cluster, along with the common temperature range may be more effective as a condition to trigger active emergence of recrystallization."*

In the UP80%, we find no indication of shear. In the LO20%, weaker cluster of COF is not very favorable for shear.

[Rev1_37] We revised in the first paragraph of the Section 5.2.2.

[R1_38] *The texture is also likely more clustered in the glacial ice because dynamic recrystallization is expected to be less active due to GB pinning.*

[A1_38] Yes, it is one of points in our discussions.

[Rev1_38] As answered above.

*[R1_39] l. 330-333: there is nothing new in this statement… please refer to existing work about dynamic recrystallization along ice core that I have mentioned before.*

[A1_39] Based on the comments, we remove two sentences here. Please refer to [A8], too.

[Rev1_39] We removed the two sentences. Please refer to [Rev1_8], too.

*- Part 5.2.1:*
[R1_40] *L. 350-351: "The recrystallization processes require thermally activated molecular diffusion"… This sentence is very vague and likely incorrect. Recrystallization occurs mostly by dislocation mobility, either within the crystals (formation of subgrains, dislocation pile-ups, etc.) or at GB by enabling grain boundary migration (under the driving force of dislocation-based strain energy). I don't see why molecular diffusion is mentioned here.*

[A1_40] We will rewrite first part of 5.2.1 as follows. We simplified description to avoid any misleading or misunderstanding.

*In ice physics, it is well-accepted that the mass transport phenomena such as plastic deformation, the molecular diffusion process and recrystallization in are highly temperature-dependent (e.g., Petrenko & Whitworth, 1999). The conditions of ice sheets in Antarctica, in terms of temperature and stress, are located on a boundary zone between dislocation and diffusional creep on the deformation mechanism map (e.g., Duval et al., 1983; Goodman et al., 1981; Shoji, 1978). When ice is under temperatures close to the melting point in the LO20%, its viscosity is lower, and diffusion coefficients are higher compared to the colder ice, the rate at which recrystallization occurs also increases with temperature (e.g., Petrenko and Whitworth, 1999).*

[Rev1_40] We rewrote the first part of 5.1.1 as suggested, except the first sentence.

[R1_41] *L. 356: The shear strain component is only mentioned here. It should be evoked much before, see previous comments about part 5.1.1.*

[A1_41] Please refer to [A0_3]. In our view, shear is not necessary for explanation in the UP80% at DF and EDC.

[Rev1_41] As answered above. Please also refer to [Rev0_3].

*- Part 5.2.2:*
[R1_42] *l. 369-370: how is the mechanism mentioned here possible? The different layers are linked together and cannot rotate or shear independently? Dislocation creep accommodated by dynamic recrystallization tends to align the c-axes with the vertical to the shear plane (see references mentioned before), and the observed texture is, to my point of view, related to the fact that the shear plane is not horizontal?*

[A1_42] Please let us to explain. Yes, the adjacent layers are linked together, keeping continuity in the ice sheet. When they deform independently (accompanying rotate or shear), it cannot happen without occurrence of local shear between adjacent layers. When strain is still small, local shear at each layer boundaries will be small, sometimes being within grain scale strains. However, when strain grows, layers need to be unevenly deformed, showing features of apparent difference between layers. Such features include appearance of undulation of layers, wavy change of layer thickness from one location to another, or when it grows, it will form boudinage features. We find such features when we move away from Dome Fuji, for example. When we are away more, layer structure itself tends to be disrupted. We find many of such features near Dome Fuji with radar sounding. Dislocation creep accommodated by dynamic recrystallization tends to align the c-axes with the vertical to the shear plane (see Figure 10), and the observed texture is related to the fact that the shear plane is not horizontal. If planes are perfectly horizontal down to the ice sheet

bed, no bias, which rotates cluster of c-axis meridionally to a certain direction, will occur. In this case, if recrystallization make c-axes well away from the vertical, these will rotate towards the vertical from all orientations. This is a scenario only if this ideal situation exists. In reality, the cluster axis of single pole COF incline both at Dome Fuji (present study) and at EDC (Figure 6 in Durand et al., 2009) in the LO20% zone. We hope our explanation here makes sense for readers.

Readers can also refer to Section 4.3 of Durand et al., (2007). We can find a view by these authors.

[Rev1_42] As answered above.

[R1_43] *Fig.10: for me, fig 10d represents bending and not rigid rotation? Can a layer experiment rigid rotation independently of the mechanical constraints that surround it? ("system's rigid-body rotation", l. 372).*

[A1_43] It is just a matter of scale of view. If an observer looks at the ice sheet around the subglacial trench in a horizontal scale of several kilometers, in this scale, ice is bending toward the trench. If an observer sees ice core scale of approximately several hundred meters, ice body is rotating.

[Rev1_43] As answered above. Explanation of rigid-body rotation was added to Figure 8.

[R1_44] *l. 374-375: "Thus, in the circumstantial evidence for dominance of the simple shear…". This sentence appears useless. Why are the angles "inconsistent"? Do we need that for inferring simple shear at these depths?*

[A1_44] First, as we have documented, in the UP80%, shear is not significant. In the LO20%, it is a matter of discussions. Thus, we need to discuss simple shear at these depths. A deviation between the cluster of c-axes and a vector orthogonal to the original layers occurs solely due to the presence of simple shear strain; there are no other possibilities. Therefore, the inconsistency of these angles indicates the dominance of simple shear. Please refer to [A0_3] and [A41], too. It would be very informative if angular relation between IACC and visible layer inclination will be examined in the EDC core. IACC is published as Figure 6 in Durand et al., 2009. If we can access layer inclination data, we can investigate.

[Rev1_44] As answered above. Please also refer to [Rev0_3].

[R1_45] *Fig 10a,b,c: the sketches are not precise enough since it depends on the dominant stress, if it is compression or shear. In particular, 10c holds for compression but not for shear. If shear is parallel to the layers, c-axes will be perpendicular to them (with or without dynamic recrystallization).*

[A1_45] Fig 10a,b,c simply sketches angle relation between the IAVL and the IACC as ice becomes deep. Driving force of the ice flow in the gravity in the vertical. There is a steep bedrock with an angle of about 45 degrees. Then, as ice moves closer to bed, simple shear will grow. Precision of the stress/strain configuration is not something which should be discussed here.

[Rev1_45] As answered above.

*- Part 5.2.3:*

[R1_46] *There are many repetitions from previous parts, analyses/interpretations that are mixed with discussion… It goes in favor of the necessity to re-organise the discussion with a clearer story to tell, especially focusing on either the confirmation of existing works (that must be referred to) and/or on new interpretations (although they appear to be few in this work).*

[A1_46] Please refer to [A0_1].

[Rev1_46] We re-organized through the manuscript with beware of repetition. Please also refer to [Rev0_1].

[R1_47] *l. 435: about the impact of dust, work by Weiss et al., Durand et al., mentioned before suggest another explanation with dust impeding grain boundary motion during recrystallization, and therefore enhancing the impact of deformation versus recrystallization on the final COF. The final COF will be either less favorable for compression creep or more favorable for horizontal shear… Therefore not so straight forward!*

[A1_47] We agree that the role of dust particles is not so straightforward. However, in our previous studies at the upper 80% of the DF ice core Saruya et al. 2022b, we found that the dust particles could have an impeding effect on the COF clustering. The mechanism is still unresolved. We suppose whether GB pinning occurs or not depends on the various factors, such as GB migration velocity, microparticle sizes, temperature and so on, as it was proposed in Durand et al., 2006. We could find both GB pinning and microparticles segregation at GBs in our observations in the deeper sections.

[Rev1_47] As answered above.

[R1_48] *l. 441-445: please refer to existing work! Dust particles (and/or insoluble particles?) are mainly located at GBs. They may enhance dislocation production (it has not been clearly proved) BUT they pin GBs. In the works mentioned before (Weiss et al. 2002 for instance), it is shown that the pinning force lead to a critical grain size very close to the one measured in glacial ice.*

[A1_48] According to the review by Stoll et al. (2021b) and a case study for NEEM ice core by Eichler et al. (2017), dust particles located not only at GBs but also at grain interiors and triple junctions. To the best of our knowledge, the role of microparticles in ice deformation (dislocation creep) is not well understood. Production of dislocation is one of the possibilities. In contrast, microparticles may act as sink of dislocation like GBs. There are many existing works regarding impurities. So, we will refer Stoll et al. (2021b) here since this article reviews the impact of impurities on deformation and microstructures of polar ice.

[Rev1_48] We added the description above by referring to Stoll et al. (2021b), at the second paragraph of Section 5.1.3.

[R1_49] *Durand, Gillet-Chaulet et al. 2007 (Climate of the Past) have modeled the impact of changes in ice viscosity with climatic transitions on ice flow. This work could be mentioned.*

[A1_49] Regarding this, please refer to [A0_3]. We recognise that Durand et al. (2009) concluded that simple shear is the most important factor in the COF contrast between glacial and interglacial periods.

[Rev1_49] Please refer to [Rev0_3].
In addition, the work by Durand et al. 2007 (Climate of the Past) is mentioned in the section of conclusion. We proposed that development of COF in the UP80% is explainable without significant presence of simple shear both at DF and EDC.

[R1_50] *I think diffusion creep should not be mentioned here, as there are too few evidence of its likeliness to occur… Unless the authors can provide some NEW statements about it.*

[A1_50] While there is no new information about diffusion creep, it is also true that diffusion creep cannot be ruled out. Azuma et al. (2000) suggested that the contribution of diffusion creep might be significant at depths with smaller grains.

[Rev1_50] As answered above.

[R1_51] *L. 481: please refer to Weiss et al. 2002 and Durand et al. 2006 regarding Dome C.*

[A1_51] Thanks for the comment. We will write this as,

*The same trend is common in the Vostok, GRIP and EDC ice cores (Durand et al., 2006; Lipenkov et al., 1989; Thorsteinsson et al., 1997; Weiss et al., 2002).*

[Rev1_51] To make the paper more concise, we removed the related paragraph.

[R1_52] *l. 481-482: "we can hypothesize that the physical phenomena…" this sentence is too vague. Please be more precise. Which mechanism does what, and what does your observation bring "to the story" already written by previous works?*

[A1_52] Thanks for the comment. To address this comment, we will write sentences including this as,

*Azuma et al, (1999; 2000) reported that the grain sizes in the DF1 ice core in the UP80% is correlated with δ18O variation and concentration of impurities, which was confirmed in this study (Figures 5 and 9). That is, grain size, dusts and ions, and δ18O are all well correlated. The same trend is common in the Vostok, GRIP and EDC ice cores (Durand et al., 2006; Lipenkov et al., 1989; Thorsteinsson et al., 1997; Weiss et al., 2002). Thus, we can hypothesize that the grain growth determined by deposition of impurities are common across a wide area of the ice sheet.*
We limited the statement to grain growth.

[Rev1_52] To make the paper more concise, we removed the related paragraph.

[R1_53] *- Part 5.2.4: again, what is the effect of the grain size, and therefore the number of grains in the measured area on the standard deviation?*

[A1_53] Please refer to [A5] and [A34].  DTM is a volume-weighted average within the volume covered by the Gaussian distribution of the beam. Therefore, the number of grains in the beam (at a spot) is different depending on depth (i.e., grain size). In the LO20%, total number of grains in a beam varies by an order of magnitude from 10 to $10^2$. Thus, when grain is large, a spot measurement is not sufficient to derive statistics. However, when we perform moving average, such as over from 10 cm to about 50 cm, total grain number in the moving average is of the order of $10^2$ to $10^3$. Thus, by making moving averages, we can assume that both averaged value and the SD has no significant grain size effect (no large influence from limited grains) in the DTM data and discussion of COF development. In the revised manuscript, we will explain that the DTM has no grain size effect.

[Rev1_53] Please refer to [Rev1_5].

*Part 5.3:*
*- Part 5.3.1:*
[R1_54] *To my point of view, this part reveals a misunderstanding of the deformation mechanisms of ice (maybe this is due to the way it is written?). As mentioned in my general statement, GBS should not be evoked here with so few proof of its existence. This is a mechanism that is very unlikely to occur along ice cores (large grains, low strain rate, low stresses, efficient accommodation mechanisms…) and that cannot be proven by a few 2D microstructure observations.*

[A1_54] As indicated by the reviewer, there is no significant evidence that GBS via microshear is occurring. In the revised manuscript, we will emphasize that the GBS via microshear is one of the possible mechanisms that can explain the observed microstructure (brick-wall pattern) in the impure layers. We believe that it is worth mentioning that the brick-wall patterns were observed only in impure layers. Also, we will add quantitative discussions on grain shape (e.g., aspect ratio of grains). Also, please refer to [A9] above.

[Rev1_54] We introduced GBS via microshear as one of the possible mechanisms suggested by Faria et al. (2009) in Section 4.5.2. Also, we added the description about the aspect ratio in Section 4.5.2 and data in Table A4 in Appendix A5

[R1_55] *Microshear is mentioned, but I personally don't know what it means! Any dislocation glide along a basal plane created a microshear…*

[A1_55] Microshear deformation mechanism and characteristics of the microstructure are explained in Faria et al. (2009) in detail. Their interpretation was based on analogue experiments by Bons and Jessel (1999) using octachloropropane. Llorens et al. (2016) also discussed based on the numerical simulation. Please see also [A9 and A54].

[Rev1_55] We removed detailed descriptions of microshear to make paper more concise.

[R1_56] *How can GBS contribute to grain size reduction? How can it create new boundaries?? Please refer to the work of Ashby 1973 (Acta Metallurgica) for instance to better understand GBS. GBS is not a recrystallization mechanism! This is a deformation mechanism that is likely to take place in very fine-grained materials leading to superplasticity, in very specific conditions.*

[A1_56] According to Faria et al. (2009), the formation of microshear bands creates new, flat GBs parallel to the shear plane, resulting in the grain size reduction by splitting the grain in two. This behavior is similar to rotation recrystallization (as stated in L549). Also, please refer to [A9] above.

[Rev1_56] We removed the sentence referring to the relationship between GBS and grain size reduction because the process is not robust.

[R1_57] *To correctly interpret fig. 12, one would need the measurement of misorientations in order to be able to provide a clear distinction between subgrains, grains, and interpret them in terms of mechanisms at play. For instance, figure 12a could also present an example of some GB pinning by a subgrain as observed in Montagnat et al. 2015. In figures 12f and g, it is not straightforward to interpret the observations as grain boundary migration. The shape of the GB is not sufficient to my point of view. In figure 12i I do not see quadruple junctions but the effect of a 2D sectioning of a larger grain. Just to say that these figures can easily be "over interpreted" in line with our scientific prejudice.*

[A1_57] In our measurements, it is difficult to discuss the misorientaiton within a grain. Thus, we provide these figures as possible examples of boundary migration based on the shape of GBs and sGBs. Microstructural examples of migration recrystallization with and without nucleation are shown in following articles: Fig.4 in Weikusat et al. (2009), JoG; Fig.7 in Kipfstuhl et al. (2009), JGR; Figs 5 and 11 in Faria et al. (2014b). According to them, for example, many sGBs and dislocation walls irradiating from a bulged grain boundary, a newly nucleated grain grows into the highly strained region characterized by numerous sGBs and dislocation walls. Our observations shown in Figure 12f,g are consistent to their interpretations. In Figure 12e,h, we could observe microparticles segregation and pinning. Thin lines adjacent to GBs are the reverse side GBs. So microparticles are segregated on the migration planes of the crystal grains (2.5D). Additionally, pinning by microparticles could indicate boundary migration (e.g., Stoll et al., 2021b in Frontiers in Earth Science). For these reasons, we interpreted that the boundary migration and migration recrystallization occurred. In Figure 12i, we interpreted the two locations indicated by the arrows as quadruple junctions. However, we cannot exclude the effect of a 2D sectioning. We will remove it. Instead, we emphasize that there are no evident brick-wall patterns (as shown in panel a and b) despite the high concentration of dust particles. We might integrate that description into Figure 6e.

In the revised manuscript, we will reconstruct microstructure images (Figure 12) to provide clear examples and add further explanation with appropriate references.

[Rev1_57] We redesigned microstructure images (Figures 6 and 7) to provide clear examples (with bulged GBs and interlocking grains) and explanations. Interpretative words (e.g., migrating GB) were removed.

- *Part 5.3.2:*

[R1_58] *l.570: the statement about the orientation of nucleus can not be proven with the presented results. Observations, with direct measurements, of nucleus orientations show that they are mainly oriented close to their parent grains (see e.g. Chauve et al. 2017, Phil Trans Roy Soc A).*

[A1_58] In the revised manuscript, we will reconstruct "grain nucleation" images to provide clear examples and focus on the strain induced grain boundary migration with nucleation (SIBM-N) (nomenclature following Faria et al., 2014b).

[Rev1_58] We redesigned grain nucleation images (Figures 7e and 7f). Also, we referred to as "potentially nucleated grains".

[R1_59] *l. 584: Figure 12 can not be used to prove the existence of GBM, please be more cautious in your statement. Some more proof is required.*

[A1_59] Please refer to [A57] above.

[Rev1_59] Please refer to [Rev1_57] above.

[R1_60] *Similarly, the only observations in figure 12 c1 and d1 of square-looking grains are not enough to demonstrate GBS since, first, they could also be a 2D sectioning effect, and second, more statistics is required to demonstrate GBS. Please be more cautious or remove.*

[A1_60] We showed Figure 12 c1 and d1 as possible examples of nucleated grains. They have no internal structures. In the revised manuscript, we will reconstruct "grain nucleation" images to provide clear examples and focus on the strain induced grain boundary migration with nucleation (SIBM-N) (nomenclature following Faria et al., 2014b).

[Rev1_60] Please refer to [Rev1_58] above.

[R1_61] *"The orientations of the c-axis in nucleated grains…" you do not have any direct observation of nucleus orientations, and some existing observations (e.g. Chauve et al. 2017, but maybe not only) are not in favor of such a statement. Please be more cautious or remove.*

[A1_61] We will remove the sentence because we do not have direct observation of the grain orientations. In a situation that we find so wide scatter of c-axis, it is highly likely that such small grains with c-axis well away from the surrounding grains are nucleated grain, which appeared in the LO20%. Of course, we are not observing a moment when nucleation is occurring.

[Rev1_61] We removed the sentence.

[R1_62] *l. 585: "The presence of numerous sGBs implies high stored strain energy". This statement is wrong. Stored energy is related to the total dislocation density, during deformation, while sGBs only represent the population of geometrically necessary dislocations that remains after unloading! There exist not link between the density of geometrically necessary dislocations and the total dislocation density, and Lopez-Sanchez et al. 2023 (EPSL) have even shown that you can have a lot of strain in some given grains and NO sGBs remaining after unloading. Please remove.*

[A1_62] We rephrase to "highly and heterogeneously strained region" with references Faria et al. (2014b) and Stoll et al. (2021a).

[Rev1_62] This rephrase was done in the first paragraph of the revised 4.5.3.

[R63] *For the remaining discussion related to nucleation and dynamic recrystallization, please refer and mention existing work in order not to "propose" explanations that have already been given, but rather to "confirm" that the processes you observe are in phase with previous observations and analyses of dynamic recrystallization along ice cores.*

[A1_63] In the revised manuscript, we will add references when we find it appropriate. We will state earlier knowledge as it is.

[Rev1_63] These were addressed mainly in Section 1.4 and Section 4.5.3 in the revised manuscript.

[R1_64] *l. 596: please site Weiss et al. 2002 and/or Durand et al. 2006 closely related to what you observed instead of Cuffey and Patterson that is a general book.*

[A1_64] We will refer to Durand et al. (2006) for the case study of EDC ice core and Stoll et al. (2021b) in Frontiers in Earth Sciences as a review paper.

[Rev1_64] We addressed it at Section 5.1.4.

[R1_65] *l. 603: Please remove the following statement "Given the high concentration of dust particles, it is likely that the area experienced GBS via microshear" since, first, you have not enough proof for such an assertion, secondly, how can dust particle concentration be linked to GBS, where does that come from, and finally because the meaning of "GBS via microshear" is very uncertain!*

[A1_65] We will modify it as follows:
*There are no evident brick-wall patterns observed in panel a and b despite the concentration of dust particles is high. We suspect that the patterns could be lost by recovery and recrystallization due to high temperature close to melting point.*

[Rev1_65] We addressed it in the first paragraph in Section 4.5.2.

*Part 5.4:*
[R1_66] *The zoning must be put in relation with what has already been observed and analyzed in other ice cores. For instance, Dome C, Talos Dome, NEEM ice cores. Along Talos Dome we have shown that some dynamic recrystallization was necessary to reproduce the observed COF in complement to dislocation creep (this latter would lead to too strong COF) (Montagnat et al. 2012, EPSL). Along the NEEM ice core we have observed the impact of shear on the COF in exact connection with the change in the climatic origin of the ice, for instance.*

[A1_66] Considering the similar development of COF in each dome site core (DF and EDC in Fig.8; EDC and GRIP from Durand et al., 2009; depth of maximum COF cluster in Table 3), we believe that the zoning of DF ice core could be applied to EDC and GRIP ice cores. However, regarding the detailed zoning defined in the DF ice core depths at below 2400 m, it is difficult to discern in the EDC and GRIP core because the COF fluctuation is too large, and data are sparser.
We will not mention the NEEM ice core since it is not a dome core. We should not make complex mixing. Also, we will refer to the Talos Dome in Sect. 5.1.2. Status of the Talos Dome is as we stated in [A0_2].

[Rev1_66] As we explained previously, zoning cannot be easily applied to sites with sparse data. In addition, we did not mention NEEM core since it is not a dome core. We should not make complex mixing. Status of the Talos Dome is given in introduction, at the end of Section 1.3.

*Part 5.5:*
[R1_67] *Information about COF and the resulting layering of ice viscosity along ice cores cannot be straightforward related to the dating of ice core or what is called "radioglaciology" (if I understood well?). See e.g. Durand et al. 2007 (Climate of the Past) or Buiron et al. 2011 about Talos Dome age-scale (Climate of the Past). Maybe some recent radar measurements studies could be mentioned here too.*

[A1_67] Please refer to [A0_2]. In the present paper, dating or strain complexity at coastal ridge at the edge of the ice sheet plateau is not, at least, one of major subjects. Priority is low to develop statements on the Talos site. It will make discussions unnecessarily complex, lengthy, and unfocused.

We already cited key papers that explicitly discussed radio echo free zone. If we find some paper suitable to add as citation in this context, we will do it. We continue to survey literatures.

[Rev1_67] In the stage of the preprint, we already cited key papers that explicitly discussed radio echo free zone. If did not find additional paper suitable as citation in this context.

*Conclusion:*

[R1_68] *l. 689: I think it is not correct to mention here the "elongated shape" of crystals that has not been statistically characterized, and for which you can not quantify the relative occurrence. You could mention it in the text as an example of some local observations but not in the conclusion since it will give too much weight to this weak observation.*

[A1_68] We will add quantitative discussion in the revised manuscript (main text or appendix). We investigated the aspect ratio of grains in impure layers and other (interglacial) layers.

[Rev1_68] We added information of the aspect ratio at the beginning in Section 4.5.2 and Appendix A5 and throughout the discussions. We also added information of the aspect ratio to Figure 9. The discussion on the aspect ratio is now one of the core topics in this paper.

[R1_69] *l. 690: same comment about this mention of GBS and microshear that has not been shown to occur significantly in your observation, with strong enough statistics and scientific evidence, to worth being mentioned as an important mechanism unless with the aim to orient the reader interpretation.*

[A1_69] We will not mention GBS via microshear in Conclusion because this mechanism is one of the possibilities of mechanism that can explain the microstructures (brick-wall pattern) of impure layers and there is not significant evidence.

[Rev1_69] As answered above.

[R1_70] *l. 705: temperature also play a role in the observed mechanisms, not only strain. For instance, along NEEM we observed dynamic recrystallization high along the core that was not observed for the same amount of strain along Dome C or Talos Dome.*

[A1_70] Temperature is also an important factor in controlling recrystallization process. We will add "temperature".

[Rev1_70] We mentioned temperature.

*Conclusive statement:*

[R1_71] *I recommend that major revisions are made prior to publication of this article. In particular, attention must be paid to providing sufficiently well-founded scientific evidence before concluding on the importance of a mechanism.*

[A1_71] With an opportunity of the major revision, we will largely improve this paper. Also, please refer to [A0_1].

[Rev1_71] We modified through the revised manuscript. Please also refer to [Rev0_1]

[R1_72] *There is also a lack of reference to the work that have already been done for many years about ice core texture and microstructure measurements and their interpretation. This should be corrected.*

[A1_72] We will add many reference papers widely.

[Rev1_72] We added references through the revised manuscript.

[R1_73] *At the end, the novelty of the study stands only on the fact that these deep core measurements have never been published. Care should be taken in the discussion in order to focus on what is potentially new or relevant for the community.*

[A1_73] In the present study, we obtained many results and insights listed in the conclusion section in terms of layered structures and physical properties in the Dome ice core. Especially, knowledge of the deep sections is significantly valuable because there is limited information available so far. Although the discussion of physical processes (such as deformation mechanisms and microstructures) in deep Dome ice core has long been stalled since EDC studies by Durand et al. (2007; 2009), we believe that the new findings obtained by present study would activate debate again (with knowledge from EDML, East GRIP and Oldest ice cores).

Finally, we wish to express once again our sincere gratitude to the reviewer for dedicating the reviewer's valuable time to review our manuscript and for providing critical insights aimed at its improvement.

**References**

Alley, R. B. (1988). Fabrics in polar ice sheets: development and prediction. *Science*, 240(4851), 493-495.

Azuma, N., & Higashi, A. (1985). Formation processes of ice fabric pattern in ice sheets. *Ann. Glaciol.*, 6, 130-134.

Azuma, N., Wang, Y., Mori, K., Narita, H., Hondoh, T., Shoji, H., & Watanabe, O. (1999). Textures and fabrics in the Dome F (Antarctica) ice core. *Ann. Glaciol.*, 29, 163-168.

Azuma, N., Wang, Y., Yoshida, Y., Narita, H., Hondoh, T., Shoji, H., & Watanabe, O. (2000). Crystallographic analysis of the Dome Fuji ice core. In Physics of ice core records (pp. 45-61). Sapporo: Hokkaido University Press.

Bons, P. D., and Jessell, M. W.: Micro-shear zones in experimentally deformed octachloropropane, *J. Struct. Geol.*, 21(3), 323–334, https://doi.org/10.1016/S0191-8141(98)90116-X, 1999.

Budd, W. F., & Jacka, T. H. (1989). A Review of Ice Rheology for Ice Sheet Modelling. *Cold Regions Science and Technology*, 16, 107-144.

De La Chapelle, S., O. Castelnau, V. Lipenkov, and P. Duval (1998), Dynamic recrystallization and texture development in ice as revealed by the study of deep ice cores in Antarctica and Greenland, *J. Geophys. Res.*, 103(B3), 5091–5105, doi:10.1029/97JB02621.

Doquet, V., & Barkia, B. (2016). Combined AFM, SEM and crystal plasticity analysis of grain boundary sliding in titanium at room temperature. *Mechanics of Materials*, 103, 18-27. https://www.sciencedirect.com/science/article/pii/S0167663616302848

Durand, Gillet-Chaulet, F., Svensson, A., Gagliardini, O., Kipfstuhl, S., Meyssonnier, J., et al. (2007). Change in ice rheology during climate variations - implications for ice flow modelling and dating of the EPICA Dome C core. Climate of the Past, 3(1), 155-167. <Go to ISI>://WOS:000247031100002

Durand, Svensson, A., Persson, A., Gagliardini, O., Gillet-Chaulet, F., Sjolte, J., et al. (2009). Evolution of the Texture along the EPICA Dome C Ice Core. 低温科学, 68(Supplement), 91-105. http://hdl.handle.net/2115/45436

Durand, G., Gagliardini, O., Thorsteinsson, T., Svensson, A., Kipfstuhl, S., & Dahl-Jensen, D. (2006). Ice microstructure and fabric: an up-to-date approach for measuring textures. *Journal of Glaciology*, 52(179), 619-630. https://www.cambridge.org/core/product/9F8EF78DC9D831247AA651A4BD646BF6

Duval, P., Ashby, M. F., & Anderman, I. (1983). Rate-controlling processes in the creep of polycrystalline ice. *The Journal of Physical Chemistry*, 87(21), 4066-4074. https://doi.org/10.1021/j100244a014

Eichler, J., Kleitz, I., Bayer-Giraldi, M., Jansen, D., Kipfstuhl, S., Shigeyama, W., et al. (2017). Location and distribution of micro-inclusions in the EDML and NEEM ice cores using optical microscopy and in situ Raman spectroscopy. *The Cryosphere*, 11(3), 1075-1090. https://tc.copernicus.org/articles/11/1075/2017/

Faria, S. H., Kipfstuhl, S., Azuma, N., Freitag, J., Weikusat., I., Murshed, M. M., and Kuhs, W. F.: The Multiscale Structure of Antarctica Part I: Inland Ice, in: Physics of Ice Core Records II, edited by: Hondoh, T., Hokkaido University Press, Sapporo, 39–59, 2009.

Faria, S. H., Weikusat, I., & Azuma, N. (2014a). The microstructure of polar ice. Part I: Highlights from ice core research. *Journal of Structural Geology*, 61, 2-20. https://www.sciencedirect.com/science/article/pii/S0191814113001740

Faria, S. H., Weikusat, I., & Azuma, N. (2014b). The microstructure of polar ice. Part II: State of the art. *Journal of Structural Geology*, 61, 21-49. https://www.sciencedirect.com/science/article/pii/S0191814113002009

Fujita, S., Parrenin, F., Severi, M., Motoyama, H., and Wolff, E. W.: Volcanic synchronization of Dome Fuji and Dome C Antarctic deep ice cores over the past 216 kyr, *Clim. Past*, 11, 1395-1416, 2015.

Goldsby, D. L., & Kohlstedt, D. L. (1997). Grain boundary sliding in fine-grained Ice I. *Scripta Materialia*, 37(9), 1399-1406. https://www.sciencedirect.com/science/article/pii/S1359646297002467

Goodman, D. J., Frost, H. J., & Ashby, M. F. (1981). The plasticity of polycrystalline ice. *Philosophical Magazine A*, 43(3), 665-695. https://doi.org/10.1080/01418618108240401

Humphreys, F. J., & Hatherly, M. (2004). Recrystallization and Related Annealing Phenomena.

Jacka, T. H., & Budd, W. F. (1989). Isotropic and Anisotropic Flow Relations for Ice Dynamics. *Annals of Glaciology*, 12, 81-84.

Kipfstuhl, S., Faria, S. H., Azuma, N., Freitag, J., Hamann, I., Kaufmann, P., et al. (2009). Evidence of dynamic recrystallization in polar firn. *Journal of Geophysical Research-Solid Earth,* 114(B05204). <Go to ISI>://WOS:000266365300001

Kuiper, E.-J. N., Weikusat, I., de Bresser, J. H. P., Jansen, D., Pennock, G. M., and Drury, M. R.: Using a composite flow law to model deformation in the NEEM deep ice core, Greenland – Part 1: The role of grain size and grain size distribution on deformation of the upper 2207 m, The Cryosphere, 14, 2429–2448, https://doi.org/10.5194/tc-14-2429-2020, 2020.

Langway, C. C.: Ice fabrics and the universal stage, SIPRE Tech. Rep., 62, 1958.

Linne, M. A., Bieler, T. R., & Daly, S. (2020). The effect of microstructure on the relationship between grain boundary sliding and slip transmission in high purity aluminum. *International Journal of Plasticity*, 135, 102818. https://www.sciencedirect.com/science/article/pii/S0749641920300152

Lipenkov, V. Y., Barkov, N. I., Duval, P., & Pimienta, P. (1989). Crystalline Texture of the 2083 m Ice Core at Vostok Station, Antarctica. *Journal of Glaciology*, 35(121), 392-398. https://www.cambridge.org/core/article/crystalline-texture-of-the-2083-m-ice-core-at-vostok-station-antarctica/493567B5503E2584021CEC2072F9CBFE

Llorens, M.-G., Griera, A., Bons, P. D., Lebensohn, R. A., Evans, L. A., Jansen, D., and Weikusat, I.: Full-field predictions of ice dynamic recrystallisation under simple shear conditions, *Earth Planet. Sci. Lett.*, 450, 233–242, http://dx.doi.org/10.1016/j.epsl.2016.06.045, 2016.

Montagnat, M., Azuma, N., Dahl-Jensen, D., Eichler, J., Fujita, S., Gillet-Chaulet, F., et al. (2014). Fabric along the NEEM ice core, Greenland, and its comparison with GRIP and NGRIP ice cores. *The Cryosphere*, 8(4), 1129-1138. https://tc.copernicus.org/articles/8/1129/2014/

Montagnat, M., Buiron, D., Arnaud, L., Broquet, A., Schlitz, P., Jacob, R., & Kipfstuhl, S. (2012). Measurements and numerical simulation of fabric evolution along the Talos Dome ice core, Antarctica. *Earth and Planetary Science Letters*, 357-358, 168-178. https://www.sciencedirect.com/science/article/pii/S0012821X12005213

Montagnat M, Chauve T, Barou F, Tommasi A, Beausir B and Fressengeas C (2015) Analysis of Dynamic Recrystallization of Ice from EBSD Orientation Mapping. Front. Earth Sci. 3:81. doi: 10.3389/feart.2015.00081

Oyabu, I., Iizuka, Y., Kawamura, K., Wolff, E., Severi, M., Ohgaito, R., et al. (2020). Compositions of dust and sea salts in the Dome C and Dome Fuji ice cores from Last Glacial Maximum to early Holocene based on ice-sublimation and single-particle measurements. *Journal of Geophysical Research: Atmospheres*, 125, e2019JD032208. https://doi.org/10.1029/2019JD032208

Parrenin, F., Fujita, S., Abe-Ouchi, A., Kawamura, K., Masson-Delmotte, V., Motoyama, H., Saito, F., Severi, M., Stenni, B., Uemura, R., and Wolff, E. W.: Climate dependent contrast in surface mass balance in East Antarctica over the past 216 ka, *Journal of Glaciology*, doi: 10.1017/jog.2016.85, 2016. 1-12, 2016.

Petrenko, V. F., & Whitworth, R. W. (1999). Physics of Ice. Oxford: Oxford University Press.

Poirier, J.-P. (1985). J.-P. Poirier 1985. Creep of Crystals. Cambridge Earth Science Series. xiv + 260 pp. Cambridge University Press. https://www.cambridge.org/core/product/7B662D4F9279DE7B58004F74E67CCC25

Saruya, T., Fujita, S., and Inoue, R.: Dielectric anisotropy as indicator of crystal orientation fabric in Dome Fuji ice core: method and initial results, J. Glaciol., 68(267), 65–76, https://doi.org/10.1017/jog.2021.73, 2022a.

Saruya, T., Fujita, S., Iizuka, Y., Miyamoto, A., Ohno, H., Hori, A., Shigeyama, W., Hirabayashi, M., and Goto-Azuma, K.: Development of crystal orientation fabric in the Dome Fuji ice core in East Antarctica: implications for the deformation regime in ice sheets, The Cryosphere 16(7), 2985–3003, https://doi.org/10.5194/tc-16-2985-2022, 2022b.

Shoji, H. (1978). Deformation mechanism map of ice. *Journal of Glaciology*, 21(85), 419-427.

Stoll, N., Eichler, J., Hörhold, M., Erhardt, T., Jensen, C., and Weikusat, I.: Microstructure, micro-inclusions, and mineralogy along the EGRIP ice core – Part 1: Localisation of inclu- sions and deformation patterns, The Cryosphere, 15, 5717–5737, https://doi.org/10.5194/tc-15-5717-2021, 2021a.

Stoll, N., Eichler, J., Hörhold, M., Shigeyama, W., & Weikusat, I. (2021b). A Review of the Microstructural Location of Impurities in Polar Ice and Their Impacts on Deformation. 8. Review. https://www.frontiersin.org/articles/10.3389/feart.2020.615613

Thorsteinsson, T., Kipfstuhl, J., & Miller, H. (1997). Textures and fabrics in the GRIP ice core. *Journal of Geophysical Research*, 102(C12), 26583-26599.

Urbini, S., Frezzotti, M., Gandolfi, S., Vincent, C., Scarchilli, C., Vittuari, V., & Fily, M. (2008). Historical behaviour of Dome C and Talos Dome (East Antarctica) as investigated by snow accumulation and ice velocity measurements. *Global and Planetary Change*, 60, 576-588.

Weikusat, I., Kipfstuhl, S., Faria, S. H., Azuma, N., & Miyamoto, A. (2009). Subgrain boundaries and related microstructural features in EDML (Antarctica) deep ice core. *Journal of Glaciology*, 55(191), 461-472. https://www.cambridge.org/core/product/842AB84D845020F75860A6B1B4B5F1FB

Weikusat, I., Jansen, D., Binder, T., Eichler, J., Faria, S. H., Wilhelms, F., Kipfstuhl, S., Sheldon, S., Miller, H., Dahl-Jensen, D., and Kleiner, T.: Physical analysis of an Antarctic ice core – towards an integration of micro- and macrodynamics of polar ice, Phil. Trans. R. Soc. A, 375, 20150347, http://dx.doi.org/10.1098/rsta.2015.0347, 2017.

Weiss, J., Arnaud, L., Duval, P., Gay, M., Petit, J. R., & Vidot, J. (2002). Dome Concordia ice microstructure: impurities effect on grain growth. *Annals of Glaciology*, 35, 552-558. https://www.cambridge.org/core/product/0D65909780E3C7AD641475DB1BD0DB7C

**How the authors addressed the comments by RC2 in their responses to the revised manuscript**

Tomotaka Saruya[1], Atsushi Miyamoto[2], Shuji Fujita[1,3], Kumiko Goto-Azuma[1,3], Motohiro Hirabayashi[1], Akira Hori[4], Makoto Igarashi[1], Yoshinori Iizuka[5], Takao Kameda[4], Hiroshi Ohno[4], Wataru Shigeyama[3*], Shun Tsutaki[1,3]

[1] National Institute of Polar Research, Tokyo 190-8518, Japan

[2] Institute for the Advancement of Graduate Education, Hokkaido University, Sapporo 060-0817, Japan

[3] Polar Science Program, Graduate Institute for Advanced Studies, SOKENDAI, Tokyo 190-8518, Japan

[4] Kitami Institute of Technology, Kitami 090-8507, Japan

[5] Institute of Low Temperature Science, Hokkaido University, Sapporo 060-0819, Japan

*Currently at: JEOL Ltd., Tokyo, Japan

**Correspondence**: Tomotaka Saruya (saruya.tomotaka@nipr.ac.jp), Atsushi Miyamoto (miyamoto@high.hokudai.ac.jp), Shuji Fujita (sfujita@nipr.ac.jp)

**Explanation of the text colors:**
- *Black Italic text: Comments from Reviewer 2 dated April 15, 2024*
- Blue text: Authors' reply to RC2 on May 21, 2024
- Brown text: Explanations of how the authors addressed Reviewer 2's comments in their revised manuscript, submitted on August 17, 2024

In this document, we explain how we addressed the comments provided by RC2 on the revised manuscript. We will use the identifier [Rev2_#] for our explanations.

General comments

In the revised manuscript, we significantly revised the discussions based on the reviewers' comments. Some detailed discussions on specific topics, such as ions and dust particles, were removed or condensed to reduce the overall volume. Following Reviewer 2's comments, we investigated and expanded the discussions on *a*-axis fabric. This led to the discovery of a new and important aspect of *a*-axis organization and grain shape information (aspect ratio). We thank the Reviewer 2 very much for suggesting us this point. Consequently, we needed to discuss the possible causes of *a*-axis organization within the context of dynamic recrystallization. Possibility of the crystal twinning needed to be speculated. In the discussion sections, we focused more on the contrast between impure and less-impure layers. Microstructures are now positioned as supporting information for the discussion on crystal orientation fabric, layered structures, and their relationships. We reorganized sections, combining previous sections 5.3.1 and 5.3.2 with the former section 4.5. Additionally, we redesigned the figures throughout the paper. Including items above, the major modifications are as follows:

- The introduction was expanded and structured, with additional references on the background of the subjects.
- Descriptions of microstructures were reduced and merged into Section 4.5.

- ➤     Microstructural images in Figures 6 and 7 were redesigned.
- ➤     Detailed descriptions of chemical impurities were significantly reduced.
- ➤     The order of discussions within the Discussion section was rearranged (1st: deep sections – 2nd: entire).
- ➤     Discussions on a-axis distribution were added as Section 5.1.5.
- ➤     The description of zoning (previously Section 5.4) was removed.
- ➤     Acronyms were significantly reduced.

[Authors' reply to RC2 on May 21, 2024]

We again thank the reviewers for their careful and critical review of our manuscript and for their thoughtful comments, which have helped to improve it. We also extend our gratitude to all related individuals (reviewers and editors) for dedicating their precious time to this process. Below, we describe our responses to each of the reviewer's comments, presented point-by-point in blue text (with the original comments in black). For convenience in referencing, we provide an ID number for each of the reviewer's comments and for each of our responses. To make these IDs distinguishable from those used for Reviewer 1, we will use labels such as [R2_1] or [A2_1] in this document, indicating review comments and responses for Reviewer 2.

[R2_1] *This paper contains some very interesting and potentially important new data and can make a significant contribution to ice core science. The use of the Dielectric Tensor method to get a much higher resolution data set for COF strength as a function of depth is particularly impressive. The paper also presents full crystal orientation data ([c] and <a> axes and this is an important development as c axes alone do not usually give the full information needed to infer deformation kinematics. Although the paper represents a substantial contribution to ice core science, it needs a lot of re-writing and possibly some more research work.*

*The paper is way too large and tries to cover too much. Any significant outcome is lost as the paper is poorly focused. I would focus on the orientation relationships of COF to layering and the evidence from the microstructure that helps understand deformation kinematics and mechanisms. The cross-correlation work linking COF to chemical impurities seems superfluous and could be thinned down and simplified. There is a lot of repetition in the figures and the paper can be shortened and improved by a redesign of the figure structure. For example figs 7 and 8 can be merged, figs 5 and 9 can be merged.*

[A2_1] Thank you very much for the critical reviews. Based on the reviews, we plan to address major points to the revised manuscript as follows.

We agree to shorten the paper. In the revised manuscript, we plan to reduce the discussion of impurities and microstructures. Regarding impurities, we plan to discuss COF with a focus on the contrast between impure and less-impure layers. Detailed discussion about individual factors (such as chloride ions and dust particles) will be removed or moved to appendix (or to supplementary materials). Microstructures are positioned as supporting information for the discussion of COF, layered structures and their relationship. We plan to combine Sections 5.3.1 and 5.3.2 with Section 4.5. Also, we plan to redesign the figures throughout the paper.

[Rev2_1] Please refer to "A message to the reviewers and editors, and general comments" above.

[A2_1] T*he really key new information is the relative orientation of layers and c-axis maxima. The analysis of these data is not robust as it does not consider the orientation relationships in 3 dimensions. I am pretty sure that the authors already have the information they need to do the analysis in 3D or can get this information with a little extra work and this should be done. If grain shapes were incorporated into this it would be even better. I start my review with this main point, followed by other key issues that I think the authors should address. Apologies I have run out of time on this so have not listed all the minor issues.*

[A2_2] Before this review comment was provided, we presumed that the horizontal orientation of the c-axes cluster and the normal axis of the layer inclination are within the same vertical plain, which means common orientation. We had no reason to suspect, for example, there is a kind of torsion between these two. After reading the review comment, we checked if these two are in the same vertical plane or not. We confirmed that it is true throughout the LO20%. Grain shapes were not a focus of the present paper. We have limited information. It is beyond the scope of the present paper.

[Rev2_2] We mentioned this point at the beginning of the second paragraph in Section 3.1.2 as "*We confirmed that these two are in the same vertical plane throughout the LO20%*".

[R2_3] *The paper is very difficult to read because of the ridiculous number of acronyms. Many of the acronyms represent long expressions that could easily be replaced with much shorter expressions, that could then be used as words so that the reader can read the paper without constant reference to an array of acronyms. This might make the paper a little longer, but it will take less time to read and the reader will understand it better. Some of the acronyms relate to words that do not clearly indicate meaning. I will send an annotated version of the acronym list later.*

[A2_3] Based on the suggestion, we will decrease the number of the acronyms.

[Rev2_3] We removed very many acronyms.

**Specific comments.**

**Major issue: Layering, c-axis inclination and core inclination.**

[R2_4] The relative orientation of layering and c-axis maxima is really important. In this paper it is presented in a way that is ambiguous and potentially misleading. "Layered structures" is in the paper title so it is crucial that there is no uncertainty as to the nature of the constraints on layer orientation.

[A2_4] Please refer to A2_2. We confirmed that the horizontal orientation of the c-axes cluster and the normal axis of the layer inclination are within the same vertical plane throughout the LO20%.

[Rev2_4] Please refer to [Rev2_2].

[R2_5] *Firstly, the description of how layering is measured in this manuscript and in the cited report (DFICPM: et al., 2017) is incomplete. It simply says it was measured with a protractor - it is not clear whether this is a measurement on an arbitrary cut surface (in which case the true inclination is equal to or larger than the measured value) or whether it is a measure of the maximum inclination (true dip) - e.g. by cutting the core along a plane that contains the maximum inclination (perpendicular to strike). This needs to be clarified and the layering added schematically to figure 2.*

[A2_5] The inclination of the layers in the Dome Fuji ice core was measured using two methods, as follows: Thin, cloudy bands are faint features with thicknesses ranging from about 10 mm to 1 mm. We observed the ice cores using a light stage, approximately 250 mm by 600 mm in size, placed on a table. The ice cores, shaped as half vertical cuts of the original cylindrical form, were visually inspected by observers who looked directly down at the core from above the light stage. The ice cores were positioned between the observers' eyes and the light stage. The faint and thin cloudy layers are identified when oriented vertically; only in this orientation do the observers recognize the layers as faint but sharp lines. From angles deviating from this, it is difficult to recognize such layers. By orienting the layers vertically and keeping them orthogonal to the core's inclination on the light stage,

the observers could measure and record the inclination angle of the ice cores using a large protractor. The inclination angles of the ice cores are the same as the inclination of the layers. With this procedure, layer inclination was measured in a 3D manner, and we always measured the maximum inclination angle.

In some cases, we employed another method where the observers used the coordinates of three or more points in each layer within ice cores shaped as half of the core. Using these points, they measured the inclination angle of the layers in 3D, ensuring the orientation was at the point of maximum layer inclination.

At least two observers (in most cases, Miyamoto and Fujita) measured the inclination angles for each layer as a cross-check. Additionally, we repeated measurements several times to gain skills and confirm reproducibility. The observers estimated that the maximum errors in measurement were about 5 degrees. In the year 2006 when the measurements were performed, the orientation of the inclination was not recorded because core orientation was not a topic of interest at that time.

[Rev2_5] Our explanations above were added as "Appendix A3: Detail of the grain size and layer inclination measurements" in the revised manuscript.

[R2_6] *There is no discussion of 3D geometrical relationships. The presentation of inclinations of c axis maxima and visible layers in figure 5 and the written interpretation suggests to the reader that the c-axis maximum and the pole to layering has the same azimuth and lie in a vertical plane. If this is the case, then the cartoon in fig 10c is valid. However, the inclination constraints as presented have an infinite set of possible orientations on cones around the core axis as shown on the stereonet below (Fig. R1). End member solutions where the c-axis maximum and the pole to layering are in the same plane as the core axis (see Fig R1. below) are quite different. An added complication is that the core axis is not vertical, with an inclination of 3 to 6 degrees in this part of the hole (Motoyama et al., 2021), although this angle is small and unlikely to cause significant complications in analysis.*

[A2_6] Please refer to A2_2. We confirmed that the horizontal orientation of the c-axes cluster and the normal axis of the layer inclination are within the same vertical plane throughout the LO20%. For each ice core, borehole inclination is known, we will add the data to one of figures. However, orientation of the borehole inclination for each core is unknown.

[Rev2_6] Please refer to Rev2_2 again. Borehole inclination data were added in Figure 5h.

[R2_7] I *think the authors can do much better here. If there is visible layering in the core then you should be able to measure it in 3D; either by extracting the sine curve of the layer from the outside of the cylindrical core (although I realise that the cylinder probably does not exist now), by measuring the apparent inclination on two non-parallel surfaces or by cutting a plane that contains maximum inclination (maybe this is done, as I said before). A great circle (with a point for the pole) can then be plotted on each stereonet along with the COF data (fig 4) to show the orientation of layering to show the true 3D relative orientation of the c-axis maximum and layers in that sample. If layers are hard to see in the core, they are usually easy to see in a 5mm slice viewed in crossed polars.*

[A2_7] We agree with the idea. This better procedure needs to be used for future measurements, based on present achievement and understanding.

[Rev2_7] As answered above.

[R2_8] *With 3D data the relative orientations of layers, core axis and c-axis max can be shown for all samples in one stereonet. Below (Fig R2) I present three possible data outcomes (Fig R2 b,c,d) presented in a reference frame where the core axis is fixed*

*(plotted in the centre of the stereonet, as it is easier to see the patterns that way) and the c-axis maxima are all assigned to a common azimuth (arbitrary as you do not have azimuthal data).*

[A2_8] Same as above.

[Rev2_8] As answered above.

[R2_9] *This figure could equally be plotted by assigning a common azimuth of the poles to layers - but as drawn it would match better with layers marked on individual sample data in figure 4. The outcome in Fig R2d would be consistent with the generalized cartoon in fig R1b and the geometrical interpretation provided in the paper in fig 10c. The outcome in Fig R2c would be consistent with the generalized cartoon in fig R1c. You cannot draw figure 10c in the paper, nor make any useful interpretation of the c-axis and layer inclinations without this analysis in 3D. Without this data are pretty meaningless.*

 [A2_9] Please refer to A2_2.

[Rev2_9] Please refer to Rev2_2.

**A few other things related to orientations**

[R2_10] *In Appendix D there is a statement: "We assumed that the IACC consistently develops towards the same horizontal orientation within the ice sheet...". I agree that this is plausible, but I don't think you need to make this assumption. Fig C1b is in the measurement ref frame: the transformation of Δε' to Δε is simply a function of the vector orientation of the c-axis maximum relative the measurement reference frame. There should be no need to make any assumption about how this varies from one core piece to the next? Appendix C is easy to understand. (although line 801:"horizontally rotating the frame..." is not a good description: "rotating around the core axis to align the c-axis maximum within plane of the electric field vector" would be better). I really do not understand what Appendix D is saying. Is this about correcting measurements between different COF measurements??*

[A2_10] It seems to us that there is either some poor writing on our part or unsuccessful communication from the authors to the readers. In the ice sheet, there is a possibility that the inclination angle of the c-axis cluster may rotate with increasing depths if the dome summit position migrates in a complex manner. Although it seems unlikely, we cannot completely deny this possibility. Thus, we required this assumption. This is one aspect.

Another aspect is that when handling many cylindrical ice cores, sometimes the continuity of orientation between adjacent cores is lost, especially when irregular ice core breaks occur. When we perform the DTM measurements, we do not know the core orientation in advance. We can determine the orientation only from the Laue data (or orientation of layer inclination if they were recorded). Only with the assumptions we made were we able to apply corrections from Δε' to Δε.

[Rev2_10] Correction in terms of English expression was addressed in the revised paper, based on the suggestion by the reviewer. Also, we added the explanations above to the revised Appendix B, so that readers can better understand.

[R2_11] *It is unclear whether there are constraints independent of the core on the layering orientation in the DF hole. Are there televiewer data (Hubbard and Malone, 2013)? Are the radar data (Karlsson et al., 2018; Wang et al., 2023) good enough to extract the dip direction at the borehole site? What are the constraints, other than from the core, on the layer structure shown in cartoon form in fig 10. Knowing the dip direction could turn the data into 3D in a geographic reference frame.*

[A2_11] We have no independent data source to determine the orientation of the inclination angles. Additionally, we lack borehole-logging data to detect such orientations. Please note that the literature you provided deals with data from relatively coarse airborne survey lines (typically every 10 km), compared to our ground-based survey map in Figure 1c, which features very dense data within an area of 580 m x 580 m. Figure 1c provides the best geographic information on the subglacial topography at Dome Fuji.

[Rev2_11] As answered above.

**Eigenvector analysis outcomes: error related to the number of grains.**

[R2_12] Robustness of the eigenvalues, eigenvector orientation (Inclination angle of c-axis cluster) and median cone half angle (median inclination of c-axis cluster) from the c-axis measurements depends on the number of grains measured. None of the five deepest samples (as shown in fig 4) have enough grains to provide robust eigenvector measurements and points related to these (eigenvalue, Inclination angle of c-axis cluster, median inclination of c-axis cluster) should not be shown on figure 5 or any other figure. I wonder whether other samples with large grain size, particularly between 2860 and 2890 m also have too few grains to make the eigenvector analysis robust.

[A2_12] Thank you for your comment. At a depth of 2860m, where the largest grain size is about 3 cm², a single measurement of a microwave beam (with diameter of approximately 16φ) contains only a few grains. By scanning a 50-cm-long core, at least 50 grains are included in the data. This situation is similar even when we use thin-section methods. When the grain size is about 3 cm², a thin section with an area of 90mm x 50mm contains only about 16 grains. By analyzing a wider area, for example, 500mm x 50mm, we can measure about 80 grains. This situation of large grain size is common for analyzing the deep zones of the summit ice cores at Dome Fuji and EDC.

[Rev2_12] We added a table indicating grain numbers included in a Gaussian beam or in a thin section as Table A3

[R2_13] *Error analysis for eigenvectors is tricky - a simple way to do this is, is to recalculate the eigenvectors for different randomly selected subsets of the full data set. Since all of the c-axis patterns (where there are enough data) seem to be single maxima I suggest that you do this with one sample to give a representative value for errors of all the eigenvector derived numbers for all samples. Pick a data set with many grains (e.g. the data in fig 4 b). Compare the eigenvector results for ten randomly selected sets of n grains from this larger data set. Use n values from 10 to 200 (in increments of 10?) to get an idea of the errors for the parameters you show, as a function of the number of grains in the data set. Then all the parameters derived from eigenvector analysis can she shown with error bars corresponding to the number of grains measured.*

[A2_13] In our preprint, we did not perform a derivation of the error range for the COF data obtained through thin-section methods. Generally, if the total number of grain samples is N (in our Laue example, ranging from about 50 to about 250), the addition of a grain means the normalized eigenvalue can vary by about 1/N at maximum. When c-axes are clustered, as in the Dome Fuji core, the range of variation will be much smaller than this 1/N. Assuming each grain contributes an eigenvalue between 2/3 and 1 (see Figure 5a), the range of variation will be 1/(3N) or less. Therefore, the total number of grains is very important. The error in the eigenvalue is on the order of 1/(3N) or less.

[Rev2_13] The point here was added as Appendix A4.

*[R2_14] You should show all of the stereonets of c-axis and <a>-axis data (in an appendix?) and tabulate the measurements (e.g. eigenvalues) for all the samples. The paper refers to fabric analyser data as well as Laue data. Where are these data? They should be included.*

[A2_14] In revision, we plan to present all the data in Appendix or Supplementary information.

[Rev2_14] In revision, we presented it as Supplementary information. In the present paper, G50 fabric analyzer was used to obtain COF images (shown in Figures 6 and 7). Furthermore, measurement accuracy is not validated and lower than the Laue X-ray measurement. Thus, we used only Laue X-ray measurement data for the correction from $\Delta\varepsilon'$ to $\Delta\varepsilon$. In the revised manuscript, we emphasized that the G50 fabric analysed was used to obtain COF images (in Section 3.2).

**Elongate [c] axis patterns**

*[R2_15] The c-axis maxima in fig 5 are not all circularly symmetric. Subfigs a,b,d,e,g, and h are all elongate. You should highlight this observation. This is important because all laboratory experiments in shear give elongate c-axis maxima (Bouchez and Duval, 1982; Journaux et al., 2019; Kamb, 1972; Li et al., 2000; Qi et al., 2019; Wang et al., 2024) - with elongation perpendicular to the shear direction. The elongation is less clear in natural samples from shear zones: it is present in the Whillans shear margin (Jackson and Kamb, 1997) but absent in other studies (Disbrow-Monz et al., 2024; Monz et al., 2021; Thomas et al., 2021).*

[A2_15] We agree that it is important to comment on the elongated single pole fabric. As previously mentioned in papers by Azuma et al., 2000 and Fujita et al., 2006 (DOI:10.3189/172756506781828548), this phenomenon is observable even in dome summit areas of polar ice sheets, where ice does not flow perfectly radially from the dome but undergoes deviatoric strain depending on orientations. Such summit areas often feature ridge-like ice divides, which cause uneven strain along two principal orientations. Because this elongated single pole fabric is already observable at shallow depths, it should not necessarily be linked to the presence of simple shear. Simple shear is only one of the possible causes mainly in deeper zones; the phenomenon can be explained without it.

In a paper on polarimetric radar sounding at Dome Fuji by Fujita et al., 2006, the authors discuss the amount of radio wave birefringence caused by this elongated single pole at the Dome Fuji drilling site, up to depths of about 2200 m. They demonstrated that the orientations of the elongated single pole are consistent at least to this depth. Figure 1b in this paper by Fujita et al. (2006) shows two principal axes of the elongated fabric, inferred from polarimetric radar sounding. These axes are comparable to the bedrock topographic map shown in Figure 1c of our preprint. We observe that the two principal axes tend to align with the orientation of the subglacial slope (WNW) and its orthogonal direction. Below, we present two principal axes.

According to View 3 in Figure 4 of the preprint, in many cases, the elongation of the single pole includes the maximum inclination, from Figures 4a to 4h. At depths deeper than about 2860 m, the feature of elongated single pole is almost lost, which is another aspect we should comment on in the revised paper.

In summary, and importantly, the elongated single pole is not necessarily due to shear.

Additionally, we have reviewed the reference papers introduced by the reviewer and have created a table to present the experimental setup as follows.

[Figure]

Figure A2_15. On the Figure 1c in the preprint, two principal axes of elongated single pole fabric inferred from the polarimetric radar sounding (Fujita et al., 2006) was indicated.

Table A2_15: Summary of the key experimental conditions for studies introduced by the reviewer.

| Paper | Ice type | Type of deformation | Temperature | strain rate |
|---|---|---|---|---|
| Bouchez and Duval, 1982 | The polycrystalline samples obtained by filling a cylindrical container with snow, saturating the snow with previously boiled water | constant-stress torsion experiments | $-8 \sim -12$ °C | from $10^{-7}$ to $10^{-6}$ sec$^{-1}$ |
| Journaux et al., 2019 | Unstrained equiaxial polycrystalline ice prepared by evenly packing 200µm sieved ice particles in a mold | Torsion experiments | $-7 \pm 0.5$ °C | from $10^{-7}$ to $10^{-6}$ sec$^{-1}$ |
| Kamb, 1972 | Ice prepared by filling a container with fresh snow, saturating the snow with water, and allowing the water to freeze | Torsion experiments | $0 \sim -5$ °C | $\sim 10^{-7}$ sec$^{-1}$ |
| Li et al., 2000 | Polycrystalline with initially random crystal orientations and a mean crystal area of about 1.2 mm$^2$ | Torsion experiments | $-2.0$ °C | from $10^{-7}$ to $10^{-8}$ sec$^{-1}$ |
| Qi et al., 2019 | polycrystalline ice samples with a controlled initial microstructure | Shear experiments | $-5, -20$ and $-30$ °C | $\sim 10^{-5}$ sec$^{-1}$ |
| Wang et al., 2024 | synthetic ice (doped with ~1 vol.% graphite) | Equal-channel angular pressing technique | $-5$ °C | not specified |
| Jackson and Kamb, 1997 | The ice was obtained from a depth of 300 m in borehole, by means of a hot-water ice-core drill | unconfined uniaxial compression | $-22.0 \pm 0.1$°C | $\sim 10^{-8}$ sec$^{-1}$ |
| Disbrow-Monz et al., 2024 | Unavailable for access because it is in press | | | |
| Monz et al., 2021 | polythermal valley glacier ice close to melting point | | close to melting point | |
| Thomas et al., 2021 | A 58m long azimuthally oriented ice core collected from the floating lateral sinistral shear margin of the lower Priestley Glacier, Terra Nova Bay, Antarctica | | $0 \sim -20$ °C | |

In some of the suggested papers on laboratory experiments conducted at temperatures near the melting point, the numbers for strain rate are larger than those in the Antarctic ice sheet by about $10^5$ to $10^7$ times. Additionally, ice samples from glaciers also come from temperature conditions close to the melting point. Our concern is that to draw analogies with dome ice cores, obviously extensive discussions—and possibly even entire papers—are necessary. This complexity makes it difficult for us

to cite these papers in a straightforward manner without issuing cautions. It seems unlikely that ice deformation with strain rates differing by $10^5$ to $10^7$ times would result in the same ice fabric. Time-dependent factors must be argued extensively, to draw analogies with dome ice cores.

[Rev2_15] A summary of the explanations above was added in the Section 2, with addition of the reference papers about a few laboratory experiments.

[R2_16] *It would be good to know whether this is a parameter that changes between the UP80% and the LO20%. I cannot find any c-axis stereonets related to UP80% (Saruya et al., 2022) in although I note that samples 1720m and deeper in DF1 (Azuma et al., 1999) have elongate c axis clusters.*

[A2_16] Please refer to [A2_15] above. In the uppermost 2200 m, the principal axes of the ice fabric development are known in terms of the geographical orientation. However, we have no direct evidence that the principal axes in the uppermost 2200 m keep continuous orientation in the deeper depths.

[Rev2_16] As answered above.

**<a> axis data**

[R2_17] *The <a> axis data are not well used. They are, in part, hard to evaluate because the quality of figure 4 is poor (this figure needs to be better). You say in the text that the <a>- axes girdles have no maxima - I don't believe that, it looks to me like some of the girdles contain maxima. Fig 5 g, h and I all look to have maxima in the middle of the stereonets in the corrected view. Remember that the maximum value for any measure of <a> axis orientation density must be less than 1/3 the maximum value for the [c] axis maximum, so <a> axis patterns will always look more subtle than [c] axis patterns. The hexagonal repeat of <a>-axes renders an eigenvector analysis uninformative. You need to present <a>-axes in a contoured form to evaluate them. To compare <a> axis and [c] axis patterns they need to be contoured independently, so that both are scaled to a maximum value for that crystal direction.*

[A2_17] The author members engaged in discussions on this point before submitting the preprint. Our views are as follows: We have conducted calculations to create histograms of the orientation distribution along the a-axis for all samples. Here, we examined whether there are any dense or sparse features in the distribution of the a-axis by dividing the data into 18 sections at 10-degree intervals. The results show that, fundamentally, the ice's a-axes have some form of non-uniform distribution, and since there are three equivalent a-axes, a 60-degree periodicity inevitably appears. However, we recognized that there are no signs that this 60-degree periodicity systematically indicates any structure. In some samples (such as at a depth of 2730 m), the sparseness interspersed with the 60-degree periodicity results in what appears to be three peaks. However, the depths at which this occurs are random, and there is no correspondence to glacial or interglacial periods. Rather, we recognized that this seems to be more appropriately considered a case of random bias. Apart from the few cases mentioned, no structured 60-degree periodicity was observed, even at depths where recrystallization is predominant. Based on this understanding, the description in the current preprint was deemed appropriate. Highlighting the sample at 2730 m deep, as mentioned above, would impose a bias as if there were meaningful significance in what is merely a random variance element. We plan to add these points in Appendix of the revised paper because these will be one of interesting points.

[Rev2_17] After providing our comments above in May, we attempted further analysis for the *a*-axis fabric. Then, we discovered the phenomena described in the Abstract, conclusions and main text (with Figures 4 and 5). We realized that they are of great importance, and then developed discussions on organization of the *a*-axis fabric. We are very grateful that you brought this issue to our attention. Thanks to your input, the scope of the discussion on the topic of the paper has been greatly expanded.

[R2_18] *This is pretty important as a distinct <a> axis maximum would add strength to the inference that these are COFs related to shear. All shear experiments where <a>-axes have been measured have a preferred <a>-axis maximum (Journaux et al., 2019; Qi et al., 2019; Wang et al., 2024). Natural examples from sheared ice (Disbrow-Monz et al., 2024; Monz et al., 2021; Thomas et al., 2021) also have a preferred <a>-axis maximum, although in the case of Thomas et al it is variably developed.*

[A2_18] Please refer to [A2_17].

[Rev2_18] The condition of ice that the reviewer commented seem to agree with those we discussed in the revised section 5.1.5 "Variations in *a*-axis fabric". Under high temperature close to melting point and "very fast" deformation, dynamic recrystallization (nucleation and migration recrystallization) should take place significantly. Then, it is possible that the a-axis caused organization. We speculate that the mechanism might be common. Therefore, we added this idea in the main text of the paper, at the final paragraph of the section 5.1.5.

[R2_19] *If you do have a -axis maxima they could also be plotted on a figure like Fig R2 or Fig R3. How to do this would depend on the nature of the data, but I am certain that they would be useful.*

[A2_19] Please refer to [A2_17].

[Rev2_19] We conducted analysis and graph display as we show in Figures 4 and 5. Again, many thanks for stimulating discussions on the *a*-axis fabric.

[R2_20] *I think your Laue data are one point per grain? In this case there will be insufficient data to calculate crystal vorticity axes (Kruckenberg et al., 2019; Michels et al., 2015). It would be worth thinking about collecting full crystal orientation data at higher density (EBSD data) to enable crystal vorticity axis (CVA) analysis. Obviously I'm not suggesting you need to do this for this paper- an idea for the future. (Thomas et al., 2021) have very clear CVA maxima that are consistent with the dominant simple shear that is also constrained from other data- with the very strong single point maxima you have a CVA analysis may provide excellent constraint on deformation kinematics.*

[A2_20] Yes, Laue data are obtained by one point per grain. This method can determine the orientation with a resolution of less than 0.5 degree (Miyamoto et al., 2011), so that misorientation angle within a grain can be estimated. We would like to challenge CVA analysis in the future.

[Rev2_20] We added following sentence in Section 3.2:
*This method can determine the orientations of all axes of each crystal grain with accuracy of ~0.5 degree (Miyamoto et al., 2011).*

**Relevant comparative data from the literature.**

[R2_21] *A key part of the paper is about what might be controlling the COF, including kinematics mechanisms and conditions. Excellent constraints on these come from laboratory experiments and field studies where deformation kinematics and/or conditions are constrained. There is virtually no reference to this extensive literature. I have already related (in the previous two sections) a couple of your observations to the literature for COFs from experiments and kinematically constrained. This is sorely needed if we are to understand ice cores such as DF where there are no measured constraints on deformation kinematics (the thinning model is not a measured constraint- it is a model with imposed kinematic assumptions).*

[A2_21] Please refer to the latter half of [A2_15] and Table A2_15. To discuss the analogy between these examples and dome ice cores, obviously extensive discussions (even entire papers) are necessary. This complexity makes it difficult for us to cite these papers in a straightforward manner without numerous cautions. It seems unlikely that ice deformation with strain rates differing by $10^5$ to $10^7$ times would result in the same ice fabric.

[Rev2_21] As answered above.

[R2_22] *A good starting point is the fact that your c-axis patterns are very tight single maxima, with most of them close to vertical. This is common in palaeo-climate focused ice cores (Faria et al., 2014). Many of the papers that describe these data infer that the primary cause of the vertical tight c axis maximum is vertical uniaxial compression due to lattice rotation. This is entirely at odds with the experimental data. There are no uniaxial experiments published that have tight c-axis maxima parallel to compression. Virtually all uniaxial experiments (my list of refs is just a subset) have open cones (small circle distributions) with the cone axis parallel to compression (Budd and Jacka, 1989; Fan et al., 2020; Hunter et al., 2023; Jacka and LI, 2000; Jacka and Maccagnan, 1984; Montagnat et al., 2015; Qi et al., 2017; Vaughan et al., 2017). Single maxima can form in uniaxial compression at high stress corresponding to high strain rates and or low temperatures (Fan et al., 2020; Qi et al., 2017) but these are weak maxima, tight maxima never form under these conditions. Lower stresses, as expected in nature, would tend towards open cones. Tight single maxima are observed in experiments, but only in shear, where they form normal to the shear plane (Bouchez and Duval, 1982; Journaux et al., 2019; Kamb, 1972; Qi et al., 2019; Wang et al., 2024).*

[A2_22] Based on the comment by the reviewer, we checked all papers mentioned above.

Table A2_22: Summary of the key experimental conditions for studies introduced by the reviewer.

| Paper | Ice type | Type of deformation | Temperature | Strain rate | Type of ice fabric |
|---|---|---|---|---|---|
| Budd and Jacka, 1989 | Laboratory-prepared ice samples with initially consisted of randomly oriented crystals. | Uniaxial compression tests | –3.0°C | ~ $10^{-8}$ sec$^{-1}$ | Small circle girdle (= cone) |
| Fan et al., 2020 | Polycrystalline ice samples with a controlled initial microstructure | Uniaxial compression tests | –10, –20 and –30°C | ~$10^{-5}$ sec$^{-1}$ | At temperatures warmer than –20°C, CPO is a cone (i.e. small circle) around the compression axis. |
| Hunter et al., 2023 | Polycrystalline D2O ice prepared using the technique described by Wilson and others (2019). | Pure shear and simple shear | Mostly above –10°C | from $10^{-5}$ to $10^{-7}$ sec$^{-1}$ | Cone |

| Jacka and Li, 2000 | Laboratory prepared initially cylindrical (60 mm long; 25.4 mm diameter), with randomly oriented, small-grained crystal structure | The compression tests with constant stress | –10 to –45°C | from $10^{-7}$ to $10^{-8}$ sec$^{-1}$ | At higher temperatures and stresses, the dominant deformation mechanism is recrystallization, while at lower temperatures and stresses crystal rotation may be more important. |
|---|---|---|---|---|---|
| Jacka and Maccagnan, 1984 | Laboratory-prepared ice samples with initially consisted of randomly-oriented crystals. | Uniaxial compression tests | –3.0°C | ~ $10^{-8}$ sec$^{-1}$ | Small circle girdle (= cone) |
| Montagnat et al., 2015 | The ice polycrystals were made from sieved seeds within a controlled size range | Uniaxial compression tests | –5 or–7°C | more than $10^{-7}$ sec$^{-1}$ | cone |
| Qi et al., 2017 | Fabricated ice samples with starting average grain sizes of either 0.23 mm or 0.63 mm | axial compression | –10°C | from $10^{-4}$ to $10^{-6}$ sec$^{-1}$ | A transition from cone to cluster CPO pattern with increasing stress. |
| Vaughan et al., 2017 | Synthetic polycrystalline ice | Uniaxial compression | –5°C | $10^{-6}$ sec$^{-1}$ | cone |

We noticed that almost all of these cited laboratory experiments were performed under temperatures very close to the melting point (between –3.0°C and –10°C), and the strain rate is extremely faster compared to deformation in the inland ice sheet. There is no experimental setup listed that can reproduce the temperature conditions of the inland plateau (–50°C or lower). Jacka and Li (2000) have already stated in their abstract, '*The results support the conclusion that at higher temperatures and stresses, the dominant deformation mechanism is recrystallization, while at lower temperatures and stresses, crystal rotation may be more important.*' This view explains why under high temperatures, a cone can appear. The reviewer's comments are partly based on comparisons between these deformation tests close to the melting point and much colder polar ice. In this case, our view is that the comparison is not straightforward. If comparison is between these laboratory-based studies and the more temperate ice within the LO20%, we understand the phenomena as follows.

Once the cluster strength of the single pole fabric reaches its maximum at the bottom of the UP80%, the temperature of the ice is still approximately –20°C. Thus, empirical knowledge based on laboratory measurements (as listed above) is more directly related to temperature conditions at depths deeper than approximately 2700m, where the ice temperature is about –10°C. Recrystallization will create new grains favorable for the maximum shear stress of the vertical compression (cone with wide angles). The c-axes within the cone continuously rotate toward the vertical. Then, a cone-like fabric, or alternation from it, such as less clustered single pole, is reasonably formed. This is entirely in agreement with our observations of COF in that very zone.

We note that this kind of phenomenon was already mentioned by Budd and Jacka (1989). We cite this here.

*If single maximum fabric ice is subjected to sustained stress of some other configuration, e.g., compression (uniaxial or confined plane strain) in the direction of the single maximum, then the fabric can change, e.g., to spread and form girdles or multiple maxima (cf. Fig. 3(e)). This can occur in ice sheets, particularly near the base. The 'overprinting' of fabrics from different stress configurations in ice sheets is also similar to the results obtained from laboratory studies of combined shear and compression, such as the torsion experiments of Duval (1981), or the unconfined compression of Azuma and Higashi (1984).*

Along with the views of Jacka and Li (2000) as mentioned above, there is no contradiction if we consider the temperature range of natural phenomena in the ice sheet and the laboratory experiments. In revision, we plan to cite papers in the list explaining these laboratory data agree with the observation in very deep part of the ice sheet at Dome Fuji.

There is another point which we hope will draw the readers' attention. In polar ice sheets, at the bottom of the firn zone where bubble close-off ends, the c-axis is already clustered by metamorphism processes in the firn. This is documented in section 5.4 of the preprint. The normalized eigenvalues of the DF cores at ~100m depths (bubble close-off depth) are about 0.4 (Fujita et al., 2009, 2016). At NEEM Camp in Greenland, the normalized eigenvalues at bubble close-off depth are about 0.5. These high values, caused by the c-axes cluster, are the initial values for further deformation. Randomly oriented fabric is often used as an initial stage of ice in laboratory experiments. However, in polar ice sheets, COF with a vertically clustered tendency is the initial state for deformation. This means that the resultant ice fabric will be different from those developed from random orientation.

In summary, there are two major differences between the laboratory experiments and polar ice sheets. One aspect is physical conditions, including temperature and strain rate. The other is the initial state of COF for compression deformation (random or already clustered to some extent).

[Rev2_22] We must repeat the same comments as [Rev2_18]. We considered including this idea in the main text of the paper. However, we hesitated to do so because it would require a discussion of significant differences in experimental conditions and would further increase the overall volume of the paper. To make this paper concise was a quite big issue. We ask your understanding.

[R2_23] *I think that some general discussion as to why you have tight single maxima, with reference to the experimental literature is needed. I suspect it is the dominance of shear on a shallow shear plane at all depths. I don't think there are many direct measurements of deformation kinematics in deep ice boreholes: (Treverrow et al., 2015) show that shear strain rate (on shallow shear plane) is much vertical shortening at all depths in the Law Dome boreholes.*

[A2_23] We already provided our view on this point. When dislocation creep is dominant, tight single maxima can appear without the presence of simple shear. This view was already given for example, by Azuma and Higashi (1985) based on numerical simulation and Jacka and Li (2000). Recrystallization which occurs dominantly under temperatures near the melting point (typically above about –15 °C) is another problem.

[Rev2_23] As answered above.

[R2_24] *My knowledge of the geometry of c-axis maxima relative to shear kinematics together with some basic knowledge of structural geology leads me to suggest that he cartoon in figure 10c (if it is correct: see section on layering) requires a sense of shear opposite to what is alluded to by the authors. In shear, the c -axis maximum will remain perpendicular to the shear plane as seen in all ice experiments (Bouchez and Duval, 1982; Journaux et al., 2019; Kamb, 1972; Li et al., 2000; Qi et al., 2019; Wang et al., 2024), with no significant rotation as a function of shear strain. In shear, layering will rotate so that the pole to layering will be aligned with the short axis of the finite strain ellipsoid (Fossen and Cavalcante, 2017; Hudleston, 2015; Jennings and Hambrey, 2021). These two relationships, shown on the lev side of the Fig R4, are consistent with observations from ice shear zones (Hudleston, 1977; Thomas et al., 2021). On the right side of the figure I have rotated the picture to match the reference frame of fig 10c.*

[A2_24] Please look at the figure here and the explanations for it. We hope that this explanation makes sense for readers. The simple shear in principle contains components of compression, extension, and rigid-body rotation of the system. This is a firm basis. Progressive increase in simple shear with increasing depth is a key point.

[Figure]

Figure A2_24. 2D schematic explanation for configuration of strains and rotations in the ice body above the steep bedrock slope. Rectangular shape means a body of ice considered for deformation. Black thin arrows and blue bold arrows mean components of shear strains and normal strains. Bold red arrows indicate rigid body rotation of the system. Open red arrows indicate the axis toward which the cluster of the c-axes rotate. (a) A condition well above the bed. Uniaxial compression in the vertical dominates. (b) A condition under more influence from the bed topography. Because of the friction force between ice and bed, simple shear system rotates the entire system including the internal layers and the c-axes cluster together. However, because of the normal components of the strains (both compression in the near-vertical and extension near the horizontal plane), the all the c-axes thus the c-axes cluster rotates toward the vertical at the same time. This mechanism of the rotation in counter-direction works only for the c-axes and not to the layers. In this way, total amount of inclinations becomes larger for the internal layers than the cluster of the c-axes.

[Rev2_24] We added this figure and explanation to Figure 8.

**Microstructures and their interpretation**

[R2_25] *Section 4.5 is not too bad, although there are some unintelligible bits (It is noteworthy that the concentration of the less impure ice is markedly smaller than in impure ice??). The key issue in this section is to make clear in Fig. 6 what are grain boundaries and what are sub-grain boundaries. The arrows are not good enough. I would suggest that an additional column is added with the photo from column 3 repeated, but with an overlay with three different coloured lines for boundaries on the top side, boundaries on the bottom side and subgrain boundaries. The description of boundaries being concave towards complex subgrain boundaries is not correct, wishful thinking I suspect. In both examples cited in the text (a and b in Fig 6) there are both concave and convex boundary segments adjacent to the subgrain structures.*

[A2_25] Thank you for the comment. Yes, the impurities concentration of the less impure layer (shown in Fig.6) is very much smaller than that of the impure layer. We meant that the contrast or difference was quite large. Please look at the concentration of dust particles indicated in the main text. Additionally, we will attempt to make new figures illustrating grain boundaries and subgrain boundaries (next to micrographs or in Appendix). We plan to modify the statement to avoid readers' misunderstanding regarding the shape of grain and subgrain boundaries.

[Rev2_25] We added an illustration of GBs, GBs on the reverse side, and sGBs in Figure A3 in Appendix A5. Two panels correspond to the top-right panels of Figure 5a and 5b. These illustrations are just speculation based on the geometry because we did not measure misorientation angles.

[R2_26] *I think you need to locate the micrographs in column 3 of Fig 6. on the micrographs in a and b. This is important for the reader to assess whether boundaries are grain or subgrain boundaries. The only one I can locate myself is b. The "subgrain" structures highlighted in b are weird. I've never seen subgrains like this in any naturally or experimentally deformed ice. I have seen grain boundaries like this and this could be a grain boundary with c-axes closely aligned but a-axes misaligned by 10-15 degrees?*

[A2_26] Thank you for the comments. We will add squares in column 1 and/or 2 indicating the location of micrographs in column 3. Subgrain boundary in panel (b) is indeed curious. Since we can find microparticle (or subgrain boundary?) pinning, it would not be an artificial error. Unfortunately, we cannot estimate the misorientation angle around this subgrain boundary. It will be very interesting how this sugbrain boundary grows.

[Rev2_26] We added the location of microstructural images as white frames in the left column in Figure 6.

[R2_27] *Observations from sections 5.3.1 and 5.3.2. should be incorporated into the descriptions in section 4.5, so all the microstructural observations are together. Section 5.3.1 and 5.3.2. are both really poor: microstructural observations and interpretations are oven intermixed and many features are described in interpretive terms (e.g. "migrating boundary" used as a description). This is poor science as the paper omits clear microstructural observations (facts that will never be wrong) that enable future researchers to build new interpretations on these. Sorry I have run out of time to make my comments clearly structured so the following sections might be a bit of a mess. There is a lot to comment on as these microstructural sections need extensive re-evaluation and re-writing for both the observations and the interpretations.*

[A2_27] Thank you for the critical comments. We plan to combine Sections 5.3.1 and 5.3.2 with Section 4.5 and reduce the discussion. We agree that the 'observation' and 'interpretation' must be distinguished. We will be careful on that point. In the present study, microstructures are positioned as supporting information for the discussion of COF and layered structures.

[Rev2_27] We combined previous Sections 5.3.1 and 5.3.2 with Section 4.5 and reduced description. We removed interpretative words such as "migration boundary". As stated above, microstructures are positioned as supporting information for the discussion of COF and layered structures.

[R2_28] I fail, to see the "brick wall" patterns you describe. You need to show a lower magnification microstructure, with many more grains for the brick wall structure to be convincing: I doubt it will be from what I can see. None of the micrographs in Fig 6 or in Fig 12 look like Fig. 9 in Faria et al, 2009. (Weikusat et al., 2017) does not show "brick wall" patterns but infers their possibility based on very high grain elongation data: I can't see mean elongations that compare with Weikusat (fig 4c) or individual grain elongations that compare with Weikusat (fig 4) that justify a comparison. The other cited reference (Kuiper et al., 2020b) does not use the term "brick wall" (nor does (Kuiper et al., 2020a)- this is a better comparison to your data as it is the deep part of NEEM).

[A2_28] In the revised manuscript, we will redesign the example of the brick-wall pattern to provide clear features. Polarized pictures in Fig.6 are provided as a comprehensive image (with low magnification). As the reviewer noted, Weikusat et al. (2017) and Kuiper et al. (2020) observed elongated grains but did not use the term "brick-wall pattern". We will refer to EDML and NEEM ice cores as similar microstructures (elongated grains) in the impure layers. In the revised manuscript, we plan to include the aspect ratio of the short and long axis of a fitted ellipse data obtained through microstructural analyses. The aspect ratio in the impure layer ranges from 1.9 to 2.0, while in less-impure layers, it ranges from 1.5 to 1.7. In deep impure layers, it ranges from 1.5 to 1.6. Kuiper et al. (2020) reported that flattened grains in the glacial ice had an aspect ratio of about 2:1. This value is similar to our results.

[Rev2_28] In the revised manuscript, we avoided the use of a "brick-wall pattern" (used only referring to Faria et al., 2009). Instead, we used "grain elongation" or "elongated grains". Also, we redesigned Figure 7a to provide clear features of grain elongation. We added the description about the aspect ratio in Section 4.5.2 (the sentences above with modifications) and values in Table A4 in Appendix A5, and reference to the flattened grains with aspect ratio of about 2:1 in NEEM ice core.

[R2_29] *The inference of microshear processes based on the description provided is unjustified. What is the evidence for shear being localized? I can believe, as a general assumption, that it may have been localized but you show no clear evidence. A shape fabric does not mean localization unless it varies in orientation reflecting variable strain (Hudleston, 1977, 2015; Jennings and Hambrey, 2021; Ramsay, 1980). The grain shape fabric provides really useful information about deformation kinematics – you should use it. Grain long axes at 2648m (fig 6a) are ~ 60-70 degrees clockwise of the vertical on the picture. If the c-axis max is vertical in this pic, then this would imply shear with top to the right (Bouchez and Duval, 1982; Fossen and Cavalcante, 2017; Hudleston, 1977). The shape fabric is another thing that would be very usefully oriented relative to layering and the COF.*

[A2_29] We have introduced the term 'localized microshear,' citing features of the brick-wall pattern in the preprint. In this type of layered alignment of grain shapes, we can presume that shear is localized at chains of grain boundaries or at junctions between layered grain boundaries and those with different, for example, orthogonal angles. This seems reasonable to us, and we believe that our presumption is valid. In contrast, the reviewer's comment is that it is not justified because we did not

provide any clear evidence for the localization of shear. For us, chains of brick-wall-like aligned grain boundaries are at least circumstantial evidence. The last four lines that the reviewer provided are related to the presence of shear on larger spatial scales. Comparing ice features at a dome with laboratory measurements (Bouchez and Duval, 1982) is not straightforward because the laboratory measurements use extremely fast strain rates compared with ice at the dome. It can be one of the references, but it is not suitable to use it as a firm basis for discussions. In summary, our view is that chains of brick-wall-like aligned grain boundaries are at least circumstantial evidence, which we will present in the paper.

[Rev2_29] Our interpretations are as answered above. However, in the revised manuscript, we removed the detailed explanation about GBS via microshear because there is no evidence and that is not a focus in the present study. So, we removed the sentence about 'localized microshear' and other ambiguous words (e.g., GB chains, microshear bands..).

[R2_30] *The paper by Faria et al, 2009 predates a lot of work that relates to the GBS process in rocks and ice. Rock deformation studies report small recrystallized grains have CPOs that are randomly dispersed equivalents of the stronger parent grain COFs (Bestmann and Prior, 2003; Jiang et al., 2000; Warren and Hirth, 2006) (and many more recent papers). These observations were interpreted as the result of an increase in the contribution of GBS in fine grains. (Craw et al., 2018) and (Fan et al., 2020) reported similar observations in uniaxially deformed Antarctic ice and synthetic ice respectively, and the reduction of COF intensity in grains with finer sizes was attributed to GBS. (Fan et al., 2020; Qi et al., 2019; Wang et al., 2024) all infer that strength of COF in experiments is a competition between grain boundary migration (strengthening COF) and "rotation" processes, where those rotation processes include lattice rotation related to dislocation creep and recover and grain boundary sliding: following broadly ideas outlined in (Alley, 1992).*

[A2_30] In the present preprint, we did not compare the textural features of ice with the GBS process in rocks or laboratory-based ice deformation tests because such a comparison would not be straightforward. The reviewer introduced many reference papers about rock deformation studies that report small recrystallized grains have CPOs that are randomly dispersed equivalents of the stronger parent grain COFs. We believe that we need papers that compare rocks and ice sheet ice at the dome, including comparisons of the physical properties of each substance, phase diagrams, or deformation mechanism maps in terms of temperature and strain rate. We did not understand the latter half of the comment. The reviewer commented that 'the strength of the COF in experiments is a competition between grain boundary migration (strengthening COF) and "rotation" processes.' It seems to us that the rotation process also strengthens the COF. Indeed, we did not observe any small grains such as those the reviewer mentioned here. Additionally, when comparing the ice sheet and laboratory-based deformation tests, the latter have conditions with extremely high strain rates. It seems that making straightforward comparisons without a lot of caution is difficult.

[Rev2_30] Our interpretations are as answered above.

[R2_31] The listed controls on GBS are not the most important. The prime drivers of whether GBS is important are likely to be grain size and stress; if you scale grain size sensitivity from experimental data (Goldsby and Kohlstedt, 1997, 2001; Qi and Goldsby, 2021) to natural grain sizes and stress (Kuiper et al., 2020a; Kuiper et al., 2020b) then a significant GBS contribution is predicted. Impurities, in small volume fractions, likely have a secondary effect through restricting grain growth (Fan et al., 2023; Qi et al., 2018), keeping grain sizes small.

[A2_31] Thank you for the comment. Based on the literature, we agree that grain size is likely an important condition. We plan to rewrite the statement at L532-533 ('A combination of high impurity content, temperature, and moderate stress causes

GBS.') as '*A condition favorable for the occurrence of GBS is likely a combination of smaller grain size, the presence of significant stress, and higher temperature. Smaller grain size is often achieved by the presence of solid impurities such as dust particles.*' The reviewer also suggested a scale comparison between laboratory-based experiments and the Dome Fuji data. However, estimating the quantitative contribution of GBS would lengthen and complicate the discussion, and it is beyond the scope of the present study.

[Rev2_31] We modified the original statement as '*A condition favorable for the occurrence of GBS is likely a combination of smaller grain size, the presence of significant stress, and higher temperature. Smaller grain size is often achieved by the presence of solid impurities such as dust particles.*'

[R2_32] *Nucleation. We only know that nucleation must occur by analogy with experiments where the number of grains increase with strain (Fan et al., 2020) and in zones of localized deformation where grain size reduces (common in silicates and carbonates: few observations in ice). We know virtually nothing about nucleation: two possible mechanisms are proposed: sub grain rotation recrystallisation and bulging nucleation and inferring these processes are difficult (Craw et al., 2018; Fan et al., 2020; Urai et al., 1986). Spontaneous nucleation in random orientations has been suggested (Chauve et al., 2017; Falus et al., 2011) but the physics of this process remains unconvincing to me. The evidence from experiments is that grains with orientations where the shear stress is high on the basal plane will grow (Fan et al., 2020; Qi et al., 2017; Qi et al., 2019) during strain induced GBM at the expense of other grain orientations. The grains that grow are already established and their nucleation is irrelevant to this process. Identifying nucleated grains is very hard: we can't do it in a natural samples.*

[A2_32] What we know about the ice sheet is as follows: (i) Despite the older and deeper ice, the number of grains per volume of ice does not decrease significantly. (ii) The clustering of c-axes becomes less pronounced as we go deeper. Our view is that these phenomena can be explained by the nucleation of grains with angles well away from the major axis of the c-axes cluster. Of course, we are not observing a process where nucleation is just occurring. However, we cannot claim that we know virtually nothing about nucleation based on this condition alone. Two possible mechanisms proposed, subgrain rotation recrystallization and bulging nucleation, will not significantly change the COF. Nucleation with crystal orientations favorable for further dislocation creep has been suggested (e.g., Alley et al., 1992, JoG; Cuffey and Paterson, 2010). The reviewer's view is, 'The grains that grow are already established, and their nucleation is irrelevant to this process.' It seems to us that this view is in disagreement with our knowledge stated as (i). We agree that identifying nucleated or nucleating grains is very difficult. However, the fact of an increasing rate of grains with a wider angle of c-axes cluster itself suggests they are nucleated grains. It is likely that in the long history of the ice sheet ice, spanning up to about $10^5$ years, nucleation and subsequent grain growth at the expense of other grains with orientations unfavorable for the continuation of the deformation occur relatively rapidly without giving a chance for the observers to identify the moment of growth. In the revised manuscript, we plan to provide microstructure images indicating "grains with nucleation potential".

[Rev2_27] As explained above, we argued that the grain nucleation and subsequent grain growth is one of the possible causes of the variation and fluctuation of the COF in LO20%. Features of observed examples regarding grain nucleation are consistent with the examples of previous research (Faria et al. 2014b), however, there is no evidence of grain nucleation. So we provided microstructures images as "potentially nucleated grains".

[R2_33] *Lobate and highly irregular boundaries are the observation oven used to infer that grain boundary migration has been an important process (Rollett et al., 2017; Urai et al., 1986). The micrographs you present do not have strongly curved/ irregular/ lobate boundaries. Compare them for example with boundaries of grains in micrographs of the NEEM core*

*(Weikusat et al., 2017). I see no highly lobate boundaries that would suggest that strain induced grain boundary migration was a dominant process. The grain boundaries are slightly curved. This needs to be the basis for this discussion. Telling the past direction of grain boundary migration is fraught with difficulties (Jessell, 1986, 1987). You have not presented what the observations are that allow you to infer migration directions. I don't think these inferences are robust.*

[A2_33] We agree that the microstructural examples in the preprint are not sufficient to illustrate migration recrystallization. In the revised manuscript, we will redesign microstructure images to provide clearer examples (more lobate and irregular boundaries.). Additionally, we will remove the description about boundary migration direction. (The direction of boundary migration is not so important in this paper.)

[Rev2_33] We redesigned the microstructural images to provide clearer examples (more lobate and irregular boundaries) in Figure 7. Also, we removed the description about migration direction.

[R2_34] *The photos of particles on grain boundaries are very nice. I'm not sure they help particularly in the analysis- they are expected. They might be better in a short paper focused on that topic.*

[A2_34] In the present study, we provide this as an interesting microstructure, but a more detailed analysis will be carried out in the future.

[Rev2_34] As answered above, we provide this as a microstructure related to GB migration.

[R2_35] *A key set of conclusions relate processes that control microstructures and COF through the whole core, with emphasis on the difference of the bottom 20% from the upper 80%. I cannot find any microstructural descriptions or micrographs of the upper 80% in this or any other paper. Similarly, I can find no directly measured COFs (Fabric Analyser, Laue, EBSD) for the upper 80% that allows the inclusion of COF shapes in the discussion. The dielectric tensor data in (Saruya et al., 2022) reduces the proxy COF to eigenvectors and loses shape and symmetry information.*

[A2_35] Microstructures down to 2500 m have been analyzed by Azuma et al. (1999, 2000), but the DF1 ice core was used for these papers. Upper 80% of the DF2 ice core is not analyzed in terms of the thin-section-based methods. In Saruya et al. (2022b), we compared DF1 eigenvalues obtained from thin section measurements (Azuma et al., 1999, 2000) and our results from dielectric tensor measurement. Their results were approximately consistent. Therefore, we presume that the upper 80% of the DF2 ice core and DF1 ice core have the same COF. (DF1 and DF2 boreholes are 44 m apart).

[Rev2_35] As answered above.

[R2_36] *Section 5.4 is not very good. You cannot have dislocation creep without both dynamic recovery and dynamic recrystallisation mechanisms. At the high homologous temperatures throughout the ice core (TH >0.8) it is inconceivable to have no dynamic recrystallisation. The key factor is the relative contribution the sub processes that are all needed for dynamic recrystallisation to occur: sub-grain rotation recrystallisation (following from recovery and sub-grain rotation: see nomenclature in (Trimby et al., 1998)), bulge nucleation, and strain induced grain boundary migration. See interpretive sections of (Fan et al., 2020; Qi et al., 2017) for example.*

[A2_36] We agree that the description in Section 5.4 is not good enough. The contribution of dislocation creep and recrystallization could be changed continuously with increasing depth. We will emphasize that the development of $\Delta\varepsilon$ values changes significantly after around 2400 m (from approximately increasing to remaining high and large fluctuation). Deletion of Section 5.4 is another possibility.

[Rev2_36] We removed Section 5.4 in the preprint. Instead, we focused on the contrast between UP80% and LO20% in Section 5.3 and Table 3 with reference to temperature condition, strain configuration, influence of insoluble impurities, and influence of dynamic recrystallization on COF variation.

**Temperature**

[R2_37] *It is clear that temperature is an important parameter in controlling ice behaviour. Fig 5 needs temperature data from the hole and any figure covering the whole hole depth (e.g. fig 9) needs this as well. I see that there are measurements in (Motoyama et al., 2021) and broader modeling by (Obase et al., 2023).*

[A2_37] Temperature at deep sections increases almost linearly and representative borehole temperatures are shown in the upper side of Figure 9. We will add temperature profiles in appropriate figures.

[Rev2_37] We added temperature profiles in Figure 5h with borehole inclination data.

Finally, we wish to express once again our sincere gratitude to the reviewer for dedicating the reviewer's valuable time to review our manuscript and for providing critical insights aimed at its improvement.

---

## Referee Report (RR1)

**Review of *Development of crystal orientation fabric, microstructures and deformational regimes in the deep sections and overall layered structures of the Dome Fuji ice core, Antarctica***
By Saruya et al.

This paper presents detailed and potentially interesting measurements of microstructural properties of the deepest part of the most recent Dome Fuji Ice Core. It provides comprehensive measurements of crystal orientation fabric using DTM, Laue X-ray diffraction, and an automatic thin section analyzer. It also presents complementary information about layers and microstructures.

Particularly in the deep part of the core, the crystal orientation fabric and grains size vary over short depth intervals, and these variations correspond to impurity content and thus climatic interval (as shown in other cores). The authors argue that angle between c-axis clusters and layering is evidence of simple shear in the bottom 20% of the ice sheet (as expected). They draw parallels with fabric in other ice cores, particularly EDC and GRIP.

Overall, I think this might be a meaningful contribution after further major revisions. After finishing my own read, I looked through the original reviews; although I was not an original reviewer, I agree with most of their comments, and I find it disappointing that the same issues I identified were already brought up by the previous round of review but not addressed. I detail these below but give examples here. The manuscript is still too long and lacks focus. I would argue that this is not simply a matter of style—it is the authors job to distill their results into new insights, and this paper does not do that. Important points about the orientation of the structures in 3d from Dr. Prior were essentially dismissed (in response to about 5 paragraphs, all of which I found insightful, it seems the authors only added the sentence in A2_2). Other revisions introduced problems of their own because of the flippancy—for example, the section headers in the introduction do not match the text of the paragraphs since the structure was shoehorned on without corresponding changes to the body, which leads to confusion for the reader. Shortening this paper does not have to mean cutting content. Much of length is due to needless complex sentences, extraneous adjectives and adverbs, and a kind of personification where "comprehension" or "understanding" of phenomena are needlessly substituted for the phenomena themselves. Simplifying the language, and making it more precise in doing so, would go a long way toward letting the reader access the full range of ideas in the paper. Other opportunities for very easy shortening include the multiple places where figure captions are essentially repeated in the main text. I think that the final line of Dr. Montagnat's original review still describes the manuscript: "At the end, the novelty of the study stands only on the fact that these deep core measurements have never been published. Care should be taken in the discussion in order to focus on what is potentially new or relevant for the community."

I have additional scientific qualms as described in overall comments. I have not gone through and made an exhaustive list of detailed comments, since I think hope that structural issues will result in major changes to the text first. I am not an expert in microstructure the way that Drs. Montagnat and Prior are, so I do not focus on the rigor of methods of the observations, or their interpretation in terms of processes at the crystal scale. Instead, I focus on interpreting the results

in the ice-sheet setting, the implications for ice flow and radar, and the structure and writing of the paper.

**Overall comments:**
The paper lacks focus. While the data are undoubtedly complex and extensive, I strongly disagree that this means that the paper cannot tell a story or stories in the way that the response to review claims (and indeed The Cryosphere's evaluation form seems to agree with me). Without better structure and clearer writing, those in very close but not identical fields (including those like myself who have worked extensively on crystal orientation fabric from other perspectives) cannot glean meaningful insight from the work. Put differently, while I agree that "Over-simplifying this paper would compromise its scientific message," the authors have instead under-synthesized and thus obscure the scientific message to all but themselves or similar experts on ice-core microstructure. Part of the issue is contextualization within the literature as Dr. Montagnat noted before. For several of the conclusions, there are citations she mentioned that make identical statements—this makes it exceedingly difficult for somebody with adjacent expertise to understand the value of the present work. I cannot see how the paper really touches on deformation the way the title suggests, other than providing some (unsurprising) evidence of simple shear in the bottom of the ice sheet. I think the paper is currently inaccessible to most of the readership of The Cryosphere, largely due to its presentation not its inherent complexity.

Using the UP80% and LO20% framework leads to misinterpretation, convoluted language, and incorrect conclusions. Take the first conclusion labeled (i). It ends with "at the bottom of the UP80% and fluctuated in the LO 20%." This is not correct (the abstract really indicates that the fluctuations begin at 2650 m, as do the figures). It would be clearer and more precise to write "down to 87% of the ice thickness and fluctuated below." The zones are confusing, both here and elsewhere, since the results described (e.g., properties to 2650 m) cross zones. Overall, getting rid of these UP80% and LO20 acronyms would reduce confusion. This is particularly important for the abstract, which currently involves tortuous language to fit the results into this artificial framework.

The change to structure in response to the first round of review is window dressing; in the introduction, the section titles do not match the sections in the new version, so they create confusion rather than insight. The section titles in the introduction might be okay in principle, but they would need text to match. However, they would need to be continued in some meaningful way further into the paper. As is, the section headers in the introduction suggest broad implications that the paper fails to deliver upon. I see no major implications for ice-sheet dynamics from this work. I would be happy to be wrong, but the authors would have to provide a structured argument (i.e., a story) that shows why such interpretation is supported by the work.

The sentence-level errors are largely fixable during production, but some are substantive and need to be fixed before acceptance. This issue is most prominent in the introduction. Some sentences are imprecise to the point of being incorrect. It is not my job to rewrite them, but here is one example: "In these studies, the influence of anisotropy on the movement of ice aggregates is so substantial that comprehending its impact on the expansive flow patterns of ice sheets is essential" (L70-71). Despite the long sentence, the authors do not argue what this influence is essential for (maybe projecting sea level rise?), define what ice aggregates are (though I surmise

this means bulk ice?), or indicate why "comprehending" rather than just the impact itself is important. In this case, simplifying to "The anisotropy is sufficiently strong to affect bulk deformation" would be less wordy and remove imprecision. Again, I am not trying to say that the authors should use my version of this sentence, but ambiguities like this are not made precise in the production process, and there are enough of them that I cannot flag them all. Instead, I think some careful editing by the authors themselves, really focusing on what they can support with evidence (in the above example, that anisotropy influences bulk flow) and not opinion (in that example, that anisotropy is "essential" or that comprehension is necessary).

I find the response to the orientation issues identified by Dr. Prior in the previous version too flippant. As he pointed out, lying in the same vertical plane is different than saying closely aligned (i.e., I do not think the changes answer the issue identified in Figure R1 in his review). For examples of continuing issues involving angles, in section 3.1.2, I had to re-read 3 times to understand how the principal-frame permittivity was identified. While I think I now understand what was done, I do not see why it was necessary to use other measurements of c-axis orientation to do the rotation if the plane matches the normal to the layers. In this vein, I do not understand why Figure 3 is worth plotting—are not these data essentially meaningless without knowing how the section was cut relative to the azimuth of the cluster maximum? Perhaps this just requires methods clarification to say how the thick sections were cut, if it was done in such a way that gives these measurements meaning, but if I understand Rev2_10 correctly then the correction happens after the DTM measurement. Dr. Prior provided great suggestions for plotting the layer normal (which he called the pole) on the same stereonet as the c axes—I really want to see these on Figure 4. They do not match the clusters very well (Fig 5f), and the reader has no idea how well they align from what is essentially a single sentence added in response to the original comment.

Observations and interpretations are still mixed. Section 4.5.3 is the clearest example—each of the paragraphs mixes observation and interpretation. Section 5.2.1 is results.

The use of the phrase "statistical significance" in the paper is incorrect. At times, I think I know what it means, while at others I do not even have a guess. Usually statistical significance means an ability to distinguish between possibilities above a pre-specified level of confidence. The paper essentially uses it to mean "precise," but with no quantification. This is unacceptable. We need clear definitions whenever the term is used. As a hypothetical, this could be "the difference between measured c-axes distributions at 2500 and 2600 m is statistically significant ($p=0.001$)" or something similar. Including the confidence and the alternatives considered is required for proper usage.

I do not think the authors' interpretation of the relationship between c axes and layer orientation near the bed is correct (line 511, Figure 8c). Layer slape is not solely caused by rotation. For example, this is my 30-second version in Illustrator—after rotation, subsequent pure shear in the vertical *does* change layer slope:

Undeformed

Rigid rotation by 30 degrees

Rigid rotation by 30 degrees then pure shear of 0.5

[Figure]

[Figure]

The language is too muddy for me to understand if this is causing misinterpretation. Overall, no explanation that treats both layers and c axes as passive tracers can cause this deviation. Instead, this could be easily explained a result of intracrystalline slip. Particularly in the conclusions (L830), it sounds instead like the authors disagree with the cartoon version above, and instead are imagining that the pure shear has no effect on slope. Adding in the actual mechanism of c-axis rotation would help.

The conclusions are excessive, variously going beyond the results, repeating themselves, and restating ideas that have long been known. I do not really understand how the first (i) and (ii) differ. Both of these conclusions look nearly identical to other locations (e.g., EDC, (Durand et al., 2009)), and given the emphasis on comparison between sites this should be cited. The second (i) is not supported by the results—at most there is evidence that right at domes climate controls the fabric, but this paper shows nothing about how that might look even 5 km off the domes. Similarly, I do not buy the second conclusion (v); the work shows very little about how large-scale deformation is taking place. The first (vii) is very similar to claims from work at EDML (Weikusat et al., 2017), and the idea was originally suggested in the 90s (Castelnau et al., 1996). While there is nothing wrong with concluding the same thing as previous work, the reader needs context. A much shorter conclusion section, focusing synthesizing rather than repeating key results and contextualizing them rather than providing so much detail, would allow the reader to see how this work fits into the literature.

**Detailed comments:**
Title: The title does not really make sense. I cannot tell if the overall layered structures are a wholly separate thing considered at all depths or whether the crystal orientation fabric is within them.

Abstract: The abstract is too long and detailed, to point that I had to read it three times to figure out what the manuscript was showing. The paper would greatly benefit from shortening the abstract to meet normal length standards (i.e., 250 words, or at least ~300 as it was before, rather than 450). As discussed above, this does not necessarily mean cutting content. For example, it would benefit both accuracy and word count to delete the first words of the abstract "An in-depth examination of."

L24,184: Statistical significance cannot stand on its own like this. I assume that the authors mean that they have measured enough grains for the results to approximate the underlying c-axis

distribution with some amount of precision (perhaps that it differs from isotropy, but I truly do not know). But a measurement itself cannot be statistically significant in the sense used here.

L35: "dislocation creep is the primary deformation mechanism in polar ice sheets" cannot be concluded from this work. There is no reason to think that these results can be extrapolated to the ice sheets writ large.

L45-50: This paragraph has multiple logical jumps to try to get from the ultra-broad (sea-level rise) to the ultra-specific (dynamic layer structure). Rather than filling all these gaps, I suggest starting less broad, since the readership of the cryosphere surely knows what ice sheets are and that they contribute to sea-level rise. I recognize that this suggestion is stylistic as well as substantive, but if not heeded then we need the gaps filled in (e.g., how is dynamic layering a process challenge that affects models, why is continuous improvement in models needed rather than one really big improvement, how does a single 11-year-old reference indicate ongoing concern).

L68: Abbassi is not a good reference for SPICEcore fabric, they measured fabric loosely from IceCube. Voigt 2017 is the correct reference (https://doi.org/10.15784/601057).

L70: The correct reference for the full EastGRIP fabric is Stoll et al., 2024 (10.5194/egusphere-2024-2653)

L81-85: references needed

L220-221: How can a plane and a vector be in the same plane unless the vector is in the plane? I think this probably means that the normal to the layer is in the same vertical plane as the COF cluster?

L223: What are non-principal components? The components in an arbitrary xyz reference frame? I have not seen this terminology before and think it needs to be defined.

L227: Still confused since I do not know what these are, but are they really "adjusted" or are they rotated in some way (thus changing the value)?

L276-287: This is a repetition of the figure caption, not results.

L436-437: This sentence needs to be re-written. It is unclear if this means temporal changes or time differences that result from the depth-age relationship of ice sheets.

L497-501: This is a repetition of a figure caption, far from the original appearance, not discussion

L648-655: This discussion of twinning is convoluted. Starting with a simple definition would help.

L725: I do not think this really differs from Durand—it is a different location, and there may simply be regional or local differences

L774-778: The implications for radioglaciology need to be clarified. The idea that recrystallization is important near the bed is not new, nor is the evidence of simple shear which also been noted previously near the bed at ice-core sites (for example, it is suggested at EDC by Durand).

Figure 1: Confusing to use red and blue on contours in c when they do not match the colorbar in b. If two colors are needed, it should be explained in the caption (and the colors should not be red/blue, or why not just use a single color?).

Figure 8f: The label is confusing. The LO20% spans almost 90% of the thickness.

---

## Referee Report (RR2)

Review of the manuscript:

**Development of crystal orientation fabric, microstructures and deformational regimes in the deep sections and overall layered structures of the Dome Fuji ice core, Antarctica**

Authors: Tomotaka Saruya, Atsushi Miyamoto, Shuji Fujita, Kumiko Goto-Azuma, Motohiro Hirabayashi, Akira Hori, Makoto Igarashi, Yoshinori Iizuka, Takao Kameda, Hiroshi Ohno, Wataru Shigeyama, Shun Tsutaki

Revised version submitted to *The Cryosphere*
* * *
This manuscript presents detailed data on fabric evolution and microstructures in the 3035 m long DF2 ice core from East Antarctica, drilled at the Dome Fuji station. In earlier work the same research group published similar data from the uppermost 2400 m of the core, but here the focus is mainly on the depth interval 2400-3035 m. An overview is, however, also given of the evolution of the above mentioned parameters throughout this deep ice core.

A major scientific effort lies behind the results presented and the data are of great importance for studies of the internal structure and deformation properties of large ice sheets. Both state-of-the-art methods and new techniques are used to measure and characterize fabric and grain structure at relatively high resolution down the core. Methods are mostly well described and sources of error are adequately adressed in the paper.

Although some assertions made by the authors will be met with criticism – in spite of improvements made from the initial version – the importance and detailed description of the data is such that the paper should be published in *The Cryosphere*. This reviewer does not have specific criticisms, but asks the authors to take into account the comments and corrections suggested below.
* * *
Saruya et al. focus on main processes believed to control the development of fabric in ice under the conditions prevailing at Dome F:

- C-axis rotation under compressive stress leading to the development of a single pole fabric
- Rotation recrystallization and subsequent splitting of a grain into 2 or more grains
- The role of simple shear in the fabric evolution
- The interplay between inclined layering (from horizontal, due to bedrock topography) and the mean direction of the single pole fabric
- The effect of impurities and grain size on c-axis rotation
- Nucleation of new grains with c-axis orientations differing from that of surrounding older grains, at relatively high temperatures in the lowest 20% of the ice sheet

A clear picture emerges on the development of c-axis orientations in the DF2 core and interesting data on a-axis orientations are presented as well.

Perhaps surprising is the fact that this paper (as well as the companion paper on the upper 80% of the ice core, Saruya et al., 2022), does not discuss the grain size development in the core in any detail. Grain growth receives fleeting mention and some insight can be gleaned from Figures 5-7 and Figure 10, but no information is given on normal grain growth rates. If the authors aim to treat this topic in another publication, this should at least be stated, since studies of crystal size and crystal orientation are closely intertwined.

This reviewer has access to two reviews on a previously submitted version of this manuscript, as well as to the replies given by the authors. It seems to me that the authors have made a major effort in responding to the major criticisms offered by the two reviewers, both of whom are leading experts in this field. Examples:

Reviewer 1 criticised the authors´ emphasis on grain boundary sliding (GBS) in the earlier version, stating that the evidence for this deformation mechanism in the core was very limited. The authors have responded to this by significantly toning down the potential occurrence of GBS in the revised version.

Reviewer 1 is critical of the authors´ interpretation of small grains on the boundary between two larger grains as evidence for nucleation of new grains (sometimes referred to as dynamic or migration recrystallization) – see Figure 7. It also seems like Reviewer 2 does not support the authors´ interpretations on this issue. Author responses are detailed and they defend their view by pointing out that observed c-axis fabric changes at depth support the assertion that new grains are being nucleated.

It may be added here that it is surprising how little discussion there is in the paper on the effect of higher temperatures near the bed on grain size development. Section 4.4 mentions very large crystals below 2950 m depth (blue curve in Figure 5), but there is no further interpretation. Grain size curves between DF2 and EDC are compared in Figure 10, but a comparison with the Vostok core – where very large crystals were observed in the lowest part of the meteoric ice – would have been particularly interesting here.

Even though the authors have made efforts to shorten the text (following reviewer advice), the paper is still very long and the text is often repetitive. Actually, the repetitions are at times helpful for the reader, who gets a bit lost at times in the details being discussed, and is thankful for a recap! The Conclusions section (2.5 pages) presents a very good overview of main results and interpretations, in a better way than the bulk of the text.

The English language of the paper still needs improvement and this reviewer typed down suggestions as he read the paper through. Hopefully these suggestions can be of help in the final revision process.

**Specific comments:**

Comments/questions directed to the authors are blue-coloured for clarity. Note that the terms grain and crystal are used synonymously by this reviewer. Numbers like L20 refer to line numbers in the manuscript version:

**egusphere-2023-3146-manuscript-version2.pdf**

L20

To examine the distribution and texture of the $c$- and $a$-axes...

[Figure]

To examine the orientation distribution of the c- and a-axes...

(„texture" refers more to grain size and shape)

L22

detailed crystal grain orientations

detailed crystal orientations

L24

as eigenvalues

→

as eigenvalues of the orientation tensor

L28

Below 1800 m in the UP80%, layers with varying dusty impurities

→

Below 1800 m in the UP80%, layers with varying concentrations of dusty impurities

L38-39

What are c-axis layers ??

L53-54

whereas shearing on alternate slip systems is significantly more difficult, nearly a hundredfold

→ (suggestion)

whereas nearly a hundred times greater (shear) stress must be applied to induce deformation along other slip systems within the crystals

L57-58

The study of anisotropic ice deformation has been conducted historically. They were done through theoretical research

[Figure]

Anisotropic ice deformation has been studied theoretically

L59
and through laboratory experiments

and in laboratory experiments

L61-62
These laboratory-based studies are characterized by experimental setting of strain rate by more than several orders of magnitude, and under temperature close to melting point,

[Figure]

The experimental setting in these laboratory-based studies is characterized by strain rates that are several orders of magnitude higher than *in situ* strain rates in ice sheets, and at temperatures close to the melting point,

L62-64
Therefore, these laboratory-based knowledge can be a valuable reference mainly for such conditions.

[Figure]

The laboratory results are thus, strictly speaking, not directly applicable to ice-sheet conditions.

L64
play another essential role. Examples of Antarctic deep ice core include

play an essential role. Examples of Antarctic deep ice cores include

L68-69
Examples of Greenland ice sheet include

[Figure]

Examples from the Greenland ice sheet include

L70
and among others

and other cores

L72-74
For instance, during the internal deformation of the ice sheet, the flow forms preferred orientations of crystal axes, at the same time, the ice flow is modulated based on them.

[Figure]

For instance, the internal deformation of an ice sheet leads to the development of preferred orientations of the ice crystals, which in turn influence the flow of the ice sheet.

L83-84
dome regions in central area of the ice sheets

→
dome regions in the central parts of ice sheets

L95
was presented by
→
was described by

L94
and it is suitable to investigate
→
and such a setting is suitable to investigate

L98
deepmost → deepest

L104-105
composition of the deepest 1% thickness (3000–3035 m).
→
Composition of the deepest 1% of the ice sheet/ice core (3000-3035 m).

L105-106
These properties were found to retain the basic layered structure of ice core signals except in the deepest few meters.
Not certain what this sentence means – was the size and concentration of air hydrates and the oxygen isotope ratio typical for the meteoric ice above (except in the deepest few meters) ?

L108
sampled at every 11 m depth. → sampled at 11 m depth intervals.

L111-113
Dynamic recrystallization within ice sheet has been widely investigated historically, as reviewed in papers or textbooks (e.g., Poirier 1985; Humphreys and Haterly 2004; Faria et al., 2014a, b) or individual papers (e.g., De La Chapelle et al. 1998; Weikusat et al. 2009, Kipfstuhl et al. 2009, Montagnat et al. 2012, 2014; Stoll et al., 2021a).
→ (suggest change to)
Studies on dynamic recrystallization in ice sheets have been reported by e.g. Poirier 1985; De La Chapelle et al. 1998; Humphreys and Haterly 2004; Weikusat et al. 2009; Kipfstuhl et al. 2009; Montagnat et al. 2012, 2014; Faria et al., 2014a and Stoll et al., 2021a.

L113-114
Exploring deep ice rheology and crystal properties, dynamic recrystallization is one of main focuses in this paper.

[Figure]
 (suggestion)

In this paper, we take advantage of the unique opportunity that the Dome F ice core offers to study the role of dynamic recrystallization in the formation of textures and fabrics in the deeper parts of the East Antarctic ice sheet.

L115
What does „Advancing ice sheet dynamics" mean here? Would it not be more suitable to shorten the title of this section to: The development and role of crystal orientation fabrics

L117
„layer structures of the c-axis fabric" is not very clear wording.

L131
variations in the deformational history of the vertical compression.
→
variations in the deformational history of an ice-sheet region where vertical compression is the dominant stress regime.

L135
across wide area of ice sheets. → across wide areas of ice sheets.

L141-143
Following this previous study, research focusing on the lowermost (deepest) approximately 20% thickness zone (ranging from about 2400 m to the deepest ice at 3035 m) remains to be done, to examine textural data for the entire thickness of the ice sheet.
→ (suggestion)
Following this previous study, research has focused on the lowermost (deepest) approximately 20% of the DF core (depth interval 2400-3035 m), to complete the picture of textural and fabric evolution in the entire ice sheet column.

L147
We compare textural data by these methods with various data analyzed from the ice core.
→
We compare textural data obtained with these methods with various data sets from studies of the ice core.

L148-149 (twice)
crystal orientation fabric → the crystal orientation fabric

L153
that correspond to ages of more than 1 million years
→
covering snow deposition over more than 1 million years

L155
extends → extend

L157
knowledge of englacial layers under various ice conditions.
→ (suggestion)
knowledge of the physical properties and the deformation history in different regions of the East Antarctic ice sheet.

L167-168
In terms of the c-axis fabric, Azuma et al. (1999, 2000) reported that at DF1 the c-axis fabric exhibited the elongated single pole fabric as the dome undergoes deviatoric strain depending on orientations.
→
This is not very clearly worded – what does „depending on orientations" mean here ? Are the authors referring to the mean c-axis orientation being perpendicular to the bedrock slope (line 172) ?

L191
signals of dynamic recrystallization
→
signs of dynamic recrystallization

L195
The principle of...have been described
→
The principles of...have been described
Or
The principle of...has been described

L219-220
deviating from the vertical (hereinafter, inclination angle) in the same direction of the maximum layer slope.
→ (suggestion)
deviating from the vertical (hereafter, inclination angle) in the same direction as the normal to the maximum layer slope.

L266-267
Data from both cases....in agreement
→
Data from both cases....are in agreement

L302 (check also L304 and L306)
generally have periodicity of 60 degrees

→(suggestion)
generally display maxima at 60 degree intervals
(„periodicity" normally refers to regular variation with time)

L314-315, this sentence needs improvement/rewriting to become intelligible:
„It is also very important to note that the SDρ is well synchronized with the grain size (Figure 5e), which implies underlining Physics."

L323-324 (Fig. 4 caption)
The vertical orientation aligns with the core axis.
According to Fig. 5 the borehole is inclined 3-6° in the depth interval being considered so the core axis hardly represents the vertical.

L351-360
The authors outline grain sizes in the entire core, mentioning that individual grains become „extremely large" below 2950. No figures are given, though, and Fig. 10e seems to indicate grain size peaks exceeding the maximum value given on the vertical axis of that diagram. The limited size of thin sections is, of course, an issue at these great depths, but it would be good if the authors could at least mention maximum grain sizes inferred from the study.

L361-364
Supporting references are needed for the statement that the c-axis (mean) angle will not coincide with the normal to the shear plane.

L381-382 – these two sentences need English-language improvement.

L382
These features were unique in the impurity-rich depths

[Figure]

These features were present in the impurity-rich layers

L386
tend to be distributed as more straight lines

are often displayed as straight lines

L388-389
However, some features of impurity-rich layers observed in panels (a) and (c) are persistently present in (e). That is, flattened grains have slanting features.
→ (suggestion)
However, some features of impurity-rich layers observed in panels (a) and (c) are persistently present in (e), where flattened grains display slanting features.

L391-394 – improve English language

L419
reported that → reported

L435-436
We suggest that the steady grain sizes in the impurity-rich layers established themselves after these layers had reached deeper depths.
→
We suggest that the steady grain sizes in the impurity-rich layers were attained when these layers had reached greater depths.

L484-485
The conditions of ice sheets in Antarctica, in terms of temperature and stress, are located on a boundary zone between dislocation and diffusional creep on the deformation mechanism map
→ (suggested rewording)
Due to prevailing temperature and stress conditions, the ice within the Antarctic ice sheet(s) is located in a boundary zone between dislocation and diffusional creep on the deformation mechanism map.

L486
When ice is under temperatures → When ice is at temperatures

L497-504
This description of Fig. 5 has already been given earlier on.

L506-507
the consistency of the inclination angles in the visual layers and the c-axes cluster depends on glacial/interglacial periods.
→
The consistency between the inclination angles of the (normal to) the visual layers and the c-axis cluster is affected by varying conditions between ice deposited during glacial and interglacial periods.

L525-533
Are there any results from surface-based GPS-measurements, that could confirm the assumption that ice flow has occurred towards the center of the trough?
And since there is a trough there, wouldn´t the internal layers of ice sitting in the trough also be convex-shaped in the absence of basal mass loss due to melting?

L546 – improvement in wording required:
the all the c-axes thus the c-axes cluster rotates

L547

total amount of inclinations

total inclination

L558

are plotted in entire depths figure (Figure 10).

[Figure]

are plotted for the entire depth of the core in Figure 10.

L566-567

dust particles located → dust particles are located

L568

„Production of dislocation is one of the possibilities.“

Does this mean that the presence of microparticles leads to dislocation production? Please clarify!

L570-572

Item (ii) – it is unclear how the content of this sentence relates to the issue mentioned in L569, that microparticles may act as a sink of dislocations like grain boundaries.

L579

In the LO20%, some thickness between 2400 m and 2650 m is

[Figure]

In the LO20%, the depth interval 2400-2650 m is

L601

from compressional axis → from the compressional axis

L602

„However, it is also noteworthy that, in Figure 5c, distribution of c-axis density approximately from 30 degrees from compressional axis (60 degrees from horizon) is always denser than 45 degrees or 60 degrees.“

Wording is not clear here. Does the sentence imply that there are more c-axes located approx. 30° from the compressional axis that there are at 45° and 60° (or in between) ?

L603-604

„It is possible that it has some underlying mechanisms in terms of nucleation recrystallization relative to the existing $c$-axis cluster.“

What does „it“ refer to here? The sentence is unclear.

L608

ice crystals will recover the c-axis orientations available for

→
ice crystals will develop c-axis orientations favourable for

L612
decreasing → decrease

L613
in the impurity-rich layer → in the impurity-rich layers

L643-644
Be more specific on why rotation recrystallization has a minimal effect on c-axis fabrics and why you assume – seemingly ad hoc – that it does not cause a-axis organization.

L714-715
the thickness of ice is approximately 10% of the original ice equivalent thickness at the time of deposition
→
the thickness of a particular ice layer is approximately 10% of the ice equivalent thickness of that layer at the time of deposition

L726
strengthening the → strengthening of the

Table 2
The ice thickness at GRIP was 3029 m (the figure 3085 m is the thickness at NGRIP).
The mean annual temperature at GRIP was -32°C during the 1990s, when the deep ice core was drilled. See Dansgaard et al. (1993). Evidence for general instability of past climate from a 250-kyr ice-core record. Nature, 1993, 364 (6434), 218-220.

L746
Based on the above discussions, we propose an overview of the layered structure of ice sheets.
To "propose an overview" sounds a bit strange.

L759-760
What is meant here by „two distinct conditions? The next sentence mentions „Four types of conditions" which adds to the confusion.

L766-767
However, these layered structures correlate directly with the vertical thinning of each layer,
Which „layered structures" and how do they „correlate directly" with the layer thinning? Are the authors referring to the effects of changing grain size, evolving fabric and variations in impurity content on the thinning rate?

L775

layers of ice c-axis fabric → ice layers with a particular c-axis fabric

L794-95

„changes in depth range" almost sounds like a statement that depth increases with depth!
The sentence is probably supposed to refer to fabric changes with increasing depth.

L807

The term „Growth of c-axis clustering" appears several times in the paper. The authors should consider if „evolution" or „development" is not a more appropriate term than „growth" in this context.

L810

The words „are observed" need to be inserted after the word „recrystallization", so that the sentence becomes meaningful.

L820

Possibly, crystal twinning,which share some crystal lattice points symmetrically with the neighboring crystals

[Figure]

Possibly, crystal twinning, which leads to crystals sharing some lattice points symmetrically with neighbouring crystals

L829

and layers → and the normal to the layers

L848-849

We argue that the bottom thickness of the ice sheet deeper than approximately 2600 m plays a special role in shear deformation

[Figure]

We argue that the lowermost part of of the ice column, deeper than approximately 2600 m plays a special role in shear deformation

L852-853

Ice core drilling, aiming the ancient climatic records, at locations away from the dome area carries serious risks of layer disturbances near the bottom.

[Figure]

Ice core drilling projects aiming to retrieve climate records from ancient ice may encounter layer disturbances near the bottom if the drilling sites are located away from dome summit regions.

L929

within plane → within the plane

L938 (and similarly in L922):
Here, the c-axes cluster of single pole fabric is inclined to an arbitrary horizontal orientation.
Shouldn´t this rather be (?):
Here, the plane perpendicular to the c-axes cluster of the single pole fabric is inclined to an arbitrary horizontal orientation.

Section A3
The faint and thin cloudy layers are identified when oriented vertically;
What does it mean here to be „oriented vertically? Are the cores turned and inclined until the observer looks straight down ("vertically") on the edge of the cloudy layers?

L973-974
the aspect ratio value defined as the ratio of the short and long axis of a fitted ellipse
→
the aspect ratio value defined as the ratio of the long and short axis of a fitted ellipse [i.e. long divided by short]

Table A4, column farthest to the left:
Average → Average aspect ratio

L1076-1077
Crystal orientation fabric data from the DF core and the EDC core are compared using a common age scale of the DF2021 age scale.
→ (suggestion)
Crystal orientation fabric data from the DF core and the EDC core compared using a common age scale, matching the EDC age scale with the DF2021 age scale.
Did the authors use isotope data or methane data to tie the age scales of the two cores together? If so, this should be stated.

---

## Author Response (AR2)

Dear Reviewers and Editors,

We sincerely appreciate your efforts in handling and reviewing our manuscript.

We have revised the manuscript according to the comments and suggestions from reviewers. The English language was checked by a native speaker of English.

The revised manuscript has been uploaded with a tracked changes version. Supplementary Information files have also uploaded.

Throughout the revised manuscript, we attempted to shorten the main text, to remove the repetition, to make stories simpler, and to improve English language. We focused on the development and fluctuation of the crystal orientation fabric at the deep sections in the DF ice core. The order of discussion was accordingly changed, and new figures were added. In addition, discussions about grain sizes and small-scale fluctuation of the fabric were added in Section 5.2 (and Figure 6) and as Section 5.4, respectively. Previous Appendices were moved to Supplementary Information and Microstructure Section was moved to Appendix.

We thank for the many comments. With the reduction of sections/paragraphs, the sentences corresponding to some comments have also been removed.

With many thanks and kind regards,
Tomotaka Saruya, Atsushi Miyamoto, and Shuji Fujita, on behalf of all authors

**How the authors addressed the comments by Reviewer 3 in their responses to the revised manuscript**

In this document, we explain how we addressed the comments provided by Reviewer 3 on the revised manuscript.

**Explanation of the text colors:**

➢ Black text: Comments from Reviewer 3

➢ Brown text: Explanations of how the authors addressed Reviewer 3's comments in the revised manuscript

Review of Development of crystal orientation fabric, microstructures and deformational regimes in the deep sections and overall layered structures of the Dome Fuji ice core, Antarctica
By Saruya et al.

This paper presents detailed and potentially interesting measurements of microstructural properties of the deepest part of the most recent Dome Fuji Ice Core. It provides comprehensive measurements of crystal orientation fabric using DTM, Laue X-ray diffraction, and an automatic thin section analyzer. It also presents complementary information about layers and microstructures.

Particularly in the deep part of the core, the crystal orientation fabric and grains size vary over short depth intervals, and these variations correspond to impurity content and thus climatic interval (as shown in other cores). The authors argue that angle between c-axis clusters and layering is evidence of simple shear in the bottom 20% of the ice sheet (as expected). They draw parallels with fabric in other ice cores, particularly EDC and GRIP.

Overall, I think this might be a meaningful contribution after further major revisions. After finishing my own read, I looked through the original reviews; although I was not an original reviewer, I agree with most of their comments, and I find it disappointing that the same issues I identified were already brought up by the previous round of review but not addressed. I detail these below but give examples here. The manuscript is still too long and lacks focus. I would argue that this is not simply a matter of style—it is the authors job to distill their results into new insights, and this paper does not do that. Important points about the orientation of the structures in 3d from Dr. Prior were essentially dismissed (in response to about 5 paragraphs, all of which I found insightful, it seems the authors only added the sentence in A2_2).

Other revisions introduced problems of their own because of the flippancy—for example, the section headers in the introduction do not match the text of the paragraphs since the structure was shoehorned on without corresponding changes to the body, which leads to confusion for the reader. Shortening this paper does not have to mean cutting content. Much of length is due to needless complex sentences, extraneous adjectives and adverbs, and a kind of personification where "comprehension" or

"understanding" of phenomena are needlessly substituted for the phenomena themselves. Simplifying the language, and making it more precise in doing so, would go a long way toward letting the reader access the full range of ideas in the paper. Other opportunities for very easy shortening include the multiple places where figure captions are essentially repeated in the main text. I think that the final line of Dr. Montagnat's original review still describes the manuscript: "At the end, the novelty of the study stands only on the fact that these deep core measurements have never been published. Care should be taken in the discussion in order to focus on what is potentially new or relevant for the community."

Thank you for these comments. Throughout the revised manuscript, we attempted to shorten the main text, to remove the repetition, to make stories simpler, and to improve English language. English language was checked by a native speaker.

In the revised manuscript, we focus on the deep sections (below 2400 m). To simplify the story, we focused on the crystal orientation fabric and microstructures at the deep sections. Since the detailed discussions about layered structures is deviated from the main story, we shorten them to a minimum.

Discussions about detailed comparison with other ice cores, a-axis anisotropy, and overall layered structures were excluded. We will publish these topics in the future. Furthermore, Microstructure Section was moved to Appendix. Microstructural information is necessary for the discussion of crystal orientation fabric; however, it is a supporting information.

We shortened the Introduction section and focused on deep sections. Then, section headers were removed.

Regarding the 3D structures of layers and c-axis clusters, as commented on by Dr. David Prior, we examined the relative orientation between the layers and the c-axis clusters by analyzing the layer inclination angles in three dimensions, focusing on regions within or near the thin-section planes used for the Laue X-ray diffraction method (results are shown in Figure 4). Among the measured thin sections, 20 samples contained a visible layer nearby. In 15 of these samples, we confirmed that the horizontal orientation of the c-axis clusters and the normal axis of the layers lie within the approximately same vertical plane. For the remaining five samples, it is likely that either half-core inversions or thin-section inversions occurred. While we did not investigate all the layers (~500 layers in total), and there may be minor discrepancies (with maximum measurement errors estimated at around 10 degrees), we concluded that the c-axis clusters and the normal axes of the layers are generally aligned in the same (or near) vertical plane(s) throughout the lowermost 20% of the DF ice core.

We added explanations in Section 3.3.

*"The relative orientation between each layer and the c-axes cluster was assessed by investigating the angle of inclination of the layer near the thin-sections used for the Laue X-ray diffraction method. These data confirmed that the horizontal direction for the c-axis cluster and the normal axis of each layer were within the approximately same vertical plane throughout the lowermost 20% of the DF ice core."*

I have additional scientific qualms as described in overall comments. I have not gone through and made an exhaustive list of detailed comments, since I think hope that structural issues will result in major changes to the text first. I am not an expert in microstructure the way that Drs. Montagnat and Prior are, so I do not focus on the rigor of methods of the observations, or their interpretation in terms of processes at the crystal scale. Instead, I focus on interpreting the results in the ice-sheet setting, the implications for ice flow and radar, and the structure and writing of the paper.

Overall comments:

The paper lacks focus. While the data are undoubtedly complex and extensive, I strongly disagree that this means that the paper cannot tell a story or stories in the way that the response to review claims (and indeed The Cryosphere's evaluation form seems to agree with me). Without better structure and clearer writing, those in very close but not identical fields (including those like myself who have worked extensively on crystal orientation fabric from other perspectives) cannot glean meaningful insight from the work. Put differently, while I agree that "Over-simplifying this paper would compromise its scientific message," the authors have instead under-synthesized and thus obscure the scientific message to all but themselves or similar experts on ice-core microstructure. Part of the issue is contextualization within the literature as Dr. Montagnat noted before. For several of the conclusions, there are citations she mentioned that make identical statements—this makes it exceedingly difficult for somebody with adjacent expertise to understand the value of the present work. I cannot see how the paper really touches on deformation the way the title suggests, other than providing some (unsurprising) evidence of simple shear in the bottom of the ice sheet. I think the paper is currently inaccessible to most of the readership of The Cryosphere, largely due to its presentation not its inherent complexity.

As answered above, we focus on the deep sections (below 2400 m), and common and unique features of the site. Our new findings of the crystal orientation fabric and microstructures obtained by dielectric tensor measurement, Laue X-ray diffraction methods, and microstructural observations can provide new knowledges about the rheology at the deep sections of ice sheets. To make it easier for readers to understand, some contents that deviated from the main stories were excluded.

As pointed, some conclusions are same as previous scientific claims from other ice core studies. We added reference to EDML and EDC ice cores in Conclusion section.

Using the UP80% and LO20% framework leads to misinterpretation, convoluted language, and incorrect conclusions. Take the first conclusion labeled (i). It ends with "at the bottom of the UP80% and fluctuated in the LO 20%." This is not correct (the abstract really indicates that the fluctuations begin at 2650 m, as do the figures). It would be clearer and more precise to write "down to 87% of the ice thickness and fluctuated below. " The zones are confusing, both here and elsewhere, since the results described (e.g., properties to 2650 m) cross zones. Overall, getting rid of these UP80% and LO20 acronyms would reduce

confusion. This is particularly important for the abstract, which currently involves tortuous language to fit the results into this artificial framework.

We agree the comment. We don't use the term "UP80%" and LO20%".

The change to structure in response to the first round of review is window dressing; in the introduction, the section titles do not match the sections in the new version, so they create confusion rather than insight. The section titles in the introduction might be okay in principle, but they would need text to match. However, they would need to be continued in some meaningful way further into the paper. As is, the section headers in the introduction suggest broad implications that the paper fails to deliver upon. I see no major implications for ice-sheet dynamics from this work. I would be happy to be wrong, but the authors would have to provide a structured argument (i.e., a story) that shows why such interpretation is supported by the work.

We shortened the Introduction section and removed section headers.

The sentence-level errors are largely fixable during production, but some are substantive and need to be fixed before acceptance. This issue is most prominent in the introduction. Some sentences are imprecise to the point of being incorrect. It is not my job to rewrite them, but here is one example: "In these studies, the influence of anisotropy on the movement of ice aggregates is so substantial that comprehending its impact on the expansive flow patterns of ice sheets is essential" (L70-71). Despite the long sentence, the authors do not argue what this influence is essential for (maybe projecting sea level rise?), define what ice aggregates are (though I surmise this means bulk ice?), or indicate why "comprehending" rather than just the impact itself is important. In this case, simplifying to "The anisotropy is sufficiently strong to affect bulk deformation" would be less wordy and remove imprecision. Again, I am not trying to say that the authors should use my version of this sentence, but ambiguities like this are not made precise in the production process, and there are enough of them that I cannot flag them all. Instead, I think some careful editing by the authors themselves, really focusing on what they can support with evidence (in the above example, that anisotropy influences bulk flow) and not opinion (in that example, that anisotropy is "essential" or that comprehension is necessary).

Throughout the manuscript, we attempted to improve English language.

I find the response to the orientation issues identified by Dr. Prior in the previous version too flippant. As he pointed out, lying in the same vertical plane is different than saying closely aligned (i.e., I do not think the changes answer the issue identified in Figure R1 in his review). For examples of continuing issues involving angles, in section 3.1.2, I had to re-read 3 times to understand how the principal-frame permittivity was identified. While I think I now understand what was done, I do not see why it was necessary to use other measurements of c-axis orientation to do the rotation if the plane matches the normal to the layers. In this vein, I do not understand why Figure 3 is worth plotting—are not these data essentially meaningless without knowing how the section was cut relative to the azimuth of the cluster

maximum? Perhaps this just requires methods clarification to say how the thick sections were cut, if it was done in such a way that gives these measurements meaning, but if I understand Rev2_10 correctly then the correction happens after the DTM measurement. Dr. Prior provided great suggestions for plotting the layer normal (which he called the pole) on the same stereonet as the c axes—I really want to see these on Figure 4. They do not match the clusters very well (Fig 5f), and the reader has no idea how well they align from what is essentially a single sentence added in response to the original comment.

A major challenge in our study was that the c-axis concentration axis inclined with increasing depth. Moreover, at the time of DTM measurements, we initially did not know how much the c-axes cluster axis was inclined in the (i.e., direction orthogonal to the slab plane) of the slab plane where the electromagnetic waves were irradiated.

Since the c-axis concentration axis was inclined relative to the core axis, we intentionally rotated the core (while maintaining orthogonality with the electromagnetic waves) to capture the two birefringent resonant components obtained from the polarized wave transmission. This allowed us to measure the dielectric anisotropy.

However, the dielectric anisotropy values obtained in this manner were based solely on the components relative to the direction of electromagnetic wave irradiation. Therefore, it remained necessary to convert these values into corrected dielectric anisotropy values, assuming no inclination in the depth direction (i.e., the properly corrected values), using information on the degree of tilt in the depth direction of the c-axis concentration.

Fortunately, we had retained data on the relationship between the thin-section cutting direction used for Laue diffraction and the DTM measurement of the core. This made it possible to perform the necessary corrections for the 'depth' direction of electromagnetic wave propagation. Without the data from the Laue method, correcting for the tilt in the depth direction would have been impossible.

Thus, the initial action of tilting the core was simply to obtain the values of the two birefringent components, even when the c-axis concentration had a rotational orientation in the depth direction. It was only with the data obtained from the Laue method that we could perform corrections for the c-axis concentration's inclination in the depth direction. In other words, the first action was not a correction but a basic measurement, and the second action was the actual correction.

Additionally, in this study, the observation and measurement of visible layer inclinations were conducted independently of the Laue and DTM measurements, without specific attention to the orientation of the tilt. During the initial drafting of the manuscript, we assumed that the tilting direction of the layers and the tilting direction of the c-axis would naturally be the same, without considering any physical principles that could cause discrepancies (such as torsion). However, following Dr. Prior's suggestion, we examined differences at 20 sites with available Laue data and confirmed that there were essentially no discrepancies between the tilting directions of the layers and the c-axis concentration.

We also received a request to plot data on a Schmidt net. Drawing orientation of layers and their normal axis on a Schmidt diagram is a good way to indicate the relative relations. However, we have already spent a significant amount of time processing a large volume of data in this study. Additionally, visual

inspections of layer orientation are not that accurate because of the errors involved. Although we have not plotted the data on a Schmidt net, we provided orientation information in Supplementary Information 2, which we believe will effectively convey the necessary details to readers.

We showed Figure 3 as measured raw data. As commented, these data cannot be used for the discussion of crystal orientation fabric without correction.

Observations and interpretations are still mixed. Section 4.5.3 is the clearest example—each of the paragraphs mixes observation and interpretation. Section 5.2.1 is results.

Thanks for pointing out. In the original manuscript (first submission), Microstructures section (4.5.3) was divided into two sections (in Results section and Discussion section). However, in accordance with comment of reviewer 2, we combined we merged into one section. In the revised manuscript, we move to Appendix as "Microstructures: Results and interpretations". Microstructural information is necessary for the discussion of crystal orientation fabric; however, it is a supporting information.

Section 5.2.1 (Depth-dependent variations of $\Delta\varepsilon$ in the entire core) was deleted.

The use of the phrase "statistical significance" in the paper is incorrect. At times, I think I know what it means, while at others I do not even have a guess. Usually statistical significance means an ability to distinguish between possibilities above a pre-specified level of confidence. The paper essentially uses it to mean "precise, " but with no quantification. This is unacceptable. We need clear definitions whenever the term is used. As a hypothetical, this could be "the difference between measured c-axes distributions at 2500 and 2600 m is statistically significant (p=0.001)" or something similar. Including the confidence and the alternatives considered is required for proper usage.

Thanks for pointing out. We used this term with the intention of including a large number of grains in the thick-section-measurement. We don't use the term "statistical significance" in the revised manuscript.

I do not think the authors' interpretation of the relationship between c axes and layer orientation near the bed is correct (line 511, Figure 8c). Layer slape is not solely caused by rotation. For example, this is my 30-second version in Illustrator—after rotation, subsequent pure shear in the vertical does change layer slope: The language is too muddy for me to understand if this is causing misinterpretation. Overall, no explanation that treats both layers and c axes as passive tracers can cause this deviation. Instead, this could be easily explained a result of intracrystalline slip. Particularly in the conclusions (L830), it sounds instead like the authors disagree with the cartoon version above, and instead are imagining that the pure shear has no effect on slope. Adding in the actual mechanism of c- axis rotation would help.

In the review comment figure, the ice is first rotated by 30 degrees (introducing a slope) and then subjected to pure shear, which changes the slope. However, we believe that it seems natural that the transition from pure shear to simple shear would occur progressively.

Our claim is that the observational fact (the apparent deviation between the layer and the c-axis cluster), observed only in the deeper sections, can be explained by the occurrence of simple shear. We assert that the deviation between the normal axis of the layer and the c-axis cluster is due to the nature of simple shear, which involves both rotation and pure shear.

We agree with the use of "intracrystalline slip". When an ice body undergoes simple shear deformation, it experiences rigid body rotation as a fundamental characteristic of simple shear. Within each crystal grain, intracrystalline slip tends to rotate the c-axes toward the compression axis of the simple shear. We added this explanation in the revised manuscript. (In the previous version, we referred this mechanism, i.e., dislocation creep, only in the Figure caption.) Here, we proposed the mechanism whereby normal components of the strain (both compression in the near-vertical and extension near the horizontal plane) cause the rotation of c-axes cluster toward the vertical, and the deviation between the normal axis of the layer and the c-axis cluster. We added this explanation in the revised manuscript (second paragraph in Section 5.1) as a possible mechanism.

We hope our explanation is clear and understandable.

The conclusions are excessive, variously going beyond the results, repeating themselves, and restating ideas that have long been known. I do not really understand how the first (i) and (ii) differ. Both of these conclusions look nearly identical to other locations (e.g., EDC, (Durand et al., 2009)),and given the emphasis on comparison between sites this should be cited. The second (i) is not supported by the results—at most there is evidence that right at domes climate controls the fabric, but this paper shows nothing about how that might look even 5 km off the domes. Similarly, I do not buy the second conclusion (v); the work shows very little about how large-scale deformation is taking place. The first (vii) is very similar to claims from work at EDML (Weikusat et al., 2017), and the idea was originally suggested in the 90s (Castelnau et al., 1996). While there is nothing wrong with concluding the same thing as previous work, the reader needs context. A much shorter conclusion section, focusing synthesizing rather than repeating key results and contextualizing them rather than providing so much detail, would allow the reader to see how this work fits into the literature.

We reconstructed Conclusion section. Some contents were deleted and merged, and the reference to EDML and EDC ice core was added. (We also added reference to simple shear deformation at the deep sections in the EDML ice core in Section 5.1.), Furthermore, we removed two conclusion categories.

We removed second (i) and (ii), instead, added following conclusion (as conclusion (ii)):
"*The general trends exhibited by the c-axis cluster strength and grain sizes in the present work were approximately the same as those seen in the EDC ice core (Durand et al. 2009). These similarities may be attributed to equivalent impurity concentration profiles (which in turn are associated with climate*

*change) and temperature profiles. Hence, it appears that rigid body rotation does not affect cluster strength.*"

We consider that the commonality of the fluctuations of the cluster strength between the DF and EDC ice cores is derived from the depositional features from atmospheric aerosols. Actually, dust flux profiles of the DF and EDC ice cores are very similar (DFICPM, 2017).

As pointed out, the drilling site of DF ice core is very close (within 10 km) to but not a true dome summit at the present condition. The c-axis fabric exhibited slightly elongated single pole fabric. However, the elongation is very slight and the profiles of c-axis cluster strength between DF and EDC above 2400 m are very similar (Saruya et al. 2022b). We believe that the DF ice core can be used as dome ice core.

In the revised manuscript, the following explanation is added in the second paragraph in Section 2, "*At present, DF is far from the true dome summit and so DF is subject to deviatoric stress in the direction of maximum inclination.*"

Second (v) was deleted. As commented, this was not a conclusion but an implication.

**Detailed comments:**

Title: The title does not really make sense. I cannot tell if the overall layered structures are a wholly separate thing considered at all depths or whether the crystal orientation fabric is within them.

We removed "overall" and "deformation regime" from the title. We modified the title to make clearer as follows: "*Development and fluctuation of crystal orientation fabric in the deep sections of the Dome Fuji ice core, Antarctica: impacts of dust particles and migration recrystallization*"

Abstract: The abstract is too long and detailed, to point that I had to read it three times to figure out what the manuscript was showing. The paper would greatly benefit from shortening the abstract to meet normal length standards (i.e., 250 words, or at least ~300 as it was before, rather than 450). As discussed above, this does not necessarily mean cutting content. For example, it would benefit both accuracy and word count to delete the first words of the abstract "An in-depth examination of."

We shortened the Abstract in the revised manuscript.

L24,184: Statistical significance cannot stand on its own like this. I assume that the authors mean that they have measured enough grains for the results to approximate the underlying c-axis distribution with some amount of precision (perhaps that it differs from isotropy, but I truly do not know). But a measurement itself cannot be statistically significant in the sense used here.

We don't use the term "statistical significance" in the revised manuscript.

L35: "dislocation creep is the primary deformation mechanism in polar ice sheets" cannot be concluded from this work. There is no reason to think that these results can be extrapolated to the ice sheets writ large.

We removed this sentence from Abstract.

L45-50: This paragraph has multiple logical jumps to try to get from the ultra-broad (sea-level rise) to the ultra-specific (dynamic layer structure). Rather than filling all these gaps, I suggest starting less broad, since the readership of the cryosphere surely knows what ice sheets are and that they contribute to sea-level rise. I recognize that this suggestion is stylistic as well as substantive, but if not heeded then we need the gaps filled in (e.g., how is dynamic layering a process challenge that affects models, why is continuous improvement in models needed rather than one really big improvement, how does a single 11-year-old reference indicate ongoing concern).

We reconstructed the Introduction section.

L68: Abbassi is not a good reference for SPICEcore fabric, they measured fabric loosely from IceCube. Voigt 2017 is the correct reference (https://doi.org/10.15784/601057).

The relevant paragraph was deleted. Thank you for pointing out.

L70: The correct reference for the full EastGRIP fabric is Stoll et al., 2024 (10.5194/egusphere-2024-2653)

The relevant paragraph was deleted. Thank you for pointing out.

L81-85: references needed

The relevant paragraph was deleted.

L220-221: How can a plane and a vector be in the same plane unless the vector is in the plane? I think this probably means that the normal to the layer is in the same vertical plane as the COF cluster?

Yes, we checked whether the c-axes cluster and normal axis of each layer are within the same vertical plane.

L223: What are non-principal components? The components in an arbitrary xyz reference frame? I have not seen this terminology before and think it needs to be defined.

We had to measure the dielectric permittivity at an angle deviated from the principal axes of the tensor (which happened to correspond to the angle at which the sample plates were prepared), because the ice core had been rotated along with the inclination of the C-axis concentration axis. This meant that we were not directly measuring the principal components of the dielectric tensor, and it was necessary to apply corrections for the inclination and rotation of the ice core. The information required to perform these corrections was provided by fabric measurements using the Laue method. The measured values that deviated from the principal axes were referred to as "non-principal components.

We added explanations as follows in second paragraph in Section 3.1.2:

*Since the electromagnetic wave used in the DTM method has a transverse electric field, if the surface of the actual plate-shaped sample does not align with the principal axes of the crystal tensor, only the components in the misaligned orientation will be obtained. This misaligned orientation is referred as non-principal.*"

L227: Still confused since I do not know what these are, but are they really "adjusted" or are they rotated in some way (thus changing the value)?

For the correction, we derived a correction factor that estimates how much the dielectric anisotropy would increase if the ice core with the measured dielectric anisotropy were rotated until the electric field of the incident electromagnetic wave became parallel to it.

L276-287: This is a repetition of the figure caption, not results.

We removed the description of Figure 4 from the main text.

L436-437: This sentence needs to be re-written. It is unclear if this means temporal changes or time differences that result from the depth-age relationship of ice sheets.

We removed "in the past". We believe that the steady state grain size in impurity-rich layer established after these layers reached to deep sections (i.e., warmer temperature). Therefore, the depth (temperature) is important.

L497-501: This is a repetition of a figure caption, far from the original appearance, not discussion

We removed the description of Figure 5 from the main text.

L648-655: This discussion of twinning is convoluted. Starting with a simple definition would help.

We removed the discussion of crystal twinning because this content deviated from the main story.

L725: I do not think this really differs from Durand—it is a different location, and there may simply be regional or local differences

This paragraph was removed. In the previous paper, we found the fluctuations of *c*-axes cluster at a few hundred-meter depth at Termination I. We consider that the vertical compression is predominant and the effect of simple shear is not sufficient at shallow depths.

L774-778: The implications for radioglaciology need to be clarified. The idea that recrystallization is important near the bed is not new, nor is the evidence of simple shear which also been noted previously near the bed at ice-core sites (for example, it is suggested at EDC by Durand).

The relevant paragraph was deleted.

Figure 1: Confusing to use red and blue on contours in c when they do not match the colorbar in b. If two colors are needed, it should be explained in the caption (and the colors should not be red/blue, or why not just use a single color?).

We added follow explanation in the caption:

"Thin and thick areas are shown in red and blue respectively, and the boundary is set at a thickness of 3000 m"

Figure 8f: The label is confusing. The LO20% spans almost 90% of the thickness.

This figure was removed. Thank you for pointing out.

**How the authors addressed the comments by Reviwer 4 in their responses to the revised manuscript**

In this document, we explain how we addressed the comments provided by Reviewr 4 on the revised manuscript.

**Explanation of the text colors:**

➢ Black and blue text: Comments from Reviewer 4

➢ Brown text: Explanations of how the authors addressed Reviewer 4's comments in the revised manuscript

This manuscript presents detailed data on fabric evolution and microstructures in the 3035 m long DF2 ice core from East Antarctica, drilled at the Dome Fuji station. In earlier work the same research group published similar data from the uppermost 2400 m of the core, but here the focus is mainly on the depth interval 2400-3035 m. An overview is, however, also given of the evolution of the above mentioned parameters throughout this deep ice core.

A major scientific effort lies behind the results presented and the data are of great importance for studies of the internal structure and deformation properties of large ice sheets. Both state-of-the-art methods and new techniques are used to measure and characterize fabric and grain structure at relatively high resolution down the core. Methods are mostly well described and sources of error are adequately adressed in the paper.

Although some assertions made by the authors will be met with criticism – in spite of improvements made from the initial version – the importance and detailed description of the data is such that the paper should be published in The Cryosphere. This reviewer does not have specific criticisms, but asks the authors to take into account the comments and corrections suggested below.
* * *
Saruya et al. focus on main processes believed to control the development of fabric in ice under the conditions prevailing at Dome F:

• C-axis rotation under compressive stress leading to the development of a single pole fabric

• Rotation recrystallization and subsequent splitting of a grain into 2 or more grains

• The role of simple shear in the fabric evolution

• The interplay between inclined layering (from horizontal, due to bedrock topography) and the mean direction of the single pole fabric

• The effect of impurities and grain size on c-axis rotation

• Nucleation of new grains with c-axis orientations differing from that of surrounding older grains, at relatively high temperatures in the lowest 20% of the ice sheet

A clear picture emerges on the development of c-axis orientations in the DF2 core and interesting data on a-axis orientations are presented as well.

We thank the referee for insightful reviewer.

Perhaps surprising is the fact that this paper (as well as the companion paper on the upper 80% of the ice core, Saruya et al., 2022), does not discuss the grain size development in the core in any detail. Grain growth receives fleeting mention and some insight can be gleaned from Figures 5-7 and Figure 10, but no information is given on normal grain growth rates. If the authors aim to treat this topic in another publication, this should at least be stated, since studies of crystal size and crystal orientation are closely intertwined.

We added the discussion about grain sizes in Section 5.2 as follows:

*"Above 2400 m, the grain size grew steadily (but with a partial decrease during the two glacial periods) (Azuma et al., 1999, 2000). However, below 2400 m, there was no overall increase in grain size. During interglacial periods, the grain size gradually increased with depth but decreased sharply during glacial periods that were associated with higher impurity concentrations. At the lower glacial ice boundary, the grain size again became smaller, indicating that grain size growth was interrupted during glacial periods. This pattern has also been reported to occur in the EDC core (Durand et al., 2009). The reason why grain growth in deeper ice is interrupted during glacial periods remains unclear, although this effect may be due to nucleation or migration recrystallization. Figures 5c and A2 show signs of nucleation and migration recrystallization, both of which can reduce the mean grain size, in interglacial ice specimens. These phenomena may suppress normal grain growth and lead to smaller grain sizes. Furthermore, extremely large grains exceeding 10 cm in radius (~300 $cm^2$) were observed below 2960 m. Grains of this size have not been found in the EDC core to date (see Figure 7). Interestingly, small grains (less than 1 $cm^2$) were maintained at a depth of approximately 2900 m (MIS16), despite the high temperatures at that depth that were close to the melting point of ice (–5 °C). Although the concentration of dust particles was not so high in this region (less than 100 ppbv), the grain boundary pinning effect caused by dust particles remained still effective even at high temperatures."*

This reviewer has access to two reviews on a previously submitted version of this manuscript, as well as to the replies given by the authors. It seems to me that the authors have made a major effort in responding to the major criticisms offered by the two reviewers, both of whom are leading experts in this field. Examples:

Reviewer 1 criticised the authors´ emphasis on grain boundary sliding (GBS) in the earlier version, stating that the evidence for this deformation mechanism in the core was very limited. The authors have responded to this by significantly toning down the potential occurrence of GBS in the revised version.

Reviewer 1 is critical of the authors´ interpretation of small grains on the boundary between two larger grains as evidence for nucleation of new grains (sometimes referred to as dynamic or migration recrystallization) – see Figure 7. It also seems like Reviewer 2 does not support the authors´ interpretations on this issue. Author responses are detailed and they defend their view by pointing out that observed c-axis fabric changes at depth support the assertion that new grains are being nucleated.

It may be added here that it is surprising how little discussion there is in the paper on the effect of higher temperatures near the bed on grain size development. Section 4.4 mentions very large crystals below 2950 m depth (blue curve in Figure 5), but there is no further interpretation. Grain size curves between DF2 and EDC are compared in Figure 10, but a comparison with the Vostok core – where very large crystals were observed in the lowest part of the meteoric ice – would have been particularly interesting here.

We added grain sized data below 2950 m as Figure 6. but could not find the grain sizes data in the deep sections of Vostok core.

Even though the authors have made efforts to shorten the text (following reviewer advice), the paper is still very long and the text is often repetitive. Actually, the repetitions are at times helpful for the reader, who gets a bit lost at times in the details being discussed, and is thankful for a recap! The Conclusions section (2.5 pages) presents a very good overview of main results and interpretations, in a better way than the bulk of the text.
The English language of the paper still needs improvement and this reviewer typed down suggestions as he read the paper through. Hopefully these suggestions can be of help in the final revision process.

Thank you for these comments. Throughout the revised manuscript, we attempted to shorten the main text, to remove the repetition, to make simple story, and to improve English language. English language was checked by a native speaker.

Specific comments:

Comments/questions directed to the authors are blue-coloured for clarity. Note that the terms grain and crystal are used synonymously by this reviewer. Numbers like L20 refer to line numbers in the manuscript version: egusphere-2023-3146-manuscript-version2.pdf

L20

To examine the distribution and texture of the c- and a-axes...

→

To examine the orientation distribution of the c- and a-axes... („texture" refers more to grain size and shape)

We modified the sentence.

L22

detailed crystal grain orientations

→

detailed crystal orientations

We modified the sentence.

L24

as eigenvalues

→

as eigenvalues of the orientation tensor

We modified the sentence.

L28

Below 1800 m in the UP80%, layers with varying dusty impurities

→

Below 1800 m in the UP80%, layers with varying concentrations of dusty impurities

We modified the sentence.

L38-39

What are c-axis layers ??

Wording was not good. We modified to "c-axis cluster"

L53-54

whereas shearing on alternate slip systems is significantly more difficult, nearly a hundredfold

→ (suggestion)

whereas nearly a hundred times greater (shear) stress must be applied to induce deformation along other slip systems within the crystals

We modified the sentence.

L57-58

The study of anisotropic ice deformation has been conducted historically. They were done through theoretical research

→

Anisotropic ice deformation has been studied theoretically

The relevant paragraph was deleted.

L59

and through laboratory experiments

→

and in laboratory experiments

The relevant paragraph was deleted.

L61-62

These laboratory-based studies are characterized by experimental setting of strain rate by more than several orders of magnitude, and under temperature close to melting point,

→

The experimental setting in these laboratory-based studies is characterized by strain rates that are several orders of magnitude higher than in situ strain rates in ice sheets, and at temperatures close to the melting point,

The relevant paragraph was deleted.

L62-64

Therefore, these laboratory-based knowledge can be a valuable reference mainly for such conditions.
→ The laboratory results are thus, strictly speaking, not directly applicable to ice-sheet conditions.

The relevant paragraph was deleted.

L64

play another essential role. Examples of Antarctic deep ice core include
→ play an essential role. Examples of Antarctic deep ice cores include

The relevant paragraph was deleted.

L68-69

Examples of Greenland ice sheet include

→

Examples from the Greenland ice sheet include

The relevant paragraph was deleted.

L70

and among others

→

and other cores

The relevant paragraph was deleted.

L72-74

For instance, during the internal deformation of the ice sheet, the flow forms preferred orientations of crystal axes, at the same time, the ice flow is modulated based on them.

$\rightarrow$

For instance, the internal deformation of an ice sheet leads to the development of preferred orientations of the ice crystals, which in turn influence the flow of the ice sheet.

The relevant paragraph was deleted.

L83-84

dome regions in central area of the ice sheets

$\rightarrow$

dome regions in the central parts of ice sheets

The relevant paragraph was deleted.

L95

was presented by

$\rightarrow$

was described by

The relevant paragraph was deleted.

L94

and it is suitable to investigate

$\rightarrow$

and such a setting is suitable to investigate

The relevant paragraph was deleted.

L98

deepmost $\rightarrow$ deepest

The word was modified.

L104-105

composition of the deepest 1% thickness (3000–3035 m).

$\rightarrow$

Composition of the deepest 1% of the ice sheet/ice core (3000-3035 m).

We modified the sentence.

L105-106

These properties were found to retain the basic layered structure of ice core signals except in the deepest few meters.

Not certain what this sentence means – was the size and concentration of air hydrates and the oxygen isotope ratio typical for the meteoric ice above (except in the deepest few meters) ?

Yes, we observed ubiquitous presence of air hydrate and the water isotope composition comparable to the upper part of ice core (i.e., continuous profiles from the upper part).

L108

sampled at every 11 m depth.

→

sampled at 11 m depth intervals.

The relevant paragraph was deleted.

L111-113

Dynamic recrystallization within ice sheet has been widely investigated historically, as reviewed in papers or textbooks (e.g., Poirier 1985; Humphreys and Haterly 2004; Faria et al., 2014a, b) or individual papers (e.g., De La Chapelle et al. 1998; Weikusat et al. 2009, Kipfstuhl et al. 2009, Montagnat et al. 2012, 2014; Stoll et al., 2021a).

→ (suggest change to)

Studies on dynamic recrystallization in ice sheets have been reported by e.g. Poirier 1985; De La Chapelle et al. 1998; Humphreys and Haterly 2004; Weikusat et al. 2009; Kipfstuhl et al. 2009; Montagnat et al. 2012, 2014; Faria et al., 2014a and Stoll et al., 2021a.

We modified the sentence.

L113-114

Exploring deep ice rheology and crystal properties, dynamic recrystallization is one of main

focuses in this paper.

→ (suggestion)

In this paper, we take advantage of the unique opportunity that the Dome F ice core offers to study the role of dynamic recrystallization in the formation of textures and fabrics in the deeper parts of the East Antarctic ice sheet.

We modified the sentence.

L115

What does „Advancing ice sheet dynamics" mean here? Would it not be more suitable to shorten the title of this section to: The development and role of crystal orientation fabrics

This sentence was removed.

L117

„layer structures of the c-axis fabric" is not very clear wording.

Wording was not good. We modified to "development of the c-axis fabric".

L131

variations in the deformational history of the vertical compression.

→

variations in the deformational history of an ice-sheet region where vertical compression is the dominant stress regime.

The relevant paragraph was deleted.

L135

across wide area of ice sheets.

→

across wide areas of ice sheets.

We added "s".

L141-143

Following this previous study, research focusing on the lowermost (deepest) approximately 20% thickness zone (ranging from about 2400 m to the deepest ice at 3035 m) remains to be done, to examine textural data for the entire thickness of the ice sheet.

→ (suggestion)

Following this previous study, research has focused on the lowermost (deepest) approximately 20% of the DF core (depth interval 2400-3035 m), to complete the picture of textural and fabric evolution in the entire ice sheet column.

The sentence was modified.

L147

We compare textural data by these methods with various data analyzed from the ice core.

→

We compare textural data obtained with these methods with various data sets from studies of the ice core.

The sentence was modified.

L148-149 (twice)

crystal orientation fabric

→

 the crystal orientation fabric

We added "the".

L153

that correspond to ages of more than 1 million years

→

covering snow deposition over more than 1 million years

The sentence was modified.

L155

extends → extend

We removed "s"

L157

knowledge of englacial layers under various ice conditions.

→ (suggestion)

knowledge of the physical properties and the deformation history in different regions of the East Antarctic ice sheet.

The sentence was modified.

L167-168

In terms of the c-axis fabric, Azuma et al. (1999, 2000) reported that at DF1 the c-axis fabric exhibited the elongated single pole fabric as the dome undergoes deviatoric strain depending on orientations.

→

This is not very clearly worded – what does „depending on orientations" mean here ? Are the authors referring to the mean c-axis orientation being perpendicular to the bedrock slope (line 172) ?

DF is very close (within 10 km) to but not a true summit at the present condition. Therefore, DF is subject to deviatoric stress in the direction of maximum inclination. We added this explanation in the second paragraph in Section 2.

L191

signals of dynamic recrystallization

→

signs of dynamic recrystallization

The word was modified.

L195

The principle of...have been described

→

The principles of...have been described

Or

The principle of...has been described

The sentence was modified.

L219-220

deviating from the vertical (hereinafter, inclination angle) in the same direction of the maximum layer slope.

→ (suggestion)

deviating from the vertical (hereafter, inclination angle) in the same direction as the normal to the maximum layer slope.

The sentence was modified.

L266-267

Data from both cases....in agreement

→

Data from both cases....are in agreement

The sentence was modified.

L302 (check also L304 and L306)

generally have periodicity of 60 degrees

→(suggestion)

generally display maxima at 60 degree intervals („periodicity" normally refers to regular variation with time)

The sentence was modified.

L314-315, this sentence needs improvement/rewriting to become intelligible:

„It is also very important to note that the SDρ is well synchronized with the grain size (Figure 5e), which implies underlining Physics."

Wording was not good. We removed "which implies underlining Physics".

L323-324 (Fig. 4 caption)

The vertical orientation aligns with the core axis.

According to Fig. 5 the borehole is inclined 3-6° in the depth interval being considered so the core axis hardly represents the vertical.

We modified to "*The green triangle in each diagram indicates the vertical orientation in the ice sheet, which approximately aligns with the core axis (though the borehole inclined 3–6º from the vertical)."*

L351-360

The authors outline grain sizes in the entire core, mentioning that individual grains become „extremely large" below 2950. No figures are given, though, and Fig. 10e seems to indicate grain size peaks exceeding the maximum value given on the vertical axis of that diagram. The limited size of thin sections

is, of course, an issue at these great depths, but it would be good if the authors could at least mention maximum grain sizes inferred from the study.

We added the discussion about grain size in Section 5.2 and closeup up figure of grain sizes below 2950 m as Figure 6.

L361-364

Supporting references are needed for the statement that the c-axis (mean) angle will not coincide with the normal to the shear plane.

We removed these sentences from this section. They are part of discussion.

L381-382 – these two sentences need English-language improvement.

We modified as follow (in Appendix A1):

*"These grains are sparsely distributed, and typically have sizes on the order of a millimetre or less. Flattened (or two-dimensionally elongated) grains having a noticeable slant are also evident."*

L382

These features were unique in the impurity-rich depths

→

These features were present in the impurity-rich layers

We modified the sentence.

L386

tend to be distributed as more straight lines

→

are often displayed as straight lines

We modified the sentence.

L388-389

However, some features of impurity-rich layers observed in panels (a) and (c) are persistently present in (e). That is, flattened grains have slanting features.

→ (suggestion)

However, some features of impurity-rich layers observed in panels (a) and (c) are persistently present in (e), where flattened grains display slanting features.

We modified the sentence.

L391-394 – improve English language

We modified as follow in (in Appendix A1):

*"In this regard, the coarser grains also had a greater effect compared with the more sparsely distributed smaller grains. Because the Δε values represented volume-weighted averages within the microwave beam, these values were decreased to a greater extent in the case that more and/or larger grains with c-axis orientations distinctly offset from the surrounding grains were present."*

L419

reported that → reported

We modified the word.

L435-436

We suggest that the steady grain sizes in the impurity-rich layers established themselves after these layers had reached deeper depths.

→

We suggest that the steady grain sizes in the impurity-rich layers were attained when these layers had reached greater depths.

We modified the sentence.

L484-485

The conditions of ice sheets in Antarctica, in terms of temperature and stress, are located on a boundary zone between dislocation and diffusional creep on the deformation mechanism map

→ (suggested rewording)

Due to prevailing temperature and stress conditions, the ice within the Antarctic ice sheet(s) is located in a boundary zone between dislocation and diffusional creep on the deformation mechanism map.

We modified the sentence.

L486

When ice is under temperatures → When ice is at temperatures

We modified the word.

L497-504

This description of Fig. 5 has already been given earlier on.

The description of Fig. 5 was removed from the main text.

L506-507

the consistency of the inclination angles in the visual layers and the c-axes cluster depends on glacial/interglacial periods.

→

The consistency between the inclination angles of the (normal to) the visual layers and the caxis cluster is affected by varying conditions between ice deposited during glacial and interglacial periods.

We modified the sentence.

L525-533

Are there any results from surface-based GPS-measurements, that could confirm the assumption that ice flow has occurred towards the center of the trough? And since there is a trough there, wouldn´t the internal layers of ice sitting in the trough also be convex-shaped in the absence of basal mass loss due to melting?

There is no evidence of horizontal flow near the bedrock. Convex-shaped layers would be formed by bedrock (trough) geometry; however, we believe that inhomogeneous basal melting might be cause the anomalous thinning (increase of thinning function below 2700 m).

L546 – improvement in wording required:

the all the c-axes thus the c-axes cluster rotates

This relevant figure was deleted.

L547

total amount of inclinations

→

total inclination

This relevant figure was deleted.

L558

are plotted in entire depths figure (Figure 10).

→

are plotted for the entire depth of the core in Figure 10.

The relevant figure was deleted.

L566-567

dust particles located → dust particles are located

We modified the sentence.

L568

„Production of dislocation is one of the possibilities."

Does this mean that the presence of microparticles leads to dislocation production? Please clarify!

We modified to "Dust particles can produce dislocations"

L570-572

Item (ii) – it is unclear how the content of this sentence relates to the issue mentioned in L569, that microparticles may act as a sink of dislocations like grain boundaries.

We modified as follow (first paragraph in Section 5.3):

*"It is well known that dust particles restrict grain growth and result in smaller grain sizes (e.g., Alley and Woods, 1996). This trend was observed in the present study. The scatter plot of the Δε values, grain sizes and concentration of dust particles are given in Figure 8. The grain sizes are decreased as the concentration of dust particles increased. However, the role of dust particles in ice deformation via dislocation creep is not well understood. Dust particles can product dislocation or may act as sink of dislocations similar to grain boundaries. Saruya et al. (2022b) suggested two possible actions of particles: (i) restricted deformation due to the inhibition of dislocation by dust particles and/or (ii) the promotion of diffusion creep that does not cause c-axis rotation. Diffusion creep is known to be promoted by smaller grain sizes."*

L579

In the LO20%, some thickness between 2400 m and 2650 m is

→

In the LO20%, the depth interval 2400-2650 m is

This sentence was deleted.

L601

from compressional axis → from the compressional axis

We added "the".

L602

„However, it is also noteworthy that, in Figure 5c, distribution of c-axis density approximately from 30 degrees from compressional axis (60 degrees from horizon) is always denser than 45 degrees or 60 degrees."

Wording is not clear here. Does the sentence imply that there are more c-axes located approx. 30° from the compressional axis that there are at 45° and 60° (or in between) ?

We deleted this sentence.

L603-604

„It is possible that it has some underlying mechanisms in terms of nucleation recrystallization relative to the existing c-axis cluster."

What does „it" refer to here? The sentence is unclear.

This sentence was deleted. Wording was not good.

L608

ice crystals will recover the c-axis orientations available for

→

ice crystals will develop c-axis orientations favourable for

We modified the sentence.

L612

decreasing → decrease

We modified the word.

L613

in the impurity-rich layer → in the impurity-rich layers

We added "the".

L643-644

Be more specific on why rotation recrystallization has a minimal effect on c-axis fabrics and why you assume – seemingly ad hoc – that it does not cause a-axis organization.

The relevant section was deleted. No need to mention the rotation recrystallization here.

L714-715

the thickness of ice is approximately 10% of the original ice equivalent thickness at the time of deposition

→

the thickness of a particular ice layer is approximately 10% of the ice equivalent thickness of that layer at the time of deposition

We modified the sentence.

L726

strengthening the → strengthening of the

The relevant paragraph was deleted.

Table 2

The ice thickness at GRIP was 3029 m (the figure 3085 m is the thickness at NGRIP). The mean annual temperature at GRIP was -32°C during the 1990s, when the deep ice core was drilled. See Dansgaard et al. (1993). Evidence for general instability of past climate from a 250-kyr ice-core record. Nature, 1993, 364 (6434), 218-220.

This table was removed. Thank you for pointing out.

L746

Based on the above discussions, we propose an overview of the layered structure of ice sheets.

To "propose an overview" sounds a bit strange.

The relevant section was deleted.

L759-760

What is meant here by „two distinct conditions? The next sentence mentions „Four types of conditions" which adds to the confusion.

The relevant section was deleted.

L766-767

However, these layered structures correlate directly with the vertical thinning of each layer,

Which „layered structures" and how do they „correlate directly" with the layer thinning? Are the authors referring to the effects of changing grain size, evolving fabric and variations in impurity content on the thinning rate?

The relevant section was deleted.

L775

layers of ice c-axis fabric → ice layers with a particular c-axis fabric

The relevant section was deleted.

L794-95

„changes in depth range" almost sounds like a statement that depth increases with depth!

The sentence is probably supposed to refer to fabric changes with increasing depth.

We modified the sentence. Thank you for pointing out.

L807

The term „Growth of c-axis clustering" appears several times in the paper. The authors should consider if „evolution" or „development" is not a more appropriate term than„ growth" in this context.

We changed to "evolution or development".

L810

The words „are observed" need to be inserted after the word „recrystallization", so that the sentence becomes meaningful.

We modified the sentence.

L820

Possibly, crystal twinning, which share some crystal lattice points symmetrically with the neighboring crystals

→

Possibly, crystal twinning, which leads to crystals sharing some lattice points symmetrically with neighbouring crystals

The relevant conclusion was deleted.

L829

and layers → and the normal to the layers

We modified the sentence.

L848-849

We argue that the bottom thickness of the ice sheet deeper than approximately 2600 m plays a special role in shear deformation

→

We argue that the lowermost part of the ice column, deeper than approximately 2600 m plays a special role in shear deformation

The relevant paragraph was deleted.

L852-853

Ice core drilling, aiming the ancient climatic records, at locations away from the dome area carries serious risks of layer disturbances near the bottom.

→

Ice core drilling projects aiming to retrieve climate records from ancient ice may encounter layer disturbances near the bottom if the drilling sites are located away from dome summit regions.

The relevant paragraph was deleted.

L929

within plane → within the plane

We added "the".

L938 (and similarly in L922):

Here, the c-axes cluster of single pole fabric is inclined to an arbitrary horizontal orientation.

Shouldn´t this rather be (?):

Here, the plane perpendicular to the c-axes cluster of the single pole fabric is inclined to an arbitrary horizontal orientation.

The sentence was modified.

Section A3

The faint and thin cloudy layers are identified when oriented vertically;

What does it mean here to be „oriented vertically? Are the cores turned and inclined until the observer looks straight down ("vertically") on the edge of the cloudy layers?

Yes, layers can be identified only when viewed from horizontally.

L973-974

the aspect ratio value defined as the ratio of the short and long axis of a fitted ellipse

→

the aspect ratio value defined as the ratio of the long and short axis of a fitted ellipse [i.e. long divided by short]

Table A4, column farthest to the left:

Average → Average aspect ratio

The sentence and column were modified.

L1076-1077

Crystal orientation fabric data from the DF core and the EDC core are compared using a common age scale of the DF2021 age scale.

→ (suggestion)

Crystal orientation fabric data from the DF core and the EDC core compared using a common age scale, matching the EDC age scale with the DF2021 age scale.

Did the authors use isotope data or methane data to tie the age scales of the two cores together? If so, this should be stated.

This sentence (caption) was removed. DF2021 age scale uses Bayesian dating model and firn densification model, and constrained by various types of information such as new O2/N2 age markers.